# Online Strategic Classification with Noise and Partial Feedback

**Tianrun Zhao,   Xiaojie Mao,*** **Yong Liang**

School of Economics and Management, Tsinghua University, Beijing, China, 100084
ztr23@mails.tsinghua.edu.cn, maoxj@sem.tsinghua.edu.cn,
liangyong@sem.tsinghua.edu.cn,

## Abstract

In this paper, we study an online strategic classification problem, where a principal aims to learn an accurate binary linear classifier from interactions with sequentially arriving agents. For each agent, the principal announces a classifier. The agent can strategically exercise costly manipulations on his features to be classified as the favorable positive class. The principal is unaware of the true feature-label relationship, but observes all reported features and only labels of positively classified agents. We assume that the true feature-label relationship is given by a halfspace model subject to arbitrary feature-dependent but bounded noise (i.e., Massart noise). This problem faces the combined challenges of agents' strategic feature manipulations, partial feedback observations, and label noise. We tackle these challenges by a novel learning algorithm. We show that the proposed algorithm yields classifiers that converge to the clairvoyant optimal classifier and attains a regret rate of $O(\sqrt{T})$ up to poly-logarithmic and constant factors over $T$ cycles.

## 1   Introduction

Strategic classification studies the problem of learning robust classifiers in presence of self-interested strategic agents. When subjugated to decision-making aided by classification algorithms, agents may strategically modify their observable features to game the classification algorithms into making decisions that best serve the agents' goals. For example, a bank may use classification to determine whether loan applicants are qualified to grant approvals. The applicants prefer positive classification and loan approvals, so they have the incentive to modify their profiles (e.g., credit score), potentially at certain costs, without actually improving their financial status. It is crucial that classification algorithms used for decision-making be robust to such strategic manipulation.

Besides the strategic feature manipulation, another common challenge in classification-based decision-making is that the decision-maker often only observes partial feedback. In particular, the decision-maker may only observe the true labels of agents who have received the positive decision. For example, the bank can observe the true financial qualification only for applicants who have already been classified as qualified and granted loan approvals, but has no chance to observe the true qualification of rejected applicants. This type of partial feedback is sometimes called one-sided feedback, apple-tasting feedback [e.g., Harris et al., 2023, Helmbold et al., 2000] or selective label feedback [e.g., Lakkaraju et al., 2017, Chen et al., 2025].

In this paper, we study an online strategic classification problem with partial feedback. In this problem, a principal (decision-maker) interacts with sequentially arriving agents. The principal

---
*Corresponding author.

39th Conference on Neural Information Processing Systems (NeurIPS 2025).

announces a binary linear classifier to each agent and makes the decision according to the agent's reported feature that may differ from the truth due to strategic manipulation. Following the existing literature, we assume that the agents manipulate their features to maximize the net utility from the classification decision and the cost of feature manipulation. We assume that the agents' true feature-label relationship is characterized by a linear halfspace model with arbitrary feature-dependent but bounded noise (i.e., Massart noise). This model is widely adopted in the learning theory literature (see references in Section 1.1). The principal does not know the true feature-label relationship, but needs to learn accurate binary classifiers from observations of agents' reported features (but not the original true features) and the true labels of only positively classified agents.

Notably, this problem faces the combination of three challenges: agents' strategic feature manipulations, partial label observations, and label noise. First, because of strategic feature manipulation, the agents' true features may not be faithfully observed by the principal, which impedes the learning process. This is particularly a challenge in the online setting, as the agents' strategic behaviors depend on the classifiers announced to them, so their behaviors change over time as the classifiers evolve. Second, the principal can only observe true labels from positively classified agents, without feedback from negatively classified agents. This means that the principal can learn only when a positive classification is made, while the strategic agents are incentivized to manipulate their features to achieve positive classification. Third, the label noise results in noisy feedback, which further complicates the learning process.

Our work contributes to the literature along the following dimensions. First, to the best of our knowledge, our work is the first to study online strategic classification under Massart noise and partial feedback. This advances the literature of learning halfspaces under noise [e.g., Zhang et al., 2020, Diakonikolas et al., 2020] to the strategic setting. Moreover, within the online strategic classification literature, our halfspace model with Massart noise extends the noise-free model of deterministic feature-label relationship in Ahmadi et al. [2021], Shen et al. [2024] and complements the fully adversarial setting [Dong et al., 2018, Chen et al., 2020]. Second, we propose a novel learning algorithm that effectively addresses the aforementioned three key challenges. This algorithm has an initialization-refinement-enhancement pipeline, proceeding in batches and iterations. It features several key components: 1) a localization scheme that iteratively improves the classifiers via online linear optimization, using data within increasingly narrow bands around the classification boundary; 2) a projection-based method to construct proxy features from agents' reported features; 3) a pairwise contrastive inference technique to infer information of the localization bands by contrasting data from pairs of carefully constructed classifiers. Third, we rigorously prove that the proposed algorithm yields classifiers that converge to the clairvoyant optimal one and attains a regret rate of $O(\sqrt{T})$ up to poly-logarithmic and constant factors over $T$ cycles.

## 1.1 Related Literature

**Strategic Classification**  Strategic classification, introduced by Hardt et al. [2016], has gained increasing attention. The existing literature has studied strategic classification in both offline settings [e.g., Hardt et al., 2016, Sundaram et al., 2023, Levanon and Rosenfeld, 2021] and online settings. In online strategic classification, a principal sequentially interacts with strategic agents, aiming to learn accurate classifiers in the presence of strategic feature manipulation. Some literature models agents' strategic behaviors by a manipulation graph that defines agents' feasible feature manipulations [Ahmadi et al., 2023, 2024, Cohen et al., 2024, Shao et al., 2025]. Meanwhile, other literature considers agents that maximize the utility net the cost of feature manipulation. For example, Dong et al. [2018] derive conditions on the manipulation cost function that enable convex optimization techniques to achieve a sublinear regret rate under different fractions of strategic agents. Chen et al. [2020] consider a distance-based manipulation cost function and a zero-one loss function. Our work considers the same cost function and loss function. However, Chen et al. [2020] studies a fully adversarial setting, while our work studies a stochastic setting where agents' true features and labels follow some probability distributions. Our work is closely related to Ahmadi et al. [2021] and Shen et al. [2024], as we study similar models for agents' strategic behaviors and linear classifiers. However, their works focus on the noise-free setting with a deterministic feature-label relationship, while our work tackles label noise.

Notably, nearly all prior studies focus on full feedback settings, whereas our work studies a partial feedback setting. One exception is Harris et al. [2023], where the feedback can be observed also

only under a positive decision. However, they consider continuous feedback following a linear regression model with feature-independent noise and target a different objective. In contrast, our work considers binary classification feedback and studies a halfspace model with potentially feature-dependent bounded noise, which directly extends the models in Ahmadi et al. [2021], Shen et al. [2024].

**Learning Halfspaces with Noise**   Our paper adopts a halfspace model with the label flipped at a potentially feature-dependent bounded probability, i.e., Massart noise [Massart and Nédélec, 2006]. Recent studies find that even in the absence of strategic agent manipulations, learning halfspaces under Massart noise presents significant challenges [Zhang et al., 2020, Diakonikolas et al., 2019, 2020, 2024]. The key challenge stems from the nonconvexity of the 0-1 loss function that characterizes the misclassification error. A standard approach to overcome the non-convexity of 0-1 loss in classification is to use a convex surrogate loss function [Bartlett et al., 2006]. However, Awasthi et al. [2015] show that popular algorithms such as SVM or hinge loss minimization fails to learn a halfspace that achieves arbitrarily small excess error under Massart noise. More generally, Diakonikolas et al. [2019] show that one cannot achieve non-trivial misclassification error for learning halfspaces under Massart Noise by optimizing convex surrogates. Instead, a "localization" scheme has been proposed to learn halfspaces under a variety of noise models [e.g., Shen, 2021a, Awasthi et al., 2017, Zhang and Li, 2021, Shen, 2021b, Awasthi et al., 2017]. In particular, Zhang et al. [2020] and Diakonikolas et al. [2020] apply localization to learn halfspaces with Massart noise. The core idea is to iteratively improve classification via convex optimization, using data within increasingly narrow bands around the classification boundary. This localization scheme focuses more on data near the classification boundary, as data far away from the boundary tend to be less informative since they can be either easily correctly classified or misclassified mainly due to noise. However, naïvely extending this localization scheme to strategic classification poses significant challenges, because data points close to the classification boundary are the most prone to feature manipulation. Our work effectively overcomes these challenges by leveraging carefully constructed proxy data and a novel pairwise contrastive inference approach.

## 1.2   Notation

We employ the following notation throughout the paper. Boldface letters such as $\mathbf{x}, \mathbf{r}, \mathbf{w}$ denote vectors. The operator $\|\cdot\|_p$ denotes any $\ell_p$ norm of a vector. The inner product of two vectors is denoted by $\langle \cdot, \cdot \rangle$ and the angle between two vectors is represented by $\theta(\cdot, \cdot)$, *i.e.*, $\theta(\mathbf{v}_1, \mathbf{v}_2) = \arccos\left(\frac{\langle \mathbf{v}_1, \mathbf{v}_2 \rangle}{\|\mathbf{v}_1\|_2 \cdot \|\mathbf{v}_2\|_2}\right)$ for $\forall \mathbf{v}_1, \mathbf{v}_2 \in \mathbb{R}^d$. Symbols $\mathbb{B}^d$ and $\mathbb{S}^d$ denote the $d$-dimensional Euclidean unit ball and sphere, respectively. $\mathbb{B}^d(R)$ denotes the ball with radius $R > 0$. For any positive integer $N$, $[N]$ represents the set $\{1, 2, \ldots, N\}$. The indicator function $\mathbb{I}(\cdot)$ gives the value 1 if the event within the parentheses holds and the value 0 otherwise.

## 2   Problem Setup

We consider a setting where a principal repeatedly interacts with sequentially arriving agents (e.g., applicants). Without loss of generality, time is discretized into $T$ cycles, where one agent arrives in each cycle $t \in [T]$. The agent is characterized by a feature-label pair $(\mathbf{x}_t, y_t)$, where $\mathbf{x}_t \in \mathbb{R}^d$ denotes a $d$-dimensional feature vector and $y_t \in \{+1, -1\}$ denotes the agent label (i.e., qualified or not). At the beginning of each cycle $t \in [T]$, the principal announces a classifier $\tilde{h}_t(\cdot)$ as the admission rule for the arriving agent. The agent may strategically manipulate and report feature value $\mathbf{r}_t \neq \mathbf{x}_t$ to the principal at some costs, aiming to get admitted (i.e., classified as the positive class, $\tilde{h}_t(\mathbf{r}_t) = +1$). The principal observes the reported features $\mathbf{r}_t$, makes the classification decision $\tilde{h}_t(\mathbf{r}_t)$ accordingly, and observes the true label $y_t$ only when this agent is admitted. Importantly, the principal has no chance to observe the true label of rejected agents. Based on the data of reported features and admitted agents' labels, the principal aims to learn accurate classifiers over the $T$ cycles.

**Distributional Assumptions**   We assume that the feature-label pairs $(\mathbf{x}_1, y_1), \ldots, (\mathbf{x}_T, y_T)$ are independently and identically distributed (i.i.d) draws from a common population denoted by $(\mathbf{x}, y)$. We need to first impose some distributional assumptions on $(\mathbf{x}, y)$.

**Assumption 1** (Halfspace with Massart noise). *There exists a Boolean-valued function $h^*(\mathbf{x}) = sgn(\langle \mathbf{w}^\star, \mathbf{x} \rangle)$ for a coefficient vector $\mathbf{w}^\star$ with $\|\mathbf{w}^\star\|_2 = 1$ and a noise level bound $\bar{\eta} \in [0, 1/2)$, such that $y = h^\star(\mathbf{x})$ with probability $1 - \eta(\mathbf{x})$ and $y = -h^\star(\mathbf{x})$ with probability $\eta(\mathbf{x})$, where $\eta(\mathbf{x})$ characterizes the potentially feature-dependent noise satisfying $0 \leq \eta(\mathbf{x}) \leq \bar{\eta}$ almost surely.*

Assumption 1 is a standard assumption in the literature of learning halfspaces without strategic manipulation [e.g., Zhang et al., 2020, Diakonikolas et al., 2020, Massart and Nédélec, 2006]. It allows for arbitrary feature-dependent label noise with an upper bound $\bar{\eta} \in [0, 1/2)$, and relaxes the assumptions in some existing strategic classification literature that assumes a noiseless halfspace model $y = h^\star(\mathbf{x})$ and that the positive and negative classes are strictly separated by a margin [e.g., Ahmadi et al., 2021, Shen et al., 2024]. Our assumption can more aptly model real applications where label noises are common and even feature-dependent. Nonetheless, unlike the existing literature, we need to simultaneously handle both the strategic manipulation and label noise.

**Assumption 2** (Regular feature distribution). *Fix constants $R, L_1, L_2, U_1, U_2, \delta, Q > 0$, and let $\mathbf{x}_V$ denote the projection of $\mathbf{x}$ onto any subspace $V \subseteq \mathbb{R}^d$ and $\phi_V$ denote its probability density function. The distribution of features $\mathbf{x}$ satisfies the following regularity conditions for any 1-dimensional subspace $V_1 \subseteq \mathbb{R}$ and any 2-dimensional subspace $V_2 \subseteq \mathbb{R}^2$:*

*1. $\phi_{V_1}(\mathbf{x}_{V_1}) \geq L_1$ and $\phi_{V_2}(\mathbf{x}_{V_2}) \geq L_2$ for any $\mathbf{x}_{V_1} \in V_1 \cap \mathbb{B}^1(R)$, $\mathbf{x}_{V_2} \in V_2 \cap \mathbb{B}^2(R)$.*

*2. $\phi_{V_1}(\mathbf{x}_{V_1}) \leq U_1$ and $\phi_{V_2}(\mathbf{x}_{V_2}) \leq U_2 e^{-\delta \|\mathbf{x}_{V_2}\|_2}$ for any $\mathbf{x}_{V_1} \in V_1$, $\mathbf{x}_{V_2} \in V_2$.*

*3. For any $t > 0$ and unit vector $\mathbf{w} \in \mathbb{S}^d$, we have that $\mathbb{P}[|\langle \mathbf{w}, \mathbf{x} \rangle| \geq t] \leq \exp(1 - Qt)$.*

In Assumption 2, condition 1 requires that the densities of any 1-dimensional and 2-dimensional projections of feature $\mathbf{x}$ are lower bounded around the origin. Condition 2 indicates that these densities have proper upper bounds. Condition 3 requires that the inner product of $\mathbf{x}$ with any unit vector $\mathbf{w}$ has a sub-exponential tail bound. These conditions generalize the feature distribution conditions in a large body of literature on learning halfspaces with noise [e.g., Diakonikolas et al., 2020, 2021, Zhang et al., 2020, Dasgupta, 2005, Yan and Zhang, 2017, Shen, 2021a, Awasthi et al., 2017]. This existing literature typically assumes that the feature $\mathbf{x}$ has an isotropic log-concave distribution, such as a uniform distribution over a unit sphere. In Appendix A.1, we show that Assumption 2 accommodates even non-isotropic log-concave distributions, including many common distributions such as uniform, Gaussian, exponential, logistic distributions, etc. Notably, we impose distributional assumptions on the feature-label pairs, which differ from and complement the fully adversarial setting in the literature [e.g., Dong et al., 2018, Chen et al., 2020, Ahmadi et al., 2024].

**Agent Feature Manipulation** We assume that each agent gains a utility of $+1$ for admission (classified as $+1$) and $-1$ for rejection (classified as $-1$). An agent with true feature $\mathbf{x}$ may report his feature as $\mathbf{r}$ to sway the classifier's decision. Following Shen et al. [2024], Ahmadi et al. [2021], we assume that this misreporting or manipulation incurs a cost $\text{Cost}(\mathbf{x}, \mathbf{r}) = 2\|\mathbf{x} - \mathbf{r}\|_2/\gamma$, where $\gamma > 0$ indicates the maximum manipulation distance. Therefore, upon the principal announcing a classifier $\tilde{h}(\cdot)$, the agent's optimal reported feature that maximizes the net utility would be $\mathbf{r}^\star(\mathbf{x}, \tilde{h}) = \arg\max_{\mathbf{r} \in \mathbb{R}^d} \tilde{h}(\mathbf{r}) - 2\|\mathbf{x} - \mathbf{r}\|_2/\gamma$.

Given the linear model in Assumption 1, we restrict the principal's classifier $\tilde{h}$ to linear classifiers parameterized by $(\mathbf{w}, m) \in \mathbb{S}^d \times \mathbb{R}$, i.e., $\tilde{h}(\mathbf{r}) = sgn(\langle \mathbf{w}, \mathbf{r} \rangle + m)$ . In this case, an agent's optimal reported feature is given in the following lemma [Shen et al., 2024, Ahmadi et al., 2021].

**Lemma 1.** *Given an announced classifier $\tilde{h}(\mathbf{r}) = sgn(\langle \mathbf{w}, \mathbf{r} \rangle + m)$, the optimal reported feature for an agent with true feature $\mathbf{x}$ is*

$$\mathbf{r}^\star(\mathbf{x}, \tilde{h}) = \begin{cases} \mathbf{x} - (\langle \mathbf{w}, \mathbf{x} \rangle + m)\mathbf{w}, & -\gamma \leq \langle \mathbf{w}, \mathbf{x} \rangle + m < 0; \\ \mathbf{x}, & \text{otherwise.} \end{cases}$$

**The Clairvoyant Optimal Classifier** Under the manipulated feature in Lemma 1, the misclassification rate of a classifier $\tilde{h}$ can be measured by $\text{Err}(\tilde{h}) := \mathbb{P}(\tilde{h}(\mathbf{r}^\star(\mathbf{x}, \tilde{h})) \neq y)$. We hope to characterize a clairvoyant optimal classifier achieving the minimal misclassification rate: $\tilde{h}^\star \in \arg\min_{\tilde{h}: \mathbb{R}^d \mapsto \{\pm 1\}} \text{Err}(\tilde{h})$. To this end, we first connect a classifier $\tilde{h}$ under the manipulated feature $\mathbf{r}$ with a hypothetical classifier $h$ under the corresponding true feature $\mathbf{x}$.

**Proposition 1.** *For any $(\mathbf{w}, m) \in \mathbb{S}^d \times \mathbb{R}$, the output of $\tilde{h}(\mathbf{r}) = sgn(\langle \mathbf{w}, \mathbf{r} \rangle + m - \gamma)$ for $\mathbf{r} = \mathbf{r}^\star(\mathbf{x}, \tilde{h})$ is identical to the output of $h(\mathbf{x}) = sgn(\langle \mathbf{w}, \mathbf{x} \rangle + m)$ for any $\mathbf{x} \in \mathbb{R}^d$.*

According to Assumption 1, the optimal classifier in absence of manipulation is $h^*(\mathbf{x}) = \mathrm{sgn}(\langle \mathbf{w}^\star, \mathbf{x} \rangle)$. Following Proposition 1, we can achieve the same classification by a corresponding classifier subject to manipulation, which gives the clairvoyant optimal classifier. This structural knowledge of a clairvoyant optimal classifier will guide our algorithm design in Section 3.

**Corollary 1.** *The classifier $\tilde{h}^\star(\mathbf{r}) = sgn(\langle \mathbf{w}^\star, \mathbf{r} \rangle - \gamma)$ minimizes $\mathrm{Err}(\tilde{h}) = \mathbb{P}(\tilde{h}(\mathbf{r}^\star(\mathbf{x}, \tilde{h})) \neq y)$.*

Notably, the clairvoyant optimal classifier on the manipulated feature $\mathbf{r}$ has a higher threshold to classify an agent into $+1$ than the corresponding optimal classifier $h^\star(\mathbf{x}) = \mathrm{sgn}(\langle \mathbf{w}^\star, \mathbf{x} \rangle)$ on the true feature $\mathbf{x}$. Indeed, the principal would like to raise the bar for positive classification, in order to avoid errors due to unqualified agents (label $-1$) who game the classifier by manipulating their features.

**Principal's Regret** Over the $T$ cycles, the principal learns a sequence of classifiers $\tilde{\mathbf{h}} = (\tilde{h}_1, \ldots, \tilde{h}_T)$, where each $\tilde{h}_t$ only depends on the observed data of reported features and admitted agents' labels prior to cycle $t$. The goal is to achieve a small cumulative misclassification rate over all cycles. This is equivalent to achieving a small total suboptimality gap, or regret, relative to the clairvoyant optimal classifier. Formally, the regret is defined as:

$$\mathrm{Reg}(\tilde{\mathbf{h}}; T) := \sum_{t=1}^{T} \mathrm{Err}(\tilde{h}_t) - T \times \mathrm{Err}(\tilde{h}^\star). \tag{1}$$

This regret corresponds to the "Stackelberg regret" in the strategic classification literature, where the term "Stackelberg" emphasizes that agents consistently choose their best feature manipulation in response to the principal's announced classifiers [Dong et al., 2018, Chen et al., 2020, Ahmadi et al., 2024]. In the next section, we will propose a learning algorithm that effectively tackles the combined challenges of agents' feature manipulations, partial feedback observations, and label noise. We prove that this algorithm achieves a $\sqrt{T}$-regret rate up to poly-logarithmic and constant factors.

## 3 The Algorithm

### 3.1 Overview of our Algorithm

---

**Algorithm 1:** Main-Algorithm

---

**Input:** Maximum manipulation distance $\gamma$, noise level bound $\bar{\eta}$, lengths $\{T_{\mathrm{init}}\} \cup \{T_k\}_{k=0}^{K}$, bandwidths $\{b_k\}_{k=0}^{K}$, stepsizes $\{\alpha_k\}_{k=0}^{K}$, feature dimension $d$

1    $\bar{\mathbf{w}}_0 = \mathrm{Initialization}(T_{\mathrm{init}})$                        `// See Algorithm 2`
2    $\mathbf{w}_1 = \mathrm{Refinement}(\bar{\mathbf{w}}_0, \bar{\eta}, T_0, b_0, \alpha_0, d)$          `// See Algorithm 3`
3    **for** $k \leftarrow 1$ **to** $K$ **do**
4        $\mathbf{w}_{k+1} = \mathrm{Batched\text{-}Enhancement}(\gamma, \mathbf{w}_1, \bar{\eta}, k, T_k, b_k, \alpha_k, d)$    `// See Algorithm 4`

---

Our main Algorithm, outlined in Algorithm 1, comprises three sub-algorithms: an Initialization Algorithm (Algorithm 2), a Refinement Algorithm (Algorithm 3) and a Batched Enhancement Algorithm (Algorithm 4). These algorithms are executed sequentially to generate a sequence of coefficient vectors such that the corresponding classifiers converge to the clairvoyant optimal classifier as specified in Corollary 1. Specifically, we partition the horizon of $T$ cycles (one agent arrives in each cycle) into consecutive batches indexed by $k \in \{\mathrm{init}, 0, 1, 2, \cdots, K\}$. Index "init" and "0" denote the batches executing the Initialization and Refinement Algorithms, respectively, while indices "1" to "$K$" represent the $K$ batches that run the Enhancement Algorithm iteratively. Each batch $k$ takes the result of the previous batch $k - 1$ as input. Cycles in each batch $k$ are further grouped into iterations indexed by $i \in \{1, 2, \cdots, T_k\}$, where each iteration $i$ performs an update for the coefficient vector $\mathbf{w}$. At the end of batch $k$, the algorithms output the (normalized) average vectors of the $T_k$ iterations in the batch. During Refinement, each iteration consists of only one cycle. In contrast, during Initialization and Enhancement, each iteration contains two cycles, denoted by the superscript $j = 1$ or $2$ to differentiate between the first and second cycles within the same iteration. Note that the indices $(k, i, j)$ can be mapped to the corresponding cycle $t$, for convenience, we will use these indices in the remainder of this paper.

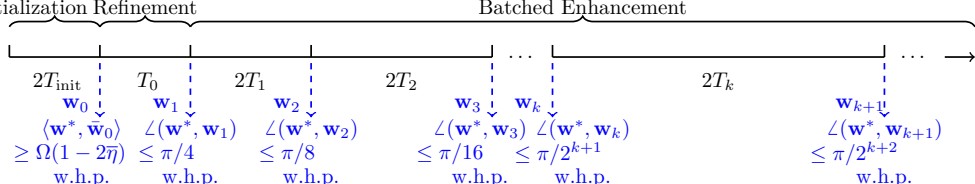

Figure 1: Roles of the three sub-algorithms

The roles of the three sub-algorithms are summarized in Figure 1. First, the *Initialization Algorithm* runs for $T_{\text{init}} = O\left(\ln T/(1 - 2\bar{\eta})^2\right)$ iterations (with $2T_{\text{init}}$ cycles) to find a coefficient vector $\bar{\mathbf{w}}_0$ such that $\theta\left(\mathbf{w}^\star, \bar{\mathbf{w}}_0\right) \leq \frac{\pi}{2}$ with high probability (see Proposition 2). Second, the *Refinement Algorithm* takes $\mathbf{w}_0 = \bar{\mathbf{w}}_0/\|\bar{\mathbf{w}}_0\|_2$ as the initial vector and runs for $T_0 = O\left(d\ln d\ln T/(1 - 2\bar{\eta})^8\right)$ iterations (with $T_0$ cycles) to obtain a refined vector $\mathbf{w}_1$ such that $\theta\left(\mathbf{w}^\star, \mathbf{w}_1\right) \leq \frac{\pi}{4}$ with high probability (see Proposition 3). Third, the *Batched Enhancement Algorithm* runs for $K = O\left(\log_4\left(1 - 2\bar{\eta}\right)^4 T/(\gamma d\ln d\ln T)\right)$ batches, where each batch $k$ enhances its initial coefficient vector $\mathbf{w}_k$ through $T_k = O\left(4^k d\ln d\ln T/(1 - 2\bar{\eta})^4\right)$ iterations (with $2T_k$ cycles), yielding a vector $\mathbf{w}_{k+1}$ such that $\theta\left(\mathbf{w}^\star, \mathbf{w}_{k+1}\right) \leq \frac{\pi}{2^{k+2}}$ with high probability (see Proposition 4). The specification of the algorithms involves absolute constants $c_0$ to $c_7$, which are derived from the parameters in Assumption 2. Detailed calculations are available in Appendix A.5.

## 3.2 Initialization

---
**Algorithm 2:** Initialization
---
**Input:** Iteration length $T_{\text{init}}$
1 **for** $i \leftarrow 1$ **to** $T_{\text{init}}$ **do**
2      Uniformly draw $\mathbf{w}_{\text{init},i} \in \mathbb{S}^d$
3      **for** $j \leftarrow 1$ **to** $2$ **do**
4          Declare $\tilde{h}_{\text{init},i}^{(j)}(\mathbf{r}) = (-1)^{j-1}\text{sgn}(\langle\mathbf{w}_{\text{init},i}, \mathbf{r}\rangle)$, agent $(\mathbf{x}_{\text{init},i}^{(j)}, y_{\text{init},i}^{(j)})$ arrives and reports $\mathbf{r}_{\text{init},i}^{(j)}$
5          Make classification decision $\tilde{h}_{\text{init},i}^{(j)}(\mathbf{r}_{\text{init},i}^{(j)})$ and collect label $y_{\text{init},i}^{(j)}$ if $\tilde{h}_{\text{init},i}^{(j)}(\mathbf{r}_{\text{init},i}^{(j)}) = 1$
6 **return** $\bar{\mathbf{w}}_0 = \frac{1}{T_{\text{init}}}\sum_{i=1}^{T_{\text{init}}}\sum_{j=1}^{2} y_{\text{init},i}^{(j)}\mathbf{r}_{\text{init},i}^{(j)}\mathbb{I}\left((-1)^{(j-1)}\left\langle\mathbf{w}_{\text{init},i}, \mathbf{r}_{\text{init},i}^{(j)}\right\rangle > 0\right)$

---

The initialization algorithm runs for $T_{\text{init}} = O(\ln T/(1 - 2\bar{\eta})^2)$ iterations. In each iteration, we randomly explore a coefficient vector $\mathbf{w}_{\text{init},i} \in \mathbb{S}^d$ and offer two opposing classifiers based on $\mathbf{w}_{\text{init},i}$ to two successive agents. Using the reported features $\mathbf{r}_{\text{init},i}^{(1)}, \mathbf{r}_{\text{init},i}^{(2)}$ and true labels $y_{\text{init},i}^{(1)}, y_{\text{init},i}^{(2)}$ of all positively classified agents over the $T_{\text{init}}$ iterations, we construct an initial coefficient vector $\bar{\mathbf{w}}_0$.

The design of our initialization algorithm stems from the well-known "averaging" technique for learning halfspaces [Servedio, 2001]. In the non-strategic, noiseless and full feedback setting, $y\langle\mathbf{w}^\star, \mathbf{x}\rangle = \langle\mathbf{w}^\star, y\mathbf{x}\rangle \geq 0$ for all $(\mathbf{x}, y) \in \mathbb{R}^d \times \{\pm 1\}$, so $y\mathbf{x}$ forms an acute angle with the optimal normal vector $\mathbf{w}^\star$ almost surely (since the set $\{\mathbf{x} \mid \langle\mathbf{w}^\star, \mathbf{x}\rangle = 0\}$ is zero-measure). Analogously, in the noisy feedback setting, we have $\langle\mathbf{w}^\star, \mathbb{E}[y\mathbf{x}]\rangle > 0$, so $\mathbb{E}[y\mathbf{x}]$ forms an acute angle with $\mathbf{w}^\star$. In a non-strategic and full feedback setting, the literature uses the sample average of $y\mathbf{x}$ to approximate $\mathbb{E}[y\mathbf{x}]$ as an initial estimate of $\mathbf{w}^\star$ [Zhang et al., 2020]. However, this estimator is unavailable for us because of agents' feature manipulation and the partial feedback setting. Instead, our algorithm declares pairs of opposite classifiers. We collect and average the $y\mathbf{r}$ of agents whose reported feature $\mathbf{r}$ falls above the hyperplane. Note that these agents report their features truthfully ($\mathbf{r} = \mathbf{x}$), so we are able to form $\bar{\mathbf{w}}_0$ from these agents' $y\mathbf{r}$ as a proper approximation of $\mathbb{E}[y\mathbf{x}]$. We can show that, for large enough $T_{\text{init}}$, this vector $\bar{\mathbf{w}}_0$ forms an acute angle with the optimal $\mathbf{w}^\star$ with high probability.

**Proposition 2.** *For some constants $c_0, c_1 > 0$, when Algorithm 2 runs for $T_{\text{init}} = c_0 \ln T/(1 - 2\bar{\eta})^2$ iterations, its output $\bar{\mathbf{w}}_0$ satisfies $\langle\mathbf{w}^\star, \bar{\mathbf{w}}_0\rangle > c_1(1 - 2\bar{\eta}) > 0$ and $\theta\left(\mathbf{w}^\star, \bar{\mathbf{w}}_0\right) \leq \frac{\pi}{2}$ with probability at least $1 - 2/T^2$.*

## 3.3 Refinement of the Initial Coefficient Vector

---

**Algorithm 3:** Refinement

---

**Input:** Initial vector $\bar{\mathbf{w}}_0$, noise level $\bar{\eta}$, iteration length $T_0$, bandwidth $b_0$, step size $\alpha_0$, feature dimension $d$

**Initialization:** $\mathbf{w}_{0,1} = \bar{\mathbf{w}}_0 / \|\bar{\mathbf{w}}_0\|_2$

1 **for** $i \leftarrow 1$ **to** $T_0$ **do**

2      Declare classifier $\tilde{h}_{0,i}(\mathbf{r}) = \text{sgn}(\langle \mathbf{w}_{0,i}, \mathbf{r} \rangle)$, agent $(\mathbf{x}_{0,i}, y_{0,i})$ arrives and reports $\mathbf{r}_{0,i}$

3      Make classification decision $\tilde{h}_{0,i}(\mathbf{r}_{0,i})$ and collect label $y_{0,i}$ if $\tilde{h}_{0,i}(\mathbf{r}_{0,i}) = 1$

4      Compute gradient: $\tilde{\mathbf{g}}_{0,i} = [-\bar{\eta}\mathbf{r}_{0,i}\mathbb{I}(y_{0,i} = 1) + (1 - \bar{\eta})\mathbf{r}_{0,i}\mathbb{I}(y_{0,i} = -1)]\mathbb{I}(0 < \langle \mathbf{w}_{0,i}, \mathbf{r}_{0,i} \rangle \leq b_0)$

5      Set constraint set: $\mathcal{W}_0 = \{\mathbf{w} \mid \|\mathbf{w}\|_2 \leq 1, \langle \mathbf{w}, \bar{\mathbf{w}}_0 \rangle \geq c_1(1 - 2\bar{\eta})\}$

6      Update $\mathbf{w}$: $\hat{\mathbf{w}}_{0,i+1} = \arg\min_{\mathbf{w} \in \mathcal{W}_0} \langle \tilde{\mathbf{g}}_{0,i}, \mathbf{w} \rangle + \frac{1}{\alpha_0} \frac{\|\mathbf{w} - \mathbf{w}_{0,i}\|_p^2}{2(p-1)}$, where $p = \frac{\ln(8d)}{\ln(8d) - 1}$

7      Normalize: $\mathbf{w}_{0,i+1} = \hat{\mathbf{w}}_{0,i+1} / \|\hat{\mathbf{w}}_{0,i+1}\|_2$

8 Compute mean vector: $\bar{\mathbf{w}}_1 = \frac{1}{T_0} \sum_{i=1}^{T_0} \mathbf{w}_{0,i}$

9 **return** $\mathbf{w}_1 = \bar{\mathbf{w}}_1 / \|\bar{\mathbf{w}}_1\|_2$

---

The refinement algorithm adopts a "localization" scheme to refine the output $\bar{\mathbf{w}}_0$ of Algorithm 2 to better approximate $\mathbf{w}^\star$. In every iteration $i$, we consider only data within a band $0 < \langle \mathbf{w}_{0,i}, \mathbf{r} \rangle \leq b_0$ adjacent to the boundary of the current classifier $\tilde{h}_{0,i}(\mathbf{r}) = \text{sgn}(\langle \mathbf{w}_{0,i}, \mathbf{r} \rangle)$. Agents in this band are positively classified and have no incentives for feature manipulation, allowing us to observe both the true feature $\mathbf{x} = \mathbf{r}$ and the true label $y$. Moreover, this effectively probes the localized region $D_{0,i} = \{\mathbf{x} : 0 < \langle \mathbf{w}_{0,i}, \mathbf{x} \rangle \leq b_0\}$ in the true feature space. Similar "localization" is widely used in the literature of learning half-spaces with label noises (see references in Section 1.1), since data near the classification boundary is the most informative, while data far from the boundary are either correctly classified with ease or are misclassified mainly due to noises, providing little information.

We formulate an online linear optimization problem with constructed losses $\{\mathbf{w} \mapsto \langle \mathbf{w}, \tilde{\mathbf{g}}_{0,i} \rangle\}_{i=1}^{T_0}$ over a proper constraint set $\mathcal{W}_0 = \{\mathbf{w} \mid \|\mathbf{w}\|_2 \leq 1, \langle \mathbf{w}, \bar{\mathbf{w}}_0 \rangle \geq c_1(1 - 2\bar{\eta})\}$. This constraint set, according to Proposition 2, contains $\mathbf{w}^\star$ with high probability. We then solve this problem by online mirror descent with a stepsize $\alpha_0$ and regularizer $\|\mathbf{w} - \mathbf{w}_{0,i}\|_p^2 / 2(p-1)$ for $p = \ln(8d)/(\ln(8d) - 1)$, perform proper normalization in each iteration, and normalize the average of all iterates to obtain the output $\mathbf{w}_1$. By focusing on the band $\{\mathbf{r} : 0 < \langle \mathbf{w}_{0,i}, \mathbf{r} \rangle \leq b_0\}$, we can perfectly observe $\mathbf{r}_{0,i} = \mathbf{x}_{0,i}$ and $y_{0,i}$ and the gradients[2] $\tilde{\mathbf{g}}_{0,i}$ coincide with the counterparts in Zhang et al. [2020]. As a result, we can follow their analysis to bound the error of the output $\mathbf{w}_1$.

**Proposition 3.** *For the constant $c_1$ in Proposition 2 and some constants $c_2, c_3, c_4 > 0$, when the initial vector $\bar{\mathbf{w}}_0$ satisfies $\langle \mathbf{w}^\star, \bar{\mathbf{w}}_0 \rangle \geq c_1(1 - 2\bar{\eta})$ and Algorithm 3 runs with bandwidth $b_0 = c_2(1 - 2\bar{\eta})^2$ for $T_0 = c_3 d \ln d (\ln T)^2 / (1 - 2\bar{\eta})^8$ iterations with step size $\alpha_0 = c_4 \sqrt{d \ln(d)} / (\sqrt{T_0} \ln T)$, then its output $\mathbf{w}_1$ satisfies $\theta(\mathbf{w}^\star, \mathbf{w}_1) \leq \pi/4$ with probability at least $1 - 3/T^2$.*

The main idea in proving Proposition 3 is outlined as follows. By the theory of online convex optimization, we can upper bound the cumulative regret for the constructed loss in this stage, i.e., $\sum_{i=1}^{T_0} \langle \mathbf{w}_{0,i}, \tilde{\mathbf{g}}_{0,i} \rangle - \langle \mathbf{w}^\star, \tilde{\mathbf{g}}_{0,i} \rangle$. This regret bound, together with a bound on $\sum_{i=1}^{T_0} \langle \mathbf{w}_{0,i}, \tilde{\mathbf{g}}_{0,i} \rangle$ and a concentration bound on $\sum_{i=1}^{T_0} \langle \mathbf{w}^\star, -\tilde{\mathbf{g}}_{0,i} \rangle$, leads to a high probability upper bound on $\sum_{i=1}^{T_0} \mathbb{E}[\langle \mathbf{w}^\star, -\tilde{\mathbf{g}}_{0,i} \rangle]$. Moreover, it can be shown that $\mathbb{E}[\langle \mathbf{w}^\star, -\tilde{\mathbf{g}}_{0,i} \rangle]$ is lower bounded by $\theta(\mathbf{w}^\star, \mathbf{w}_{0,i})$ up to some proportional factors. This is why we expect to obtain a high probability upper bound on $\theta(\mathbf{w}^\star, \mathbf{w}_1)$. Importantly, the gradients $\tilde{\mathbf{g}}_{0,i}$ are carefully constructed to ensure that $|\langle \mathbf{w}_{0,i}, \tilde{\mathbf{g}}_{0,i} \rangle|$ is small and meanwhile $\mathbb{E}[\langle \mathbf{w}^\star, -\tilde{\mathbf{g}}_{0,i} \rangle]$ upper bounds $\theta(\mathbf{w}^\star, \mathbf{w}_{0,i})$.

Notably, while Algorithm 3 collects true feature-label data and implements localization by focusing on local bands around the origin-crossing classification hyperplanes $\tilde{h}_{0,i}$'s, this approach can be costly. According to Lemma 1, unqualified agents with true features $\mathbf{x}$ satisfying $-\gamma \leq \langle \mathbf{w}_{0,i}, \mathbf{x} \rangle \leq 0$ would manipulate their features to achieve positive classifications, resulting in constant instantaneous regret. Fortunately, Algorithm 3 runs for only $\tilde{O}(\ln T)$ cycles, so this refinement algorithm obtains an improved coefficient $\mathbf{w}_1$ for the next stage at the cost of at most only $\tilde{O}(\ln T)$ regret.

---

[2]It can be verified that $\tilde{\mathbf{g}}_{0,i}$ is the gradient of a Leaky ReLu loss restricted to the band $D_{0,i}$.

## 3.4 Batched Enhancement: Proxy Features and Pairwise Contrastive Inference

---

**Algorithm 4:** Batched Enhancement

---

**Input:** Maximum manipulation distance $\gamma$, initial vector $\mathbf{w}_k$, noise level $\bar{\eta}$, batch index $k$, iteration length $T_k$, bandwidth $b_k$, step size $\alpha_k$, feature dimension $d$

**Initialization :** $\mathbf{w}_{k,1} = \mathbf{w}_k$

1 **for** $i \leftarrow 1$ **to** $T_k$ **do**

2      Construct classifiers $\tilde{h}_{k,i}^{(1)}(\mathbf{r}) = \mathrm{sgn}(\langle \mathbf{w}_{k,i}, \mathbf{r}\rangle - \gamma)$ and $\tilde{h}_{k,i}^{(2)}(\mathbf{r}) = \mathrm{sgn}(\langle \mathbf{w}_{k,i}, \mathbf{r}\rangle - \gamma - b_k)$

3      **for** $j \leftarrow 1$ **to** $2$ **do**

4          Declare classifier $\tilde{h}_{k,i}^{(j)}$, agent $(\mathbf{x}_{k,i}^{(j)}, y_{k,i}^{(j)})$ arrives and reports $\mathbf{r}_{k,i}^{(j)}$

5          Make classification decision $\tilde{h}_{k,i}^{(j)}(\mathbf{r}_{k,i})$ and collect label $y_{k,i}^{(j)}$ if $\tilde{h}_{k,i}^{(j)}(\mathbf{r}_{k,i}^{(j)}) = 1$

6          Construct proxy data: $\hat{\mathbf{x}}_{k,i}^{(j,+)} = \mathrm{Proj}_{D_{k,i}}^+ (\mathbf{r}_{k,i}^{(j)}) \mathbb{I}(y_{k,i}^{(j)} = 1, \mathbf{r}_{k,i}^{(j)} \in \tilde{D}_{k,i}^{(j)}), \hat{\mathbf{x}}_{k,i}^{(j,-)} = $
         $\mathrm{Proj}_{D_{k,i}}^- (\mathbf{r}_{k,i}^{(j)}) \mathbb{I}(y_{k,i}^{(j)} = -1, \mathbf{r}_{k,i}^{(j)} \in \tilde{D}_{k,i}^{(j)})$

7      Use the proxy data to compute the gradient: $\hat{\mathbf{g}}_{k,i} = -\bar{\eta}(\hat{\mathbf{x}}_{k,i}^{(1,+)} - \hat{\mathbf{x}}_{k,i}^{(2,+)}) + (1 - \bar{\eta})(\hat{\mathbf{x}}_{k,i}^{(1,-)} - \hat{\mathbf{x}}_{k,i}^{(2,-)})$

8      Update: $\hat{\mathbf{w}}_{k,i+1} \leftarrow \arg\min_{\mathbf{w} \in \mathcal{W}_k} \langle \hat{\mathbf{g}}_{k,i}, \mathbf{w}\rangle + \frac{1}{\alpha_k} \frac{\|\mathbf{w} - \mathbf{w}_{k,i}\|_p^2}{2(p-1)}$, where $p = \frac{\ln(8d)}{\ln(8d)-1}$, the constraint set
     $\mathcal{W}_k = \{\mathbf{w}|\ \|\mathbf{w}\|_2 \leq 1, \langle \mathbf{w}, \mathbf{w}_k\rangle \geq \cos\theta_k\}$, starting angle $\theta_k = \frac{\pi}{2^{k+1}}$

9      Normalize: $\mathbf{w}_{k,i+1} = \hat{\mathbf{w}}_{k,i+1}/\|\hat{\mathbf{w}}_{k,i+1}\|_2$

10 Compute mean vector $\bar{\mathbf{w}}_{k+1} = \frac{1}{T_k}\sum_{i=1}^{T_k} \mathbf{w}_{k.i}$

11 **return** $\mathbf{w}_{k+1} = \bar{\mathbf{w}}_{k+1}/\|\bar{\mathbf{w}}_{k+1}\|_2$

---

In the non-strategic and full feedback setting, after obtaining the refined coefficient $\mathbf{w}_1$, Zhang et al. [2020] further improves it by solving a sequence of adaptively constructed online linear optimization problems $\min_{\mathbf{w} \in \mathcal{W}_k} \sum_{i=1}^{T_k} \langle \mathbf{w}, \mathbf{g}_{k,i}\rangle$ with $\mathbf{g}_{k,i} = [-\bar{\eta}\mathbf{x}_{k,i}\mathbb{I}(y_{k,i} = 1) + (1 - \bar{\eta})\mathbf{x}_{k,i}\mathbb{I}(y_{k,i} = -1)]\mathbb{I}(-b_k < \langle \mathbf{w}_{k,i}, \mathbf{x}_{k,i}\rangle \leq b_k)$ via mirror descent over $k = 1\ldots, K$ batches, using local data within increasingly narrow bands $\{\mathbf{x}\,|-b_k < \langle \mathbf{w}_{k,i}, \mathbf{x}\rangle \leq b_k\}$ around the classification hyperplanes. This process can geometrically reduce the error of the coefficient estimates, outputting a final classifier that approaches the optimal classifier after enough batches. The key ingredient underlying this guarantee is that the gradients $\mathbf{g}_{k,i}$ are well constructed so that $|\langle \mathbf{w}_{k,i}, \mathbf{g}_{k,i}\rangle|$ is small and meanwhile $\mathbb{E}\left[\langle \mathbf{w}^\star, -\mathbf{g}_{k,i}\rangle\right]$ upper bounds $\theta\left(\mathbf{w}^\star, \mathbf{w}_{k,i}\right)$ (see discussions below Proposition 3). One may consider directly implementing this batched enhancement approach in our strategic classification. In particular, one may again use classifiers $\tilde{h}_{k,i}(\mathbf{r}) = \mathrm{sgn}(\langle \mathbf{w}_{k,i}, \mathbf{r}\rangle)$ and focus on the band $D_{k,i} = \{\mathbf{x}\,|\,0 < \langle \mathbf{w}_{k,i}, \mathbf{x}\rangle \leq b_k\}$ in each batch $k$ and iteration $i$, since this enables us to collect the true feature-label data and probe the localized region $D_{k,i}$. However, as we discussed at the end of Section 3.3, this approach may result in constant instantaneous regret in every cycle due to unqualified strategic agents, so that $O(T)$ regret accumulates over the $O(T)$ cycles in this stage.

To avoid excessive errors due to feature manipulation, we can instead employ classifiers $\tilde{h}_{k,i}(\mathbf{r}) = \mathrm{sgn}(\langle \mathbf{w}_{k,i}, \mathbf{r}\rangle - \gamma)$, mimicking the form of the clairvoyant optimal strategic classifier $\tilde{h}^\star$ and raising the bar for positive classification to tackle strategic behaviors (see Corollary 1). However, this gives rise to new challenges: it is unclear how to construct the gradients $\mathbf{g}_{k,i}$ and probe the localized regions $D_{k,i}$, since both depend on the true features, but all agents in the localized regions $D_{k,i}$ misreport their features. This means that we know neither which agents' true feature values belong to the regions $D_{k,i}$ nor their true feature values. To tackle these challenges, we propose two key ideas: proxy features and pairwise contrastive inference.

**Proxy Features**    Even if we assume, for the sake of argument, that we can identify agents whose true features lie in $D_{k,i}$, their true feature values remain unobservable, since they all misreport their features to secure positive classification (so their reported feature values fall on the hyperplane of the announced classifier). To resolve this, we construct proxy features from the reported features.

Specifically, consider an agent with true feature value $\mathbf{x} \in D_{k,i}$ and reported feature value $\mathbf{r}$. This agent will manipulate his feature to get positively classified, and thus we can observe his true label. If his true label is $y = +1$, then we construct his proxy feature $\tilde{\mathbf{x}}$ as the projection of $\mathbf{r}$ onto the *upper* boundary of $D_{k,i}$, i.e., $\tilde{\mathbf{x}} = \mathrm{Proj}_{D_{k,i}}^+(\mathbf{r}) := \mathbf{r} + (b_k - \langle \mathbf{w}_{k,i}, \mathbf{r}\rangle)\mathbf{w}_{k,i}$. On the contrary, if his true label is $y = -1$, then we construct his proxy feature $\tilde{\mathbf{x}}$ as the projection of $\mathbf{r}$ onto the *lower*

boundary of $D_{k,i}$, i.e., $\tilde{\mathbf{x}} = \mathrm{Proj}^-_{D_{k,i}}(\mathbf{r}) := \mathbf{r} - \langle \mathbf{w}_{k,i}, \mathbf{r} \rangle \mathbf{w}_{k,i}$. As a result, this agent's proxy feature value, like his true feature value, also belongs to $D_{k,i}$, and the proxy feature value under a positive label (i.e., projection onto the upper boundary of $D_{k,i}$) is more aligned with the direction of positive classification than the proxy feature value under a negative label (i.e., projection onto the lower boundary of $D_{k,i}$). See the illustration in Figure 2(a).

Using the proxy features, we can approximate the ideal gradient $\mathbf{g}_{k,i}$ by a proxy gradient $\tilde{\mathbf{g}}_{k,i} = [-\bar{\eta}\mathrm{Proj}^+_{D_{k,i}}(\mathbf{r}_{k,i})\mathbb{I}(y_{k,i}=1) + (1-\bar{\eta})\mathrm{Proj}^-_{D_{k,i}}(\mathbf{r}_{k,i})\mathbb{I}(y_{k,i}=-1)]\mathbb{I}(\mathbf{x}_{k,i} \in D_{k,i})$. Although this may not exactly recover the ideal gradient, it is still effective, in that $|\langle \mathbf{w}_{k,i}, \tilde{\mathbf{g}}_{k,i} \rangle|$ is small and $\mathbb{E}[\langle \mathbf{w}^\star, -\tilde{\mathbf{g}}_{k,i} \rangle] \geq \mathbb{E}[\langle \mathbf{w}^\star, -\mathbf{g}_{k,i} \rangle]$ also upper bounds $\theta(\mathbf{w}^\star, \mathbf{w}_{k,i})$ (see Appendix A.5). Therefore, we can use the proxy gradients $\tilde{\mathbf{g}}_{k,i}$ in the algorithm to achieve similar guarantees. Nevertheless, these proxy gradients require knowing whether an agent's true feature value belongs to the localized region $D_{k,i}$ or not, which is still infeasible in our setting. This motivates our second key idea.

**Pairwise Contrastive Inference**    We propose to offer two classifiers $\tilde{h}^{(1)}_{k,i}(\mathbf{r}) = \mathrm{sgn}(\langle \mathbf{w}_{k,i}, \mathbf{r} \rangle - \gamma)$ and $\tilde{h}^{(2)}_{k,i}(\mathbf{r}) = \mathrm{sgn}(\langle \mathbf{w}_{k,i}, \mathbf{r} \rangle - \gamma - b_k)$ successively in each iteration. Under classifier $\tilde{h}^{(1)}_{k,i}(\mathbf{r})$, we consider only agents with reported features in $\tilde{D}^{(1)}_{k,i} = \{\mathbf{r} : \gamma \leq \langle \mathbf{w}_{k,i}, r \rangle \leq \gamma + b_k\}$, while under classifier $\tilde{h}^{(2)}_{k,i}(\mathbf{r})$, we consider only agents with reported features in $\tilde{D}^{(2)}_{k,i} = \{\mathbf{r} : \langle \mathbf{w}_{k,i}, \mathbf{r} \rangle = \gamma + b_k\}$. These agents are all classified into the positive class, so their true labels are observed. Moreover, according to the feature manipulation rule in Lemma 1, these agents have true feature values in $D^{(1)}_{k,i} = \{\mathbf{x} : 0 \leq \langle \mathbf{w}_{k,i}, \mathbf{x} \rangle \leq \gamma + b_k\}$ and $D^{(2)}_{k,i} = \{\mathbf{x} : b_k \leq \langle \mathbf{w}_{k,i}, \mathbf{x} \rangle \leq \gamma + b_k\}$, respectively. Since $D_{k,i} = D^{(1)}_{k,i} \setminus D^{(2)}_{k,i}$ up to a measure-zero set, we can expect to infer distributional properties of the data within the region $D_{k,i}$ of interest by contrasting the data within $D^{(1)}_{k,i}$ and the data within $D^{(2)}_{k,i}$. We call this a *pairwise contrastive inference* approach, which is illustrated in Figure 2(b).

We can use this approach to infer the two key components in the proxy gradient $\mathbf{g}_{k,i}$. Note

$$\mathbb{E}\left[\mathrm{Proj}^+_{D_{k,i}}(\mathbf{r}_{k,i})\mathbb{I}(y_{k,i}=1, \mathbf{x}_{k,i} \in D_{k,i})\right] = \mathbb{E}\left[\hat{\mathbf{x}}^{(1,+)}_{k,i} - \hat{\mathbf{x}}^{(2,+)}_{k,i}\right],$$

where $\hat{\mathbf{x}}^{(j,+)}_{k,i} = \mathrm{Proj}^+_{D_{k,i}}(\mathbf{r}^{(j)}_{k,i})\mathbb{I}\left(y^{(j)}_{k,i}=1, \mathbf{r}^{(j)}_{k,i} \in \tilde{D}^{(j)}_{k,i}\right)$ for $j=1,2$. This means that we can use $\hat{\mathbf{x}}^{(1,+)}_{k,i} - \hat{\mathbf{x}}^{(2,+)}_{k,i}$ to unbiasedly infer one key component of $\tilde{\mathbf{g}}_{k,i}$ in expectation. Similarly, we can construct $\hat{\mathbf{x}}^{(1,-)}_{k,i} - \hat{\mathbf{x}}^{(2,-)}_{k,i}$ to infer the other component. This gives our gradient estimate $\hat{\mathbf{g}}_{k,i}$ in Algorithm 4 Line 7, satisfying that $\mathbb{E}[\langle \mathbf{w}^\star, -\hat{\mathbf{g}}_{k,i} \rangle] = \mathbb{E}[\langle \mathbf{w}^\star, -\tilde{\mathbf{g}}_{k,i} \rangle]$ also upper bounds $\theta(\mathbf{w}^\star, \mathbf{w}_{k,i})$.

After getting the gradient estimator $\hat{\mathbf{g}}_{k,i}$, we again conduct online mirror decent with a regularizer similar to that in Algorithm 3 and constraint set $\mathcal{W}_k = \{\mathbf{w}| \|\mathbf{w}\|_2 \leq 1, \langle \mathbf{w}, \mathbf{w}_k \rangle \geq \cos\theta_k\}$, where $\theta_k = \frac{\pi}{2^{k+1}}$. Then we output the normalized average coefficient vector $\mathbf{w}_{k+1}$ for the next batch. Our constraint set ensures that $\theta(\mathbf{w}_{k,i}, \mathbf{w}_k) \leq \pi/2^{k+1}$ for all $i \in [T_k]$. Then, when the input vector $\mathbf{w}_k$ satisfies $\theta(\mathbf{w}^\star, \mathbf{w}_k) \leq \pi/2^{k+1}$, we have $\theta(\mathbf{w}^\star, \mathbf{w}_{k,i}) \leq \theta(\mathbf{w}^\star, \mathbf{w}_k) + \theta(\mathbf{w}_{k,i}, \mathbf{w}_k) \leq \pi/2^k$ by a triangular inequality shown in Appendix A.3, Lemma 12. This statement is critical: First, it controls the expected cumulative error in batch $k$ to be $O\left(\frac{1}{2^k} \cdot T_k\right)$. Second, the condition that $\theta(\mathbf{w}^\star, \mathbf{w}_{k,i}) \leq \pi/2^k$, together with our localized online mirror descent method, ensures that batch $k$ outputs a vector $\mathbf{w}_{k+1}$ that satisfy $\theta(\mathbf{w}^\star, \mathbf{w}_{k+1}) \leq \pi/2^{k+2}$ with high probability (see Proposition 4), which is in turn required by the next batch.

**Proposition 4.** *For some constants $c_5, c_6, c_7 > 0$, when Algorithm 4 runs with an initial vector $\mathbf{w}_k$ satisfying $\theta(\mathbf{w}^\star, \mathbf{w}_k) \leq \theta_k = \pi/2^{k+1}$, bandwidth $b_k = c_5(1-2\bar{\eta})2^{-k}$ for $T_k = c_6 4^k(\gamma+1)d\ln d(\ln T)^2/(1-2\bar{\eta})^4$ iterations with step size $\alpha_k = c_7\sqrt{d\ln d}\theta_k/(\sqrt{T_k}\ln T)$, its output $\mathbf{w}_{k+1}$ satisfies $\theta(\mathbf{w}^\star, \mathbf{w}_{k+1}) \leq \theta_{k+1} = \frac{\theta_k}{2}$ with probability at least $1 - 6/T^2$.*

Proposition 4 shows that Algorithm 4 enhances its input by reducing the error by half in every batch, generating a sequence of coefficient estimates $(\mathbf{w}_k)^K_{k=1}$ with geometrically decaying errors. Notably, we achieve the enhancement by classifiers $\tilde{h}^{(1)}_{k,i}, \tilde{h}^{(2)}_{k,i}$ that use at least $\gamma$ classification thresholds and are hence more resilient to errors due to strategic classification, which results in only a sublinear regret, as we will show in Theorem 1.

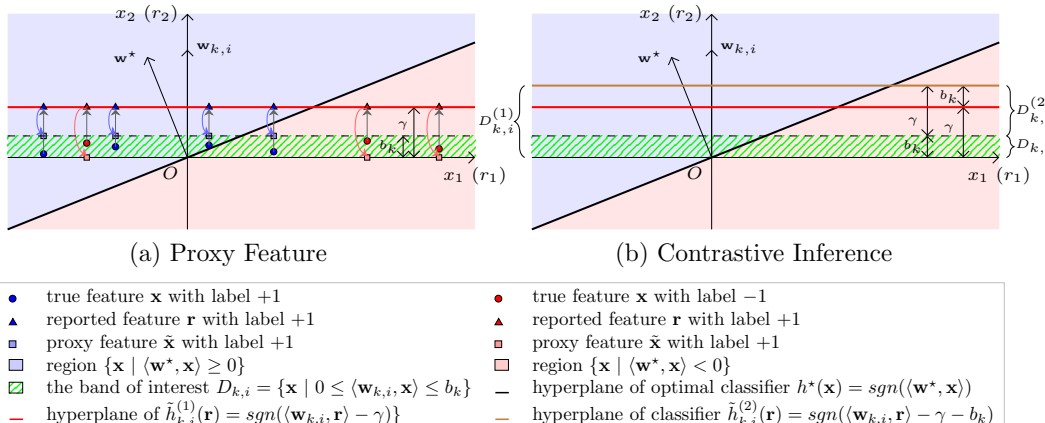

<table>
<tr><td>●</td><td>true feature **x** with label +1</td><td>●</td><td>true feature **x** with label −1</td></tr>
<tr><td>▲</td><td>reported feature **r** with label +1</td><td>▲</td><td>reported feature **r** with label +1</td></tr>
<tr><td>□</td><td>proxy feature $\tilde{\mathbf{x}}$ with label +1</td><td>□</td><td>proxy feature $\tilde{\mathbf{x}}$ with label +1</td></tr>
<tr><td>▦</td><td>region $\{\mathbf{x} \mid \langle \mathbf{w}^\star, \mathbf{x}\rangle \ge 0\}$</td><td>▦</td><td>region $\{\mathbf{x} \mid \langle \mathbf{w}^\star, \mathbf{x}\rangle < 0\}$</td></tr>
<tr><td>▨</td><td>the band of interest $D_{k,i} = \{\mathbf{x} \mid 0 \le \langle \mathbf{w}_{k,i}, \mathbf{x}\rangle \le b_k\}$</td><td>—</td><td>hyperplane of optimal classifier $h^\star(\mathbf{x}) = sgn(\langle \mathbf{w}^\star, \mathbf{x}\rangle)$</td></tr>
<tr><td>—</td><td>hyperplane of $\tilde{h}_{k,i}^{(1)}(\mathbf{r}) = sgn(\langle \mathbf{w}_{k,i}, \mathbf{r}\rangle - \gamma)\}$</td><td>—</td><td>hyperplane of classifier $\tilde{h}_{k,i}^{(2)}(\mathbf{r}) = sgn(\langle \mathbf{w}_{k,i}, \mathbf{r}\rangle - \gamma - b_k)$</td></tr>
</table>

Figure 2: (a) The gray arrows indicate how agents with true feature values (circles) within the band $D_{k,i}$ manipulate their features (triangles). The blue and red arrows indicate how we construct proxy features (squares) from the reported features of agents with labels +1 and −1, respectively. (b) By declaring classifiers $\tilde{h}_{k,i}^{(1)}$ and $\tilde{h}_{k,i}^{(2)}$, we collect data from agents with true values in $D_{k,i}^{(1)}$ and $D_{k,i}^{(2)}$ respectively, through which we infer the information for agents in the region $D_{k,i}$ of interest.

## 4 Regret Guarantee

We now provide a formal regret guarantee of Algorithm 1, showing that it achieves a sublinear regret dependent on the noise level $\bar{\eta}$ and feature dimension $d$.

**Theorem 1.** *For any instance of our online strategic classification problem with noise level $\bar{\eta}$, maximum manipulation distance $\gamma$, and feature dimension $d$, the expected regret of classifiers $\tilde{\mathbf{h}}$ from Algorithm 1 over $T$ cycles satisfies*

$$\mathbb{E}[\text{Reg}(\tilde{\mathbf{h}}; T)] = O\left(d\ln d \times (\ln T)^2/(1 - 2\bar{\eta})^8 + \sqrt{(\gamma + 1)d\ln d \times T}\ln T/(1 - 2\bar{\eta})^2\right).$$

We prove the theorem by analyzing the regret incurred by each of the three sub-algorithms in Section 3. The full proof is outlined in Appendix A.6. We also conduct numerical experiments to evaluate our proposed algorithm, with results presented in Appendix A.2.

## 5 Concluding Remarks

In this paper, we study an online strategic classification problem under Massart Noise with partial feedback. The settings are of practical relevance yet theoretically challenging. We introduce a novel algorithm that concurrently learns a linear classifier and manages instantaneous prediction errors. The algorithm leverages localization to mitigate the complexities induced by Massart noise. The strategic manipulation of agents poses a critical challenge by limiting access to reliable training data; thus, the core innovation of our approach lies in using carefully designed classifier pairs to collect some proxy data and contrasting their data for effective learning. This pairwise contrastive inference approach with proxy data effectively addresses the challenges in online strategic classification. This paper has some limitations. First, our algorithm is specifically designed for Massart Noise. Second, this paper assumes that agents' utility functions are homogeneous and known to the principal. Third, we adopt as an objective the traditional classification accuracy metric. Future research directions include extending the algorithm to overcome these limitations.

## Acknowledgement

The authors thank the anonymous review team for their insightful comments and suggestions. Yong Liang acknowledges the support from the National Natural Science Foundation of China (72325001). Xiaojie Mao acknowledges the support from the National Natural Science Foundation of China (72322001 and 72201150) and the National Key R&D Program of China (2022ZD0116700).

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

# A Technical Appendices and Supplementary Material

## A.1 Feature Regularity Conditions and General Log-concave Distributions

The literature on learning halfspaces with noise typically assumes that the feature vector $\mathbf{x}$ has an isotropic log-concave distribution [Balcan and Long, 2013, Awasthi et al., 2017, Zhang et al., 2020, Shen, 2023]. The log-concave and isotropic log-concave distributions are defined as follows.

**Definition 1** (Isotropic log-concave distribution [Lovász and Vempala, 2007]). *A random vector $\mathbf{z}$ over $\mathbb{R}^d$ with probability density function $\phi_{\mathbf{z}}(\cdot)$ follows a log-concave distribution if $\ln \phi_{\mathbf{z}}(\cdot)$ is concave. Moreover, it is isotropic if $\mathbb{E}[\mathbf{z}] = \mathbf{0}$ and $\mathbb{E}[\mathbf{z}\mathbf{z}^T] = I$.*

The following lemma summarizes some important properties of (isotropic) log-concave distributions that have been proved by literature.

**Lemma 2.** *Suppose $\mathbf{z} \in \mathbb{R}^d$ with probability density function $\phi_{\mathbf{z}}(\cdot)$ follows a log-concave distribution. Then, the following holds.*

   (a) *(Klivans et al. [2009] Lemma 5.17) For $d = 1$, assume that $\mathbb{E}[\mathbf{z}^2] = C^2$, then for every $t > 0$, $\mathbb{P}(|\mathbf{z}| > t) \leq e^{-Ct+1}$.*

*Moreover, if $\mathbf{z}$ is isotropic, then,*

   (b) *(Lovász and Vempala [2007] Lemma 5.2) $\phi_z(\mathbf{z}) \geq \beta_1(d)$ for all $0 \leq \|\mathbf{z}\|_2 \leq 1/9$, where $\beta_1(d) = 2^{-8d}$.*

   (c) *(Lovász and Vempala [2007] Lemma 5.5) For $d = 1$, $\phi_z(\mathbf{z}) \leq 1$.*

   (d) *(Klivans et al. [2009] Lemma 7 ) For $d \geq 2$, $\phi_z(\mathbf{z}) \leq \beta_2(d)e^{-\beta_3(d)\|\mathbf{z}\|_2}$, where $\beta_2(d) = 2^{8d}d^{d/2}e$ and $\beta_3(d) = \frac{2^{-7d}}{2(d-1)(20(d-1))^{(d-1)/2}}$.*

Following Lemma 2, one can show that any mean-zero isotropic log-concave distribution satisfies the regularity conditions in assumption 2. Importantly, in this part, we show that the regularity conditions can hold even for a mean-zero log-concave distribution that is not isotropic. In this case, eigenvalue bounds on the covariance matrix of the distribution determine the corresponding regularity parameters.

**Lemma 3.** *Let $\mathbf{x} \in \mathbb{R}^d$ ($d \geq 2$) have zero mean and a log-concave distribution. Suppose the eigenvalues of its covariance matrix $\Sigma = \mathbb{E}[\mathbf{x}\mathbf{x}^\top]$ are all bounded within $[\underline{\lambda}, \overline{\lambda}]$ for some positive constants $\underline{\lambda}, \overline{\lambda}$, then the distribution of $\mathbf{x}$ satisfies the regularity conditions in assumption 2, with parameters $L_1 = \frac{\beta_1(1)}{\sqrt{\overline{\lambda}}}$, $L_2 = \frac{\beta_1(2)}{\overline{\lambda}}$, $R = \frac{1}{9}\sqrt{\underline{\lambda}}$, $U_1 = \frac{1}{\sqrt{\underline{\lambda}}}$, $U_2 = \frac{\beta_2(2)}{\underline{\lambda}}$, $\delta = \frac{\beta_3(2)}{\sqrt{\overline{\lambda}}}$, $Q = \sqrt{\overline{\lambda}}$ for $\beta_1(1), \beta_1(2), \beta_2(2), \beta_3(2)$ given in Lemma 2.*

## A.2 Numerical Experiments

In this subsection, we conduct numerical experiments to evaluate our proposed algorithm. To highlight the challenges posed by Massart Noise and strategic behavior, and to demonstrate the effectiveness of our algorithm, we compare its regret against two benchmarks: (1) the Strategic Perceptron algorithm from Ahmadi et al. [2021], designed for noiseless online strategic classification, and (2) the PAC learning algorithm for halfspaces with Massart Noise from Zhang et al. [2020], designed for non-strategic classification. Note that these benchmarks are both originally designed for full feedback settings, whereas our work focuses on partial feedback. We evaluate the performance of these benchmark algorithms both when they have access to full feedback (while our algorithm does not) and when they only use partial feedback as our algorithm.

We test the algorithms under two different settings, with key parameters outlined in Table 1. Each setting is replicated 30 times, and we report the average regret for each algorithm. Our analysis includes a performance comparison of the different algorithms and an investigation of how various problem parameters influence our proposed algorithm.

**Benchmark against Strategic Perceptron by Ahmadi et al. [2021]** To understand the impact of Massart Noise, we compare our algorithm with the Strategic Perceptron algorithm from Ahmadi

Table 1: Numerical experiment settings.

| Index | $\mathcal{D}_{\mathbf{x}}$ | $\bar{\eta}$ | $\eta(\mathbf{x})$ | $\gamma$ | $\mathbf{w}^{\star}$ |
|-------|------|------|------|------|------|
| Setting 1 | Standard Normal | 0.1 | $\eta(\mathbf{x}) = \bar{\eta}(1 - \exp(-\|\mathbf{x}\|_2))$ | 0.1 | (1,0) |
| Setting 2 | Unit Ball | 0.1 | $\eta(\mathbf{x}) = \bar{\eta}$ | 0.1 | (1,0) |

et al. [2021]. This algorithm provably achieves only a finite number of mistakes under a noiseless model where the feature-label relationship is deterministic and the true and negative classes are strictly separated by a positive margin. Ahmadi et al. [2021] modify the classical Perceptron algorithm by setting a higher threshold for classifying an agent as positive and proxy surrogate features to estimate the agents' true features. Their proxy feature is defined as follows.

**Definition 2** ($\tilde{\mathbf{x}}_t$, proxy feature , Ahmadi et al. [2021]). *For a given classifier $\tilde{h}(\cdot) = sgn(\langle \mathbf{w}, \cdot \rangle + m)$, an agent $(\mathbf{x}, y)$ reports his feature as $\mathbf{r}$ according to Lemma 1. Then the corresponding proxy feature $\tilde{\mathbf{x}}_t$ in Ahmadi et al. [2021] is defined as*

$$\tilde{\mathbf{x}}_t = \begin{cases} \mathbf{r}_t - \gamma\mathbf{w} & \langle \mathbf{w}, \mathbf{r}_t \rangle = \gamma \text{ and } y_t = -1; \\ \mathbf{r}_t & otherwise. \end{cases} \tag{2}$$

---

**Algorithm 5:** Original Strategic Perceptron with Full Feedback (Ahmadi et al. [2021])

---
1  Accept the first agent without declaring any classifier
2  **if** $y_1 = 1$ **then**
3      $\hat{\mathbf{w}}_2 \leftarrow \mathbf{r}_1$
4  **else**
5      $\hat{\mathbf{w}}_2 \leftarrow -\mathbf{r}_1$
6  $\mathbf{w}_2 \leftarrow \hat{\mathbf{w}}_2 / \|\hat{\mathbf{w}}_2\|_2$
7  **for** $t = 2 \cdots, T$ **do**
8      Declare classifier $\tilde{h}_t(\mathbf{r}) = \text{sgn}(\langle \mathbf{w}_t, \mathbf{r} \rangle - \gamma)$, receive agent response $\mathbf{r}_t$
9      Classify the agent as $\hat{y}_t = \tilde{h}_t(\mathbf{r}_t)$
10     **if** $y_t \neq \hat{y}_t$ **then**
11         $\hat{\mathbf{w}}_{t+1} \leftarrow \mathbf{w}_t + y_t\tilde{\mathbf{x}}_t, \mathbf{w}_{t+1} \leftarrow \hat{\mathbf{w}}_{t+1} / \|\hat{\mathbf{w}}_{t+1}\|_2$
12     **else**
13         $\mathbf{w}_{t+1} \leftarrow \mathbf{w}_t$

---

Their original algorithm, designed for the full feedback setting, is presented in Algorithm 5. Algorithm 6 below directly adapts this algorithm to our partial feedback setting. Specifically, instead of using all data points that incur misclassifications for update, the refined algorithm uses only positively classified agents with true labels $-1$ to adjust the coefficient vector.

---

**Algorithm 6:** Strategic Perceptron with Partial Feedback

---
1  Accept the first agent without declaring any classifier
2  **if** $y_1 = 1$ **then**
3      $\hat{\mathbf{w}}_2 \leftarrow \mathbf{r}_1$
4  **else**
5      $\hat{\mathbf{w}}_2 \leftarrow -\mathbf{r}_1$
6  $\mathbf{w}_2 \leftarrow \hat{\mathbf{w}}_2 / \|\hat{\mathbf{w}}_2\|_2$
7  **for** $t = 2 \cdots, T$ **do**
8      Declare classifier $\tilde{h}_t(\mathbf{r}) = \text{sgn}(\langle \mathbf{w}_t, \mathbf{r} \rangle - \gamma)$, receive agent response $\mathbf{r}_t$
9      **if** $\langle \mathbf{w}, \mathbf{r}_t \rangle \geq \gamma$ **then**
10         Accept the agent and receive his true label $y_t$
11     **else**
12         Reject the agent without getting his true label
13     **if** $y_t = -1$ **then**
14         $\hat{\mathbf{w}}_{t+1} \leftarrow \mathbf{w}_t - \tilde{\mathbf{x}}_t, \mathbf{w}_{t+1} \leftarrow \hat{\mathbf{w}}_{t+1} / \|\hat{\mathbf{w}}_{t+1}\|_2$
15     **else**
16         $\mathbf{w}_{t+1} \leftarrow \mathbf{w}_t$

---

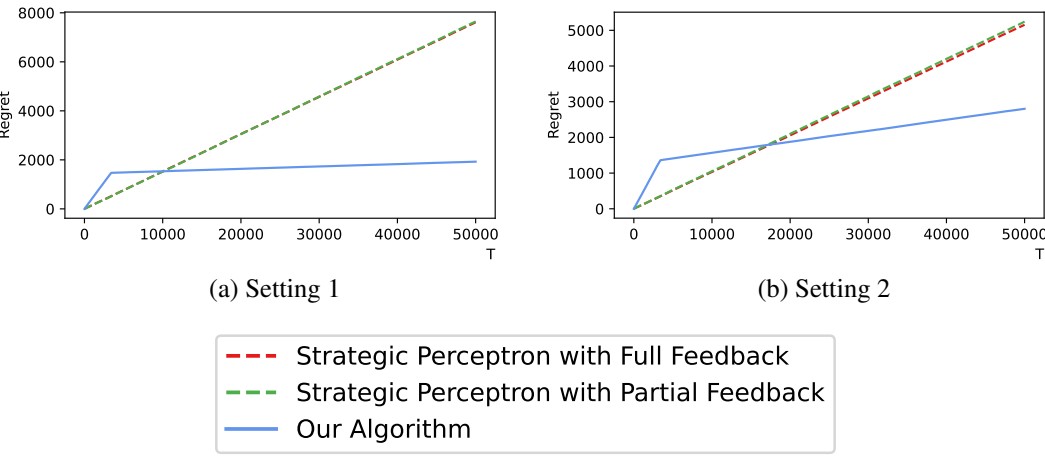

(a) Setting 1              (b) Setting 2

- - - Strategic Perceptron with Full Feedback
- - - Strategic Perceptron with Partial Feedback
—— Our Algorithm

Figure 3: Average regrets of our algorithm and two Strategic Perceptron-based benchmark algorithms over different time horizons $T$ under the two settings listed in Table 1. Results are based on 30 independent replications of the experiment.

Figure 3 illustrates the average regret of each algorithm over 50,000 cycles in both settings listed in Table 1. We observe that our algorithm's regret grows sublinearly, while the two benchmarks' regrets may accumulate linearly. The original Strategic Perceptron with full feedback (the dashed red line) demonstrates slightly better performance compared to its partially feedback-modified counterpart (the dashed green line). However, the improvement remains marginal, suggesting that our partial feedback setting is not the primary cause of the Strategic Perceptron's failure. Intuitively, the ineffectiveness of the two benchmarks stems from their sensitivity to noise when updating with data from all admitted agents. When $\theta(\mathbf{w}^\star, \mathbf{w}_t)$ is relatively small but $\langle \mathbf{w}_t, \mathbf{r}_t \rangle$ (equals $\langle \mathbf{w}_t, \mathbf{x}_t \rangle$ when $\langle \mathbf{w}_t, \mathbf{r}_t \rangle > \gamma$) is large, the algorithm is more likely to admit a 'wrong' agent ($y_t = -1$ but $\tilde{h}(\mathbf{r}_t) = 1$) due to noise rather than an inaccurate classifier. Since the perceptron algorithm updates are based on mistakes (Rosenblatt [1958]), the presence of noise increases the probability of misleading updates for the classifiers. In contrast, our algorithm explores a small band near the decision boundary, whose bandwidth decreases proportionally to $\theta(\mathbf{w}^\star, \mathbf{w}_k)$, $k = 1, 2, \cdots, K$ across batches. Within this band, wrong admissions are more likely due to suboptimal classifiers than noise, making the update more effective. This enables our algorithm to gradually converge to the optimal decision.

**Benchmark against Non-Strategic Learning under Massart Noise by Zhang et al. [2020]** Next, to highlight the impact of agents' strategic behavior, we compare our algorithm against the algorithm proposed by Zhang et al. [2020] (see Algorithm 7), which is designed for adaptively learning halfspaces with Massart Noise in the *non-strategic* classification setting. Their algorithm also adopts a localization scheme that focuses on data within an increasingly narrow band near the classification boundary and uses online mirror descent in batches for classifier updates. However, they do not consider the impact of agents' strategic behavior. We test the performance of their algorithm under strategic manipulation in both full feedback and partial feedback settings. The two settings differ in: 1) whether the principal can collect labels of those who are negatively classified (full feedback) or not (partial feedback) and 2) the algorithm chooses different bandwidths for updates, namely, $\{\mathbf{r} \,|\, -b_k \leq \langle \mathbf{w}_{k,i}, \mathbf{r}_{k,i} \rangle \leq b_k\}$ for the full feedback setting and $\{\mathbf{r} \,|\, 0 < \langle \mathbf{w}_{k,i}, \mathbf{r}_{k,i} \rangle \leq b_k\}$ for the partial feedback setting.

As shown in Figure 4, after a common pure exploration phase, the regret of the non-strategic learning algorithm in both full feedback and partial feedback settings grows linearly. This is because the non-strategic learning algorithm ignores agents' strategic manipulation. Consequently, true negative agents may misreport their features to be positively labeled. In contrast, our algorithm accounts for agents' strategic behavior and is able to efficiently learn the ground truth distribution.

---

**Algorithm 7:** Non-Strategic Learning under Massart Noise

---

**Input:** Feedback setting $F$, noise level bound $\bar{\eta}$, lengths $\{T_{\text{init}}\} \cup \{T_k\}_{k=0}^K$, bandwidths $\{b_k\}_{k=0}^K$, step
      sizes $\{\alpha_k\}_{k=0}^K$, feature dimension $d$

1  $\bar{\mathbf{w}}_0 = \text{Non-Strategic-Initialization}(F, T_{\text{init}})$                       `// See Algorithm 8`

2  $\mathbf{w}_1 = \text{Non-Strategic-Refinement}(F, \bar{\mathbf{w}}_0, \bar{\eta}, T_0, b_0, \alpha_0, d)$             `// See Algorithm 9`

3  **for** $k \leftarrow 1$ **to** $K$ **do**

4      $\mathbf{w}_{k+1} = \text{Non-Strategic-Batched-Enhancement}(F, \mathbf{w}_1, \bar{\eta}, k, T_k, b_k, \alpha_k, d)$   `// See Algorithm 10`

---

---

**Algorithm 8:** Non-Strategic-Initialization

---

**Input:** Feedback setting $F$, iteration length $T_{\text{init}}$

1  **for** $i \leftarrow 1$ **to** $T_{\text{init}}$ **do**

2      Uniformly draw $\mathbf{w}_{\text{init},i} \in \mathbb{S}^d$

3      Declare $\tilde{h}_{\text{init},i}(\mathbf{r}) = \text{sgn}(\langle \mathbf{w}_{\text{init},i}, \mathbf{r} \rangle)$, agent $(\mathbf{x}_{\text{init},i}, y_{\text{init},i})$ arrives and reports $\mathbf{r}_{\text{init},i}$

4      Make classification decision $\tilde{h}_{\text{init},i}(\mathbf{r}_{\text{init},i})$

5      **if** $F = \text{"full"}$ **then**

6         collect label $y_{\text{init},i}$

7      **if** $F = \text{"partial"}$ **then**

8         collect label $y_{\text{init},i}$ only if $\tilde{h}_{\text{init},i}(\mathbf{r}_{\text{init},i}) = 1$

9  **if** $F = \text{"full"}$ **then**

10     **return** $\bar{\mathbf{w}}_0 = \frac{1}{T_{\text{init}}} \sum_{i=1}^{T_{\text{init}}} \mathbf{r}_{init,i} y_{init,i}$

11  **if** $F = \text{"partial"}$ **then**

12     **return** $\bar{\mathbf{w}}_0 = \frac{1}{T_{\text{init}}} \sum_{i=1}^{T_{\text{init}}} \mathbf{r}_{init,i} y_{init,i} \mathbb{I}(\langle \mathbf{w}_{init,i}, \mathbf{r}_{init,i} \rangle > 0)$

---

---

**Algorithm 9:** Non-Strategic-Refinement

---

**Input:** Feedback setting $F$, Initial vector $\bar{\mathbf{w}}_0$, noise level $\bar{\eta}$, iteration length $T_0$, bandwidth $b_0$, step size
      $\alpha_0$, feature dimension $d$

**Initialization :** $\mathbf{w}_{0,1} = \bar{\mathbf{w}}_0 / \|\bar{\mathbf{w}}_0\|_2$

1  **for** $i \leftarrow 1$ **to** $T_0$ **do**

2      Declare classifier $\hat{y}_{0,i} = \tilde{h}_{0,i}(\mathbf{r}) = \text{sgn}(\langle \mathbf{w}_{0,i}, \mathbf{r} \rangle)$, agent $(\mathbf{x}_{0,i}, y_{0,i})$ arrives and reports $\mathbf{r}_{0,i}$

3      **if** $F = \text{"full"}$ **then**

4         Make classification decision $\hat{y}_{0,1} = \tilde{h}_{0,i}(\mathbf{r}_{0,i})$ and collect label $y_{0,i}$

5         Compute gradient:
         $\tilde{\mathbf{g}}_{0,i} = [-\bar{\eta}\mathbf{r}_{0,i}\mathbb{I}(y_{0,i} = \hat{y}_{0,1}) + (1 - \bar{\eta})\mathbf{r}_{0,i}\mathbb{I}(y_{0,i} \neq \hat{y}_{0,1})]\mathbb{I}(-b_0 \leq \langle \mathbf{w}_{0,i}, \mathbf{r}_{0,i} \rangle \leq b_0)$

6      **if** $F = \text{"partial"}$ **then**

7         Make classification decision $\hat{y}_{0,1} = \tilde{h}_{0,i}(\mathbf{r}_{0,i})$ and collect label $y_{0,i}$ only if $\hat{y}_{0,i} = 1$

8         Compute gradient:
         $\tilde{\mathbf{g}}_{0,i} = [-\bar{\eta}\mathbf{r}_{0,i}\mathbb{I}(y_{0,i} = \hat{y}_{0,1}) + (1 - \bar{\eta})\mathbf{r}_{0,i}\mathbb{I}(y_{0,i} \neq \hat{y}_{0,1})]\mathbb{I}(0 < \langle \mathbf{w}_{0,i}, \mathbf{r}_{0,i} \rangle \leq b_0)$

9      Set constraint set: $\mathcal{W}_0 = \{\mathbf{w} \mid \|\mathbf{w}\|_2 \leq 1, \langle \mathbf{w}, \bar{\mathbf{w}}_0 \rangle \geq c_1(1 - 2\bar{\eta})\}$

10     Update $\mathbf{w}$: $\hat{\mathbf{w}}_{0,i+1} = \arg\min_{\mathbf{w} \in \mathcal{W}_0} \langle \tilde{\mathbf{g}}_{0,i}, \mathbf{w} \rangle + \frac{1}{\alpha_0}\frac{\|\mathbf{w} - \mathbf{w}_{0,i}\|_p^2}{2(p-1)}$, where $p = \frac{\ln(8d)}{\ln(8d)-1}$

11     Normalize: $\mathbf{w}_{0,i+1} = \hat{\mathbf{w}}_{0,i+1} / \|\hat{\mathbf{w}}_{0,i+1}\|_2$

12  Compute mean vector: $\bar{\mathbf{w}}_1 = \frac{1}{T_0} \sum_{i=1}^{T_0} \mathbf{w}_{0,i}$

13  **return** $\mathbf{w}_1 = \bar{\mathbf{w}}_1 / \|\bar{\mathbf{w}}_1\|_2$

---

**Impact of Different Parameters** We examine three groups of additional settings to analyze the impact of different parameters. For each group, we test both settings from Table 1. The average regret over 30 independent experiments for each group is depicted in fig. 5 up to 50,000 cycles.

We first examine how different maximum manipulation distances $\gamma = 0.1, 0.2, 0.5$ affect the regret of our algorithm. As depicted in Figure 5 (a1) and (a2), larger $\gamma$ values result in higher regret. Intuitively, a larger $\gamma$ permits more agents to manipulate their features, so the strategic manipulation problem becomes more severe. This causes all algorithms to have worse performance.

Next, we vary the feature space dimension, setting $\mathbf{w}^\star$ to be $(1, 0)$ $(d = 2)$, $(1, 0, 0, 0)$ $(d = 4)$, and $(1, 0, 0, 0, 0, 0)$ $(d = 6)$, respectively. The average regret across different time horizons is shown in

**Algorithm 10:** Non-Strategic-Batched Enhancement

---

**Input:** Feedback setting $F$, initial vector $\mathbf{w}_k$, noise level $\bar{\eta}$, batch index $k$, iteration length $T_k$, bandwidth $b_k$, step size $\alpha_k$, feature dimension $d$

**Initialization :** $\mathbf{w}_{k,1} = \mathbf{w}_k$

1 **for** $i \leftarrow 1$ **to** $T_k$ **do**
2      Declare classifier $\tilde{h}_{k,i}(\mathbf{r}) = \mathrm{sgn}(\langle \mathbf{w}_{k,i}, \mathbf{r} \rangle)$, agent $(\mathbf{x}_{k,i}, y_{k,i})$ arrives and reports $\mathbf{r}_{k,i}$
3      **if** $F = $ *"full"* **then**
4          Make classification decision $\hat{y}_{k,i} = \tilde{h}_{k,i}(\mathbf{r}_{k,i})$ and collect label $y_{k,i}$
5          Compute gradient:
         $\tilde{\mathbf{g}}_{k,i} = [-\bar{\eta}\mathbf{r}_{k,i}\mathbb{I}(y_{0,i} = \hat{y}_{k,i}) + (1-\bar{\eta})\mathbf{r}_{0,i}\mathbb{I}(y_{0,i} \neq \hat{y}_{k,i})]\mathbb{I}(-b_0 \leq \langle \mathbf{w}_{0,i}, \mathbf{r}_{0,i} \rangle \leq b_0)$
6      **if** $F = $ *"partial"* **then**
7          Make classification decision $\hat{y}_{k,i} = \tilde{h}_{k,i}(\mathbf{r}_{k,i})$ and collect label $y_{k,i}$ only if $\hat{y}_{k,i} = 1$
8          Compute gradient:
         $\tilde{\mathbf{g}}_{k,i} = [-\bar{\eta}\mathbf{r}_{k,i}\mathbb{I}(y_{0,i} = \hat{y}_{k,i}) + (1-\bar{\eta})\mathbf{r}_{0,i}\mathbb{I}(y_{0,i} \neq \hat{y}_{k,i})]\mathbb{I}(0 < \langle \mathbf{w}_{0,i}, \mathbf{r}_{0,i} \rangle \leq b_0)$
9      Update: $\hat{\mathbf{w}}_{k,i+1} \leftarrow \arg\min_{\mathbf{w} \in \mathcal{W}_k} \langle \hat{\mathbf{g}}_{k,i}, \mathbf{w} \rangle + \frac{1}{\alpha_k} \frac{\|\mathbf{w} - \mathbf{w}_{k,i}\|_p^2}{2(p-1)}$, where $p = \frac{\ln(8d)}{\ln(8d)-1}$, the constraint set
     $\mathcal{W}_k = \{\mathbf{w}| \|\mathbf{w}\|_2 \leq 1, \|\mathbf{w} - \mathbf{w}_k\|_2 \leq \theta_k$, starting angle $\theta_k = \frac{\pi}{2^{k+1}}$
10      Normalize: $\mathbf{w}_{k,i+1} = \hat{\mathbf{w}}_{k,i+1}/\|\hat{\mathbf{w}}_{k,i+1}\|_2$
11 Compute mean vector $\hat{\mathbf{w}}_{k+1} = \frac{1}{T_k} \sum_{i=1}^{T_k} \mathbf{w}_{k.i}$
12 **return** $\mathbf{w}_{k+1} = \hat{\mathbf{w}}_{k+1}/\|\hat{\mathbf{w}}_{k+1}\|_2$

---

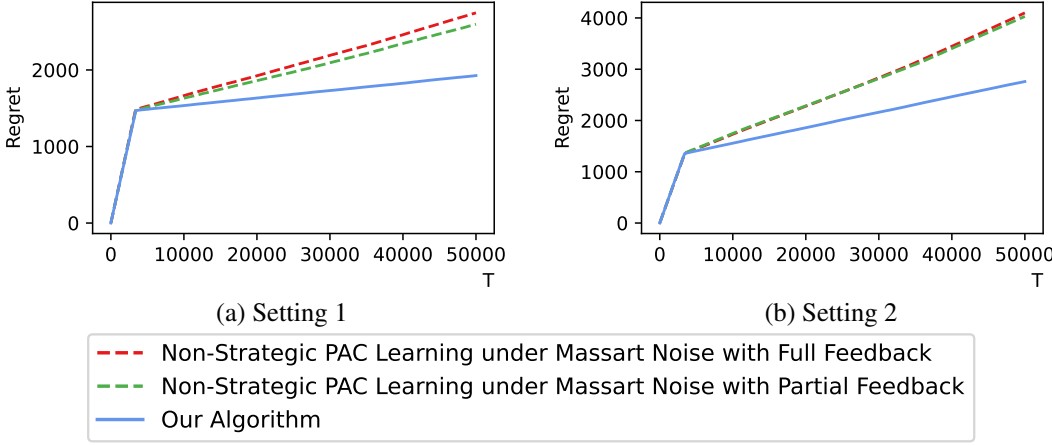

(a) Setting 1            (b) Setting 2

- - - Non-Strategic PAC Learning under Massart Noise with Full Feedback
- - - Non-Strategic PAC Learning under Massart Noise with Partial Feedback
—— Our Algorithm

Figure 4: Average regrets of our algorithm and two Non-strategic learning based-benchmark algorithms over different time horizons $T$ under the two settings listed in Table 1. Results are based on 30 independent replications of the experiment.

fig. 5 (b1) and (b2). As expected, the $d = 2$ setting yields the lowest regret, while $d = 6$ setting yields the highest, consistent with our regret bound.

We finally investigate the impact of the noise level $\bar{\eta}$ on our algorithm's convergence in fig. 5 (c1) and (c2), setting $\bar{\eta}$ to $0.1$, $0.2$ and $0.4$. Surprisingly, the impact of the noise level manifests in opposite trends across the two settings. In setting 1, a higher noise level results in greater regret when $T$ is large enough. Conversely, in setting 2, increased noise levels lead to a lower regret rate. This discrepancy might stem from the fact that, as the noise level rises, the learning accuracy of the clairvoyant optimal classifier diminishes. Given that regret is defined as the difference in cumulative error between our algorithm's classifiers and the clairvoyant optimal ones, the noisier environment could potentially narrow this gap.

### A.3 Technical Lemmas

In this subsection, we list some technical lemmas as instruments for our further proofs.

**Properties of Regular Distributions**

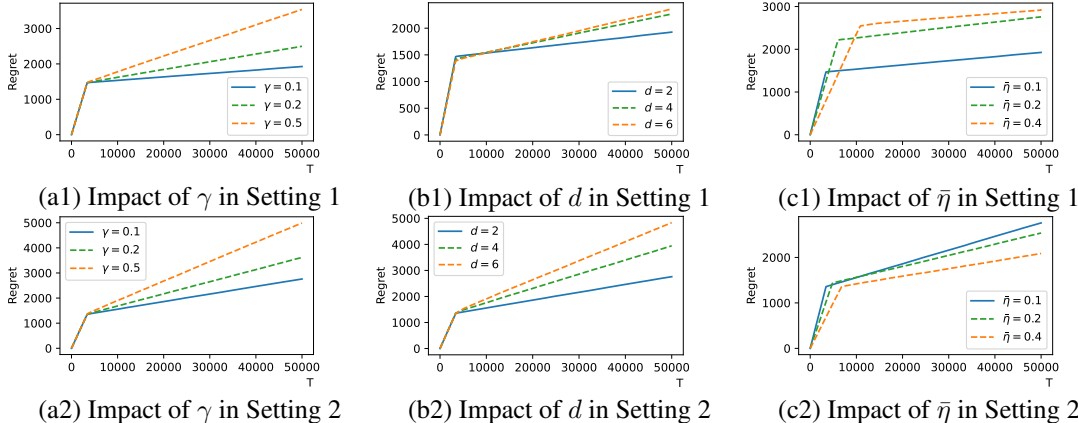

| (a1) Impact of $\gamma$ in Setting 1 | (b1) Impact of $d$ in Setting 1 | (c1) Impact of $\bar{\eta}$ in Setting 1 |
| (a2) Impact of $\gamma$ in Setting 2 | (b2) Impact of $d$ in Setting 2 | (c2) Impact of $\bar{\eta}$ in Setting 2 |

Figure 5: Average regrets of our algorithm over different time horizons $T$ on various parameters. Results are based on 30 independent replications of the experiment.

**Lemma 4.** *Suppose that the distribution of a random vector $\mathbf{x} \sim \mathcal{D}_{\mathbf{x}}$ satisfies the regularity conditions outlined in Assumption 2, then, it has the following properties.*

*(a) For $\forall \mathbf{w} \in \mathbb{S}^d$ and $\forall\, b > 0$, $L_1 \min\{R, b\} \leq \mathbb{P}\left(0 \leq \langle \mathbf{w}, \mathbf{x} \rangle \leq b\right) \leq U_1 b$.*

*(b) There exist positive constants $c_8, c_9 > 0$, such that for any two unit vectors $\mathbf{v}_1, \mathbf{v}_2 \in \mathbb{S}^d$, if $0 \leq \theta\left(\mathbf{v}_1, \mathbf{v}_2\right) \leq \frac{\pi}{2}$, then*

$$c_8 \mathbb{P}\left(sgn(\langle \mathbf{v}_1, \mathbf{x} \rangle) \neq sgn(\langle \mathbf{v}_2, \mathbf{x} \rangle)\right) \leq \theta\left(\mathbf{v}_1, \mathbf{v}_2\right) \leq c_9 \mathbb{P}\left(sgn(\langle \mathbf{v}_1, \mathbf{x} \rangle) \neq sgn\left(\langle \mathbf{v}_2, \mathbf{x} \rangle\right)\right). \tag{3}$$

*Proof.* (a) Since $\langle \mathbf{w}, \mathbf{x} \rangle$ forms a projection of $\mathbf{x}$ onto a certain 1-dimensional hyperplane, property (a) trivially holds by conditions 1 and 2 in Assumption 2.

(b) Let $\mathbf{z} := \left(\langle \mathbf{v}_1, \mathbf{x} \rangle, \langle \mathbf{v}_2, \mathbf{x} \rangle\right)$, which is a projection of $\mathbf{x}$ onto a 2-dimensional subspace $V_2$ spanned by $\mathbf{v}_1$ and $\mathbf{v}_2$. Let $\phi_{V_2}(\cdot)$ and $D_{V_2}$ denote its density and distribution, respectively. Let $G_{V_2} := \{\mathbf{z} \mid sgn(\mathbf{z}_1) \neq sgn(\mathbf{z}_2)\}$, then,

$$\begin{aligned}
\mathbb{P}_{\mathbf{x} \sim \mathcal{D}_{\mathbf{x}}}\left(sgn(\langle \mathbf{v}_1, \mathbf{x} \rangle) \neq sgn(\langle \mathbf{v}_2, \mathbf{x} \rangle)\right) &= \mathbb{P}_{\mathbf{z} \sim \mathcal{D}_{V_2}}\left(\mathbf{z} \in G_{V_2}\right) \\
&= \int_{\mathbf{z} \in G_V} \phi_{V_2}(\mathbf{z})\, d\mathbf{z} \\
&\geq \int_{\mathbf{z} \in G_V \cap \mathbb{B}^2(R)} L_2\, d\mathbf{z} \\
&\geq L_2 R^2 \theta\left(\mathbf{v}_1, \mathbf{v}_2\right),
\end{aligned}$$

where the first inequality holds by condition 1 of Assumption 2 that $L_2 \leq \phi_{V_2}(\mathbf{z})$ for all $\|\mathbf{z}\|_2 \leq R$. The last inequality holds by an observation that $\int_{\mathbf{z} \in G_{V_2} \cap \mathbb{B}^2(R)} 1\, d\mathbf{z} \geq R^2 \theta(\mathbf{v}_1, \mathbf{v}_2)$. Hence, we prove the first inequality of (3).

To prove the second inequality of (3), for $\forall \epsilon > 0$, we have

$$\begin{aligned}
&\mathbb{P}_{\mathbf{x} \sim \mathcal{D}_{\mathbf{x}}}\left(sgn(\langle \mathbf{v}_1, \mathbf{x} \rangle) \neq sgn(\langle \mathbf{v}_2, \mathbf{x} \rangle)\right) \\
&= \mathbb{P}_{\mathbf{z} \sim \mathcal{D}_{V_2}}\left(\mathbf{z} \in G_{V_2}\right) \\
&\leq \mathbb{P}_{\mathbf{z} \sim \mathcal{D}_{V_2}}\left(\mathbf{z} \in G_{V_2}, \|\mathbf{z}\|_2 \leq \epsilon\right) + \mathbb{P}_{\mathbf{z} \sim \mathcal{D}_{V_2}}\left(\|\mathbf{z}\|_2 > \epsilon\right) \\
&\leq \int_{\mathbf{z} \in G_{V_2} \cap \mathbb{B}^2(\epsilon)} \phi_{V_2}(\mathbf{z}) + \mathbb{P}_{\mathbf{x} \sim \mathcal{D}_{\mathbf{x}}}\left(|\langle \mathbf{v}_1, \mathbf{x} \rangle| > \epsilon\right) + \mathbb{P}_{\mathbf{x} \sim \mathcal{D}_{\mathbf{x}}}\left(|\langle \mathbf{v}_2, \mathbf{x} \rangle| > \epsilon\right) \\
&= \int_{\mathbf{z} \in G_{V_2} \cap \mathbb{B}^2(\epsilon)} \phi_{V_2}(\mathbf{z}) + \mathbb{P}_{\mathbf{x} \sim \mathcal{D}_{\mathbf{x}}}\left(|\langle \mathbf{v}_1, \mathbf{x} \rangle| > \epsilon\right) + \mathbb{P}_{\mathbf{x} \sim \mathcal{D}_{\mathbf{x}}}\left(|\langle \mathbf{v}_2, \mathbf{x} \rangle| > \epsilon\right) \\
&\leq U_2 \theta\left(\mathbf{v}_1, \mathbf{v}_2\right) \epsilon^2 + 2\exp(1 - Q\epsilon),
\end{aligned}$$

where the last inequality holds by the fact that $\phi_{V_2}(\mathbf{z}) \leq U_2 \exp(-\delta\|\mathbf{z}\|_2) \leq U_2$ according to Assumption 2 condition 2 and $\mathbb{P}_{\mathbf{x}\sim\mathcal{D}_\mathbf{x}}(|\langle\mathbf{w},\mathbf{x}\rangle| > \epsilon) \leq \exp(1 - Q\epsilon)$ for $\forall\mathbf{w} \in \mathbb{S}^d$ according to Assumption 2 Condition 3. Taking $\epsilon = \frac{1-\ln(\theta(\mathbf{v}_1,\mathbf{v}_2))}{Q}$, then we have

$$\mathbb{P}_{\mathbf{x}\sim\mathcal{D}_\mathbf{x}}(\operatorname{sgn}(\langle\mathbf{v}_1,\mathbf{x}\rangle) \neq \operatorname{sgn}(\langle\mathbf{v}_2,\mathbf{x}\rangle)) \leq \left(\frac{U_2}{Q^2} + 2\right)\theta(\mathbf{v}_1,\mathbf{v}_2).$$

Thus, we complete the proof of the second inequality in (3). $\qquad\square$

**Probability Tail bounds**

**Definition 3.** *(($\sigma, b$)-subexponential, Wainwright [2019], Definition 2.7) A random variable $X$ with mean $\mu = \mathbb{E}[X]$ is ($\sigma, b$)-subexponential, if for $\forall\lambda \in \left[-\frac{1}{b}, \frac{1}{b}\right]$,*

$$\mathbb{E}[\exp(\lambda(X - \mu))] \leq \exp\left(\frac{\sigma^2\lambda^2}{2}\right).$$

**Lemma 5.** *(($\sigma, b$)-subexponential tail bound, another form of Wainwright [2019], Proposition 2.9) Suppose $X$ is a ($\sigma, b$)-subexponential random variable with mean $\mathbb{E}[X] = \mu$, then with probability at least $1 - \delta$,*

$$X \leq \mu + \sqrt{2\sigma^2\ln\frac{1}{\delta}} + 2b\ln\frac{1}{\delta},$$

*also, with probability at least $1 - \delta$,*

$$X \geq \mu - \sqrt{2\sigma^2\ln\frac{1}{\delta}} - 2b\ln\frac{1}{\delta}.$$

**Lemma 6.** *(A Bernstein-type bound for i.i.d. random variables, another form of Wainwright [2019], Equation (2.18)) Suppose $\{X\}_{i=1}^N$ is sequence of i.i.d. ($\sigma, b$)-subexponential random variables, then, with probability at least $1 - \delta$,*

$$\sum_{i=1}^N X_i \leq \sum_{i=1}^N \mathbb{E}[X_i] + \sigma\sqrt{2N\ln\frac{1}{\delta}} + 2b\ln\frac{1}{\delta},$$

*and, with probability at least $1 - \delta$,*

$$\sum_{i=1}^N X_i \geq \sum_{i=1}^N \mathbb{E}[X_i] - \sigma\sqrt{2N\ln\frac{1}{\delta}} - 2b\ln\frac{1}{\delta}.$$

**Lemma 7.** *(A Bernstein-type bound for a martingale difference sequence, another form of Wainwright [2019], Theorem 2.19) Suppose $\{X\}_{i=1}^N$ is a sequence of conditionally ($\sigma, b$)-subexponential random variables adapted from filtration $\{\mathcal{F}_i\}_{i=1}^N$, i.e.,*

$$\mathbb{E}[\exp(\lambda(X_i - \mathbb{E}[X_i \mid \mathcal{F}_{i-1}])) \mid \mathcal{F}_{i-1}] \leq \exp\left(\frac{\sigma^2\lambda^2}{2}\right), \ \forall\lambda \in \left[-\frac{1}{b}, \frac{1}{b}\right].$$

*Then, with probability at least $1 - \delta$,*

$$\sum_{i=1}^N X_i \leq \sum_{i=1}^N \mathbb{E}[X_i \mid \mathcal{F}_{i-1}] + \sigma\sqrt{2N\ln\frac{1}{\delta}} + 2b\ln\frac{1}{\delta},$$

*and, with probability at least $1 - \delta$,*

$$\sum_{i=1}^N X_i \geq \sum_{i=1}^N \mathbb{E}[X_i \mid \mathcal{F}_{i-1}] - \sigma\sqrt{2N\ln\frac{1}{\delta}} - 2b\ln\frac{1}{\delta}.$$

**Lemma 8.** *(Azuma-Hoeffding's Inequality, another form of Wainwright [2019], Corollary 2.20) Suppose $\{X\}_{i=1}^N$ is a sequence adapted from filtration $\{\mathcal{F}_i\}_{i=1}^N$ such that $X_i \in [a, b]$, Then, with probability at least $1 - \delta$,*

$$\sum_{i=1}^N X_i \leq \sum_{i=1}^N \mathbb{E}\left[X_i \mid \mathcal{F}_{i-1}\right] + (b-a)\sqrt{\frac{1}{2}N\ln\frac{1}{\delta}},$$

*and, with probability at least $1 - \delta$,*

$$\sum_{i=1}^N X_t \geq \sum_{i=1}^N \mathbb{E}\left[X_i \mid \mathcal{F}_{i-1}\right] - (b-a)\sqrt{\frac{1}{2}N\ln\frac{1}{\delta}}.$$

We show in the following lemma how to determine the parameters $(\sigma, b)$ prescribed in Definition 3 by a given probability tail bound.

**Lemma 9.** *Suppose a random variable satisfies $\mathbb{P}\left(|X| \geq a\right) \leq C\exp(-\frac{a}{\nu})$ for given $C, \nu > 0$, then $X$ is $(6\nu\sqrt{1+2C}, 6\nu)$-subexponential. Also, if $Y$ is a random variable that satisfy $|Y| \leq M$, then, $XY$ is $(6M\nu\sqrt{1+2C}, 6M\nu)$-subexponential.*

*Proof.* First, consider $|X|$'s moment generating function $\mathbb{E}\left[e^{\lambda|X|}\right]$, for $\forall\lambda > 0$, we have

$$\mathbb{E}\left[e^{\lambda|X|}\right] = \int_0^{+\infty} \mathbb{P}\left(e^{\lambda|X|} \geq u\right)\, du$$

$$\leq 1 + \int_1^{+\infty} \mathbb{P}\left(|X| \geq \frac{\ln u}{\lambda}\right)\, du$$

$$\leq 1 + C\int_1^{+\infty} u^{-\frac{1}{\lambda\nu}}\, du.$$

From the above inequality, we get that $\mathbb{E}\left[e^{\lambda|X|}\right] \leq 1 + \frac{C\lambda\nu}{1-\lambda\nu} < \infty$ if $0 < \lambda < \frac{1}{\nu}$. Set $\lambda = \frac{2}{3\nu}$, as $\mathbb{E}\left[e^{\frac{2}{3\nu}|X|}\right] = \sum_{i=0}^\infty \frac{\mathbb{E}\left[|X|^i\right]}{(\frac{3}{2}\nu)^i i!}$, we have:

$$\frac{\mathbb{E}\left[|X|^i\right]}{(\frac{3}{2}\nu)^i i!} \leq \mathbb{E}\left[e^{\frac{2}{3\nu}|X|}\right] \leq 1 + 2C. \tag{4}$$

Now we introduce a new random variable $X'$ that is an independent copy of $X$, then we can bound $\mathbb{E}\left[\exp(\lambda(X - \mathbb{E}\left[X\right]))\right]$ by Jensen's inequality, $\mathbb{E}\left[\exp(\lambda(X - \mathbb{E}\left[X\right]))\right] \leq \mathbb{E}\left[\exp(\lambda(X - X'))\right]$. Therefore, we only need to bound $\mathbb{E}\left[\exp(\lambda(X - X'))\right]$. For $\forall\lambda \in \left[-\frac{1}{6\nu}, \frac{1}{6\nu}\right]$,

$$\mathbb{E}\left[\exp(\lambda(X - X'))\right] = \sum_{i=0}^\infty \frac{\mathbb{E}\left[(X-X')^i\lambda^i\right]}{i!} = \sum_{i=0}^\infty \frac{\mathbb{E}\left[(X-X')^{2i}\lambda^{2i}\right]}{(2i)!}$$

$$\leq 1 + \sum_{i=1}^\infty \frac{\mathbb{E}\left[|X|^{2i}\right]2^{2i}\lambda^{2i}}{(2i)!} \leq 1 + (1 + 2C)\sum_{i=1}^\infty \left(\frac{3}{2}\nu\right)^{2i} 2^{2i}\lambda^{2i}$$

$$= 1 + (1 + 2C)\sum_{i=1}^\infty (3\nu\lambda)^{2i} \leq 1 + 2(1 + 2C)(3\nu\lambda)^2$$

$$\leq \exp(2(1 + 2C)(3\nu\lambda)^2) = \exp\left(\frac{(6\nu\sqrt{1+2C})^2\lambda^2}{2}\right),$$

where the first equality holds by Taylor expansion, the second equality holds since $\mathbb{E}\left[(X - X')^i\right] = 0$ for all $i$'s that are odd. The first inequality holds by the fact that $|x - x'|^i \leq 2^{i-1}(|x|^i + |x'|^i)$ for all $i \geq 1$, and that $X$ and $X'$ have the same distribution. The second inequality holds by (4). The third inequality holds since $\sum_{i=1}^\infty (3\nu\lambda)^{2i} = \frac{(3\nu\lambda)^2}{1-(3\nu\lambda)^2} \leq 2 \times (3\nu\lambda)^2$ when $\lambda \in \left[-\frac{1}{6\nu}, \frac{1}{6\nu}\right]$. The last inequality holds by the fact that $1 + x \leq e^x$ for all $x \in \mathbb{R}$.

Thus, by definition of $(\sigma, b)$-subexponential, we conclude that $X$ is $(6\nu\sqrt{1+2C}, 6\nu)$-subexponential.

Now we prove the subexponential property of $XY$. Since $|Y| \leq M$ and $\mathbb{P}(|X| \geq a) \leq C\exp(-\frac{a}{\nu})$,

$$\mathbb{P}(|XY| \geq a) \leq \mathbb{P}\left(|X| \geq \frac{a}{M}\right) \leq C\exp\left(-\frac{a}{M\nu}\right).$$

Replacing $\nu$ by $M\nu$, we conclude that $XY$ is $(6M\nu\sqrt{1+2C}, 6M\nu)$-subexponential. $\qquad\square$

Lemma 9 directly accommodates our regularity assumption and leads to the following corollary.

**Corollary 2.** *(Subexponential property of regular distributions) Suppose $\mathbf{x}$ is a random variable that satisfies Assumption 2, then, for $\forall\, \mathbf{w} \in \mathbb{B}^d$, $\langle \mathbf{w}, \mathbf{x} \rangle$ is $(\frac{16}{Q}, \frac{6}{Q})$-subexponential.*

*Proof.* By Assumption 2 Condition 3, $\mathbb{P}[|\langle \mathbf{w}, \mathbf{x}\rangle| \geq t] \leq \exp(1 - Qt)$ for $\forall\, t > 0$. Then by Lemma 9, set $C = e$, $\nu = \frac{1}{Q}$, then, $6\nu\sqrt{1+2C} = \frac{6\sqrt{1+2e}}{Q} < \frac{16}{Q}$, $6\nu = \frac{6}{Q}$, we conclude that $\mathbf{x}$ is $\left(\frac{16}{Q}, \frac{6}{Q}\right)$-subexponential. $\qquad\square$

**The relationship between** $\mathbb{E}\left[\langle \mathbf{w}^\star, -\mathbf{g}_{k,i}\rangle\right]$ **and** $\theta(\mathbf{w}_{k,i}, \mathbf{w}^\star)$ Recall that in the non-strategic setting, we shall adjust the coefficient by solving a sequence of adaptively constructed online regret minimization problems $\min_{\mathbf{w} \in \mathcal{W}_k} \sum_{i=1}^{T_k} \langle \mathbf{w}, \mathbf{g}_{k,i}\rangle - \sum_{i=1}^{T_k} \langle \mathbf{w}^*, \mathbf{g}_{k,i}\rangle$ with $\mathbf{g}_{k,i} = [-\bar{\eta}\mathbf{x}_{k,i}\mathbb{I}(y_{k,i} = 1) + (1 - \bar{\eta})\mathbf{x}_{k,i}\mathbb{I}(y_{k,i} = -1)]\mathbb{I}(\mathbf{x}_{k,i} \in D_{k,i})$ via mirror descent over $k = 0 \ldots, K$ batches, using local data within increasingly narrow bands $D_{k,i} = \{\mathbf{x} : 0 < \langle \mathbf{w}_{k,i}, \mathbf{x}\rangle \leq b_k\}$. The key ingredient underlying this guarantee is that the gradients $\mathbf{g}_{k,i}$ are well constructed so that $|\langle \mathbf{w}_{k,i}, \mathbf{g}_{k,i}\rangle|$ is small and meanwhile $\mathbb{E}\left[\langle \mathbf{w}^\star, -\mathbf{g}_{k,i}\rangle\right]$ upper bounds $\theta(\mathbf{w}^\star, \mathbf{w}_{k,i})$. Here, we show the relationship between $\mathbb{E}\left[\langle \mathbf{w}^\star, -\mathbf{g}_{k,i}\rangle\right]$ and $\theta(\mathbf{w}_{k,i}, \mathbf{w}^\star)$ for $k = 0, 1, \cdots, K$, which is critical in the guarantees of Algorithm 3 and Algorithm 4.

Fix batch $k$ and iteration $i$, to connect $\mathbb{E}\left[\langle \mathbf{w}^\star, -\mathbf{g}_{k,i}\rangle\right]$ and $\theta(\mathbf{w}_{k,i}, \mathbf{w}^\star)$ we introduce a new variable $f_{k,i}(\mathbf{w}_{k,i})$ in (5). Later, we will show how $\mathbb{E}\left[\langle \mathbf{w}^\star, -\mathbf{g}_{k,i}\rangle\right]$ upper bounds $f_{k,i}(\mathbf{w}_{k,i})$ and how $f_{k,i}(\mathbf{w}_{k,i})$ approximates $\theta(\mathbf{w}_{k,i}, \mathbf{w}^\star)$.

$$f_{k,i}(\mathbf{w}_{k,i}) := \mathbb{E}\left[|\langle \mathbf{w}^\star, \mathbf{x}\rangle|\,\mathbb{I}(\langle \mathbf{w}^\star, \mathbf{x}\rangle < 0)\,|\,\mathbf{x} \in D_{k,i}\right]. \tag{5}$$

Now, we show that $\mathbb{E}\left[\langle \mathbf{w}^\star, -\mathbf{g}_{k,i}\rangle\right]$ can upper bound $f_{k,i}(\mathbf{w}_{k,i})$ by the following lemma.

**Lemma 10.** *Given a unit vector $\mathbf{w}_{k,i} \in \mathbb{S}^d$ and an agent with true feature-label pair $(\mathbf{x}_{k,i}, y_{k,i})$. For $\mathbf{g}_{k,i} = [-\bar{\eta}\mathbf{x}_{k,i}\mathbb{I}(y_{k,i} = 1) + (1 - \bar{\eta})\mathbf{x}_{k,i}\mathbb{I}(y_{k,i} = -1)]\mathbb{I}(\mathbf{x}_{k,i} \in D_{k,i})$ and $f_{k,i}(\mathbf{w}_{k,i})$ defined in (5). The following holds.*

$$\mathbb{E}\left[\langle \mathbf{w}^\star, -\mathbf{g}_{k,i}\rangle\right] \geq (1 - 2\bar{\eta})f_{k,i}(\mathbf{w}_{k,i})\mathbb{P}(\mathbf{x} \in D_{k,i}).$$

*Proof.* First, for convenience, we rewrite $\mathbf{g}_{k,i}$ as the following.

$$\begin{aligned}
\mathbf{g}_{k,i} &= [-\bar{\eta}\mathbf{x}_{k,i}\mathbb{I}(y_{k,i} = 1) + (1 - \bar{\eta})\mathbf{x}_{k,i}\mathbb{I}(y_{k,i} = -1)]\mathbb{I}(\mathbf{x}_{k,i} \in D_{k,i}) \\
&= \left(-\frac{1}{2}y_{k,i} + \left(\frac{1}{2} - \bar{\eta}\right)\right)\mathbf{x}_{k,i}\mathbb{I}(\mathbf{x}_{k,i} \in D_{k,i}).
\end{aligned}$$

Then, we have

$$
\begin{aligned}
\mathbb{E}\left[\langle \mathbf{w}^\star, -\mathbf{g}_{k,i}\rangle\right] &= \mathbb{E}\left[\left\langle \mathbf{w}^\star, \left(\frac{1}{2}y_{k,i} - \left(\frac{1}{2} - \bar{\eta}\right)\right)\mathbf{x}_{k,i}\right\rangle \mathbb{I}\left(\mathbf{x}_{k,i} \in D_{k,i}\right)\right] \\
&= \mathbb{E}\left[\left\langle \mathbf{w}^\star, \left(\frac{1}{2}y_{k,i} - \left(\frac{1}{2} - \bar{\eta}\right)\right)\mathbf{x}_{k,i}\right\rangle \middle| \mathbf{x}_{k,i} \in D_{k,i}\right]\mathbb{P}\left(\mathbf{x}_{k,i} \in D_{k,i}\right) + 0 \\
&= \frac{1}{2}\,\mathbb{E}\left[\langle \mathbf{w}^\star, \mathbf{x}_{k,i}\rangle\,\mathbb{E}\left[y_{k,i} \mid \mathbf{x}_{k,i}\right] \mid \mathbf{x}_{k,i} \in D_{k,i}\right]\mathbb{P}\left(\mathbf{x} \in D_{k,i}\right) \\
&\quad - \left(\frac{1}{2} - \bar{\eta}\right)\mathbb{E}\left[\langle \mathbf{w}^\star, \mathbf{x}_{k,i}\rangle \mid \mathbf{x}_{k,i} \in D_{k,i}\right]\mathbb{P}\left(\mathbf{x} \in D_{k,i}\right) \\
&\geq \left(\frac{1}{2} - \bar{\eta}\right)\mathbb{E}\left[|\langle \mathbf{w}^\star, \mathbf{x}_{k,i}\rangle| \mid \mathbf{x}_{k,i} \in D_{k,i}\right]\mathbb{P}\left(\mathbf{x} \in D_{k,i}\right) \\
&\quad - \left(\frac{1}{2} - \bar{\eta}\right)\mathbb{E}\left[\langle \mathbf{w}^\star, \mathbf{x}_{k,i}\rangle \mid \mathbf{x}_{k,i} \in D_{k,i}\right]\mathbb{P}\left(\mathbf{x} \in D_{k,i}\right) \\
&= (1 - 2\bar{\eta})\,\mathbb{E}\left[|\langle \mathbf{w}^\star, \mathbf{x}_{k,i}\rangle|\,\mathbb{I}\left(\langle \mathbf{w}^\star, \mathbf{x}_{k,i}\rangle < 0\right) \mid \mathbf{x}_{k,i} \in D_{k,i}\right]\mathbb{P}\left(\mathbf{x} \in D_{k,i}\right) \\
&= (1 - 2\bar{\eta})\,f_{k,i}(\mathbf{w}_{k,i})\mathbb{P}\left(\mathbf{x} \in D_{k,i}\right),
\end{aligned}
$$

where the second and third equality hold by the law of iterated expectations, and the inequality holds as the following.

$$
\begin{aligned}
\mathbb{E}\left[y_{k,i} \mid \mathbf{x}_{k,i}\right] &= (1 - \eta(\mathbf{x}_{k,i}))\mathrm{sgn}(\langle \mathbf{w}^\star, \mathbf{x}_{k,i}\rangle) - \eta(\mathbf{x}_{k,i})\mathrm{sgn}(\langle \mathbf{w}^\star, \mathbf{x}_{k,i}\rangle) \\
&= (1 - 2\eta(\mathbf{x}_{k,i}))\mathrm{sgn}(\langle \mathbf{w}^\star, \mathbf{x}_{k,i}\rangle) \\
&\geq (1 - 2\bar{\eta})\mathrm{sgn}(\langle \mathbf{w}^\star, \mathbf{x}_{k,i}\rangle).
\end{aligned}
$$

$\square$

Next, in the following lemma, we show that $f_{k,i}(\mathbf{w}_{k,i})$ measures the closeness of $\mathbf{w}^\star$ and $\mathbf{w}_{k,i}$.

**Lemma 11.** *For fixed batch $k$ and iteration $i$, if $\theta\left(\mathbf{w}^\star, \mathbf{w}_{k,i}\right) = \phi$, then the following holds.*

1. *When $0 < b_k \leq \frac{R}{4}$ and $\phi \in \left[\frac{4b_k}{R}, \frac{\pi}{2}\right]$, we have*

$$
f_{k,i}(\mathbf{w}_{k,i}) \geq \frac{L_2}{32U_1}R^2\phi.
$$

2. *When $0 < b_k \leq \frac{R}{4}$ and $\phi \in \left[\frac{\pi}{2}, \pi - \frac{4b_k}{R}\right]$, we have*

$$
f_{k,i}(\mathbf{w}_{k,i}) \geq \frac{L_2}{32U_1}R^2(\pi - \phi).
$$

*Proof.* We prove the two cases respectively. For Case 1, define the region $G_1 := \left\{\mathbf{x} \mid 0 \leq \langle \mathbf{w}_{k,i}, \mathbf{x}\rangle \leq b_k, -\frac{1}{2}R\sin\phi \leq \langle \mathbf{w}^\star, \mathbf{x}\rangle \leq -\frac{1}{4}R\sin\phi\right\}$, see Figure 6 as an illustration. We have the following

$$
\begin{aligned}
&\mathbb{E}\left[|\langle \mathbf{w}^\star, \mathbf{x}\rangle|\,\mathbb{I}\left(\langle \mathbf{w}^\star, \mathbf{x}\rangle < 0\right)\mathbb{I}\left(0 \leq \langle \mathbf{w}_{k,i}, \mathbf{x}\rangle \leq b_k\right)\right] \\
&\geq \mathbb{E}\left[|\langle \mathbf{w}^\star, \mathbf{x}\rangle|\,\mathbb{I}\left(\mathbf{x} \in G_1\right)\right] \\
&\geq \frac{1}{4}R\sin\phi\,\mathbb{E}\left[\mathbb{I}\left(\mathbf{x} \in G_1\right)\right] \\
&\geq \frac{1}{8}R\phi\,\mathbb{P}\left(\mathbf{x} \in G_1\right) \\
&\geq \frac{1}{32}L_2R^2\phi b_k,
\end{aligned}
$$

where the first inequality holds since $G_1 \subseteq \{\mathbf{x} \mid 0 \leq \langle \mathbf{w}_{k,i}, \mathbf{x}\rangle \leq b, \langle \mathbf{w}^\star, \mathbf{x}\rangle < 0\}$. The third inequality holds by the fact that $\sin\phi \geq \frac{\phi}{2}$ for $0 \leq \phi \leq \frac{\pi}{2}$. And the last inequality holds by the claim that $\mathbb{P}\left(\mathbf{x} \in G_1\right) \geq \frac{1}{4}L_2Rb_k$, which we will show later.

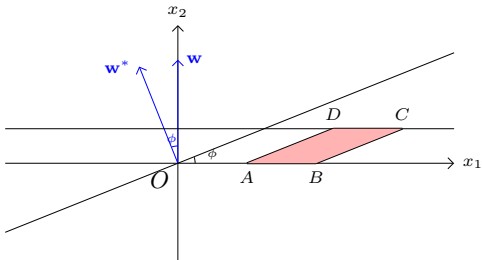

Figure 6: Illustration of region $G_1$ (the red region) in Case 1. Which satisfies $G_1 = \left\{ \mathbf{x} \,|\, 0 \le \langle \mathbf{w}, \mathbf{x} \rangle \le b_k, \; -\frac{1}{2} R \sin \phi \le \langle \mathbf{w}^\star, \mathbf{x} \rangle \le -\frac{1}{4} R \sin \phi \right\}$.

Hence, we can establish the lower bound of $f_{k,i}(\mathbf{w}_{k,i})$ by:

$$
\begin{aligned}
f_{k,i}(\mathbf{w}_{k,i}) &= \mathbb{E}\left[ | \langle \mathbf{w}^\star, \mathbf{x} \rangle | \, \mathbb{I}(\langle \mathbf{w}^\star, \mathbf{x} \rangle < 0) \,|\, 0 \le \langle \mathbf{w}_{k,i}, \mathbf{x} \rangle \le b_k \right] \\
&= \frac{\mathbb{E}\left[ | \langle \mathbf{w}^\star, \mathbf{x} \rangle | \, \mathbb{I}(\langle \mathbf{w}^\star, \mathbf{x} \rangle < 0) \, \mathbb{I}(0 \le \langle \mathbf{w}_{k,i}, \mathbf{x} \rangle \le b_k) \right]}{\mathbb{P}(0 \le \langle \mathbf{w}_{k,i}, \mathbf{x} \rangle \le b_k)} \\
&\ge \frac{\frac{1}{32} L_2 R^2 \phi b_k}{\mathbb{P}(0 \le \langle \mathbf{w}_{k,i}, \mathbf{x} \rangle \le b_k)} \\
&\ge \frac{L_2}{32 U_1} R^2 \phi,
\end{aligned}
$$

where the last inequality holds by Lemma 4, property (a).

Now we show the claim that $\mathbb{P}(\mathbf{x} \in G_1) \ge \frac{1}{4} L_2 R b_k$. For a given vector $\mathbf{x}$, we first project $\mathbf{x}$ down to the subspace $V_2 \subseteq \mathbb{R}^2$ spanned by $\mathbf{w}^\star$ and $\mathbf{w}_{k,i}$ and denote the projected value $\mathbf{z} := (\langle \mathbf{w}^\star, \mathbf{x} \rangle, \langle \mathbf{w}, \mathbf{x} \rangle)$.

Without loss of generality, let $\mathbf{w} = (0, 1)$ and $\mathbf{w}^\star = (\sin \phi, \cos \phi)$. As illustrated in Figure 6, the parallelogram ABCD denotes the region $G_1$, where $A = \left( \frac{1}{4} R, 0 \right)$, $B = \left( \frac{1}{2} R, 0 \right)$, $C = \left( \frac{1}{2} R + \frac{b_k}{\tan \phi}, b_k \right)$, $D = \left( \frac{1}{4} R + \frac{b_k}{\tan \phi}, b_k \right)$. Since $C$ is the farthest point to the origin with respect to the Euclidean Norm, and $\left\| \left( \frac{1}{2} R + \frac{b_k}{\tan \phi}, b_k \right) \right\|_2 \le \left\| \left( \frac{1}{2} R + \frac{b_k}{\tan \phi}, b_k \right) \right\|_1 = \frac{1}{2} R + \frac{b_k}{\tan \phi} + b_k \le \frac{1}{2} R + \frac{b_k}{\phi} + b_k \le R$, then for all $\mathbf{z} \in \left\{ \mathbf{z} = (z_1, z_2) \,|\, -\frac{1}{2} R \sin \phi \le z_1 \le -\frac{1}{4} \sin \phi, \; 0 \le z_2 \le b_k \right\}$, we have $\| \mathbf{z} \|_2 \le R$. Also, the area of parallelogram ABCD is $b_k \cdot \frac{1}{4} R = \frac{1}{4} R b_k$. In addition, by Assumption 2, condition 1, the density $\phi_{V_2}(\mathbf{z})$ of projected value $\mathbf{z}$ satisfies $\phi_{V_2}(\mathbf{z}) \ge L_2$ for all $\mathbf{z} \in V_2 \cap \mathbb{B}^2(R)$. Hence, we can lower bound $\mathbb{P}(\mathbf{x} \in G_1)$ by

$$
\mathbb{P}(\mathbf{x} \in G_1) \ge L_2 \cdot \frac{1}{4} R b_k = \frac{1}{4} L_2 R b_k.
$$

The proof of Case 2 is similar to that of Case 1. Define the region $G_2 = \left\{ \mathbf{x} \,|\, 0 \le \langle \mathbf{w}_{k,i}, \mathbf{x} \rangle \le b_k, \; -\frac{1}{2} R \sin(\pi - \phi) \le \langle \mathbf{w}^\star, \mathbf{x} \rangle \le -\frac{1}{4} R \sin(\pi - \phi) \right\}$ (see Figure 7). We replace $\phi$ in the proof of Case 1 by $\pi - \phi$, by choosing $A = \left( -\frac{1}{4} R, 0 \right)$, $B = \left( -\frac{1}{2} R, 0 \right)$, $C = \left( -\frac{1}{2} R + \frac{b_k}{\tan(\pi - \phi)}, b_k \right)$, $D = \left( -\frac{1}{4} R + \frac{b_k}{\tan(\pi - \phi)}, b_k \right)$ and then we complete the proof. $\qquad \square$

Lemma 11 directly leads to the following corollary.

**Corollary 3.** *If $\theta(\mathbf{w}^\star, \mathbf{w}_{k,i}) \le \frac{\pi}{2}$, and $f_{k,i}(\mathbf{w}_{k,i}) \le \frac{L_2}{160 U_1} R^2 \theta_k$, then $\theta(\mathbf{w}^\star, \mathbf{w}_{k,i}) \le \frac{\theta_k}{5}$.*

*Proof.* We conduct a case analysis.

1. If $\theta(\mathbf{w}^\star, \mathbf{w}_{k,i}) < \frac{4 b_k}{R}$, then by our setting of $b_k$ in Algorithm 3 and Algorithm 4, $\theta(\mathbf{w}^\star, \mathbf{w}_{k,i}) < \frac{4 b_k}{R} \le \frac{\theta_k}{5}$.

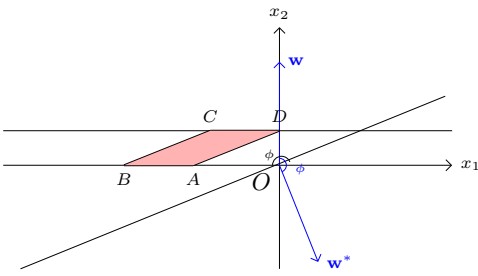

Figure 7: Illustration of region $G_2$ (the red region) in Case 2. Which satisfies $G_2 = \left\{\mathbf{x} \mid 0 \leq \langle \mathbf{w}_{k,i}, \mathbf{x} \rangle \leq b_k, -\frac{1}{2}R\sin(\pi - \phi) \leq \langle \mathbf{w}^\star, \mathbf{x} \rangle \leq -\frac{1}{4}R\sin(\pi - \phi)\right\}$.

2. If $\frac{4b_k}{R} \leq \theta(\mathbf{w}^\star, \mathbf{w}_{k,i}) \leq \frac{\pi}{2}$, then by Lemma 11, we have $f_{k,i}(\mathbf{w}_{k,i}) \geq \frac{L_2}{32U_1}R^2\theta(\mathbf{w}^\star, \mathbf{w}_{k,i})$, combing it with the condition that $f_{k,i}(\mathbf{w}_{k,i}) \leq \frac{L_2}{160U_1}R^2\theta_k$, we have $\theta(\mathbf{w}^\star, \mathbf{w}_{k,i}) \leq \frac{\theta_k}{5}$.

$\square$

**Other lemmas** We also outline some other lemmas that are used in the subsequent proof.

**Lemma 12.** *(Triangular inequality of angles) Suppose vectors $\mathbf{x}, \mathbf{y}, \mathbf{z} \in \mathbb{R}^d$ satisfy $0 \leq \theta(\mathbf{x}, \mathbf{y}) \leq \frac{\pi}{2}$ and $0 \leq \theta(\mathbf{x}, \mathbf{z}) \leq \frac{\pi}{2}$, then,*

$$|\theta(\mathbf{x}, \mathbf{y}) - \theta(\mathbf{x}, \mathbf{z})| \leq \theta(\mathbf{y}, \mathbf{z}) \leq \theta(\mathbf{x}, \mathbf{y}) + \theta(\mathbf{x}, \mathbf{z}).$$

*Proof.* Without loss of generality, assume that $\|\mathbf{x}\|_2 = \|\mathbf{y}\|_2 = \|\mathbf{z}\|_2 = 1$. First, we decompose vectors $\mathbf{y}$ and $\mathbf{z}$ into components along the direction of $\mathbf{x}$ and perpendicular to $\mathbf{x}$, respectively, as:

$$\mathbf{y} = \cos(\theta(\mathbf{x}, \mathbf{y}))\mathbf{x} + \sin(\theta(\mathbf{x}, \mathbf{y}))\mathbf{y}_{\perp\mathbf{x}},$$
$$\mathbf{z} = \cos(\theta(\mathbf{x}, \mathbf{z}))\mathbf{x} + \sin(\theta(\mathbf{x}, \mathbf{z}))\mathbf{z}_{\perp\mathbf{x}},$$

where $\mathbf{y}_{\perp\mathbf{x}}$ and $\mathbf{z}_{\perp\mathbf{x}}$ are unit vectors that are perpendicular to $\mathbf{x}$. Hence, we have

$$\cos(\theta(\mathbf{y}, \mathbf{z})) = \langle \mathbf{y}, \mathbf{z} \rangle = \cos(\theta(\mathbf{x}, \mathbf{y}))\cos(\theta(\mathbf{x}, \mathbf{z})) + \sin(\theta(\mathbf{x}, \mathbf{y}))\sin(\theta(\mathbf{x}, \mathbf{z}))\langle \mathbf{y}_\perp, \mathbf{z}_\perp \rangle,$$

where the second equality holds since $\langle \mathbf{x}, \mathbf{y}_\perp \rangle = \langle \mathbf{x}, \mathbf{z}_\perp \rangle = 0$. Since $\mathbf{y}_\perp$ and $\mathbf{z}_\perp$ are both unit vectors, by Cauchy-Schwarz inequality, $-1 \leq \langle \mathbf{y}_\perp, \mathbf{z}_\perp \rangle \leq 1$. Also, since $0 \leq \theta(\mathbf{x}, \mathbf{y}) \leq \frac{\pi}{2}$ and $0 \leq \theta(\mathbf{x}, \mathbf{z}) \leq \frac{\pi}{2}$, we have $\sin(\theta(\mathbf{x}, \mathbf{y}))\sin(\theta(\mathbf{x}, \mathbf{z})) \geq 0$. Putting all together, we have:

$$
\begin{aligned}
\cos(\theta(\mathbf{y}, \mathbf{z})) &\leq \cos(\theta(\mathbf{x}, \mathbf{y}))\cos(\theta(\mathbf{x}, \mathbf{z})) + \sin(\theta(\mathbf{x}, \mathbf{y}))\sin(\theta(\mathbf{x}, \mathbf{z})) \\
&= \cos(\theta(\mathbf{x}, \mathbf{y}) - \theta(\mathbf{x}, \mathbf{z})),
\end{aligned}
\tag{6}
$$

and

$$
\begin{aligned}
\cos(\theta(\mathbf{y}, \mathbf{z})) &\geq \cos(\theta(\mathbf{x}, \mathbf{y}))\cos(\theta(\mathbf{x}, \mathbf{z})) - \sin(\theta(\mathbf{x}, \mathbf{y}))\sin(\theta(\mathbf{x}, \mathbf{z})) \\
&= \cos(\theta(\mathbf{x}, \mathbf{y}) + \theta(\mathbf{x}, \mathbf{z})).
\end{aligned}
\tag{7}
$$

Since $\cos(x)$ is decreasing in $x \in [0, \pi]$, and $\cos(x) = \cos(-x)$ for all $x \in \mathbb{R}$, we get $\theta(\mathbf{y}, \mathbf{z}) \geq |\theta(\mathbf{x}, \mathbf{y}) - \theta(\mathbf{x}, \mathbf{z})|$ from (6), and $\theta(\mathbf{y}, \mathbf{z}) \leq \theta(\mathbf{x}, \mathbf{y}) + \theta(\mathbf{x}, \mathbf{z})$ from (7). $\square$

**Lemma 13.** *Given a random vector $\mathbf{x} \sim \mathcal{D}_\mathbf{x}$ that satisfy Assumption 2, then with probability at least $1 - \delta$,*

$$\|\mathbf{x}\|_\infty \leq \frac{1}{Q}\left(1 + \ln\left(\frac{d}{\delta}\right)\right).$$

*Proof.* We bound $\|\mathbf{x}\|_\infty$ element-wisely. Given $\mathbf{x} \sim \mathcal{D}_\mathbf{x}$ and $j \in [d]$, let $x_j$ be the $j$-th coordinate of $\mathbf{x}$. Let $\mathbf{e}^{[j]} \in \mathbb{R}^d$ denote the unit vector whose $j$'th coordinate is 1 while other coordinate is 0, then, by Assumption 2, condition 3, for $\forall a > 0$

$$\mathbb{P}(|x_j| \geq a) = \mathbb{P}\left(\left|\left\langle \mathbf{e}^{[j]}, \mathbf{x} \right\rangle\right| \geq a\right) \leq \exp(1 - Qa).$$

Taking union bound over all coordinates, we have

$$\mathbb{P}\left(\|\mathbf{x}\|_\infty \geq a\right) \leq d \exp(1 - Qa).$$

Taking $a = \frac{1}{Q}\left(1 + \ln\left(\frac{d}{\delta}\right)\right)$ in the above inequality and hence we complete the proof. $\qquad\square$

## A.4  Proofs for Section 2

In this subsection, we outline some important results to describe the relationships between classifier $h(\cdot)$ for unmanipulated features $\mathbf{x}$ and classifier $\tilde{h}$ for reported features $\mathbf{r} = \mathbf{r}^\star(\mathbf{x}, \tilde{h})$, which is critical for the subsequent algorithm design.

### Proof of Proposition 1

**Proposition 1.** *For any* $(\mathbf{w}, m) \in \mathbb{S}^d \times \mathbb{R}$*, the output of* $\tilde{h}(\mathbf{r}) = sgn(\langle \mathbf{w}, \mathbf{r}\rangle + m - \gamma)$ *for* $\mathbf{r} = \mathbf{r}^\star(\mathbf{x}, \tilde{h})$ *is identical to the output of* $h(\mathbf{x}) = sgn(\langle \mathbf{w}, \mathbf{x}\rangle + m)$ *for any* $\mathbf{x} \in \mathbb{R}^d$*.*

*Proof.* For fixed $\mathbf{w} \in \mathbb{S}^d$ and $m \in \mathbb{R}$, we categorize the agent population into three classes according to their true features $\mathbf{x}$: $\{\mathbf{x} \mid \langle \mathbf{w}, \mathbf{x}\rangle + m < 0\}$, $\{\mathbf{x} \mid 0 \leq \langle \mathbf{w}, \mathbf{x}\rangle + m < \gamma\}$ and $\{\mathbf{x} \mid \langle \mathbf{w}, \mathbf{x}\rangle + m \geq \gamma\}$. Then, we discuss their classification output by $h(\mathbf{x})$ and $\tilde{h}(\mathbf{r})$ with respect to $\mathbf{r} = \mathbf{r}^\star(\mathbf{x}, \tilde{h})$, respectively.

1. When $\langle \mathbf{w}, \mathbf{x}\rangle + m < 0$, $h(\mathbf{x}) = -1$. At the same time, $\langle \mathbf{w}, \mathbf{x}\rangle + m - \gamma < -\gamma$, by Lemma 1, agent in this region will report his feature truthfully, *i.e.*, $\mathbf{r} = \mathbf{x}$. Thus, $\tilde{h}(\mathbf{r}) = \mathrm{sgn}(\langle \mathbf{w}, \mathbf{r}\rangle + m - \gamma) = \mathrm{sgn}(\langle \mathbf{w}, \mathbf{x}\rangle + m - \gamma) = -1$. Hence, we have $h(\mathbf{x}) = \tilde{h}(\mathbf{r}) = -1$ for $\forall \mathbf{x} \in \{\mathbf{x} \mid \langle \mathbf{w}, \mathbf{x}\rangle + m < 0\}$.

2. When $0 \leq \langle \mathbf{w}, \mathbf{x}\rangle + m < \gamma$, $h(\mathbf{x}) = 1$. At the same time, $-\gamma \leq \langle \mathbf{w}, \mathbf{x}\rangle + m - \gamma < 0$, by Lemma 1, agent in this region will manipulate his feature as $\mathbf{r} = \mathbf{x} + (\gamma - m - \langle \mathbf{w}, \mathbf{x}\rangle)\mathbf{w}$. Thus, $\tilde{h}(\mathbf{r}) = \mathrm{sgn}(\langle \mathbf{w}, \mathbf{r}\rangle + m - \gamma) = \mathrm{sgn}(\langle \mathbf{w}, \mathbf{x} + (\gamma - m - \langle \mathbf{w}, \mathbf{x}\rangle)\mathbf{w}\rangle + m - \gamma) = 1$. Hence, we have $h(\mathbf{x}) = \tilde{h}(\mathbf{r}) = 1$ for $\forall \mathbf{x} \in \{\mathbf{x} \mid 0 \leq \langle \mathbf{w}, \mathbf{x}\rangle + m < \gamma\}$.

3. When $\langle \mathbf{w}, \mathbf{x}\rangle + m \geq \gamma$, $h(\mathbf{x}) = 1$. At the same time, $\langle \mathbf{w}, \mathbf{x}\rangle + m - \gamma \geq 0$, by Lemma 1, agent in this region will report his feature truthfully, *i.e.*, $\mathbf{r} = \mathbf{x}$. Thus, $\tilde{h}(\mathbf{r}) = \mathrm{sgn}(\langle \mathbf{w}, \mathbf{r}\rangle + m - \gamma) = \mathrm{sgn}(\langle \mathbf{w}, \mathbf{x}\rangle + m - \gamma) = 1$. Hence, we have $h(\mathbf{x}) = \tilde{h}(\mathbf{r}) = 1$ for $\forall \mathbf{x} \in \{\mathbf{x} \mid \langle \mathbf{w}, \mathbf{x}\rangle + m \geq \gamma\}$.

$\qquad\square$

**Inferring agents' true features from their reported features**   Proposition 1 directly leads to the following corollary, enabling us to infer an agent's true features $\mathbf{x}$ given a classification rule $\tilde{h}(\cdot)$ and his corresponding reported features $\mathbf{r}$.

**Corollary 4.** *For given announced classifier* $\tilde{h}(\cdot) = sgn\left(\langle \mathbf{w}, \cdot\rangle + m\right)$ *and agent response* $\mathbf{r}$*, then, his true features* $\mathbf{x}$ *satisfy the following.*

1. *if* $\langle \mathbf{w}, \mathbf{r}\rangle + m \neq 0$*, then* $\mathbf{x} = \mathbf{r}$*;*

2. *if* $\langle \mathbf{w}, \mathbf{r}\rangle + m = 0$*, then* $-\gamma \leq \langle \mathbf{w}, \mathbf{x}\rangle + m \leq 0$*.*

## A.5  Proofs for Section 3

In this subsection, we show the theoretical guarantees of Algorithm 2, Algorithm 3 and Algorithm 4, respectively.

**Theoretical Guarantees of Algorithm 2**   A key observation in the non-strategic and noiseless classification scenario is that $y\langle \mathbf{w}^\star, \mathbf{x}\rangle > 0$ holds for all $(\mathbf{x}, y)$. Consequently, $\mathbf{x}y$ always forms an acute angle with the optimal normal vector $\mathbf{w}^\star$. Considering agents' strategic responses and the bandit feedback setting, we introduce Lemma 14 to show that, under our construction in Algorithm 2, $\bar{\mathbf{w}}_0$ is an unbiased estimator of $\mathbb{E}[\mathbf{x}y]$.

**Lemma 14.** *In Algorithm 2, we have*

$$\mathbb{E}[\bar{\mathbf{w}}_0] = \mathbb{E}[\mathbf{x}y].$$

*Proof.* In Algorithm 2, at iteration $i$, the principal declares classifier $\tilde{h}_{\text{init,i}}^{(1)}(\mathbf{r}) = \text{sgn}(\langle \mathbf{w}_{\text{init,i}}, \mathbf{r} \rangle)$ and $\tilde{h}_{\text{init,i}}^{(2)}(\mathbf{r}) = \text{sgn}(\langle -\mathbf{w}_{\text{init,i}}, \mathbf{r} \rangle)$, and receives response $\mathbf{r}_{\text{init,i}}^{(1)}$ and $\mathbf{r}_{\text{init,i}}^{(2)}$ respectively. By Corollary 4, we get that $\mathbf{r}_{\text{init,i}}^{(1)} \mathbb{I} \left( \left\langle \mathbf{w}_{\text{init,i}}, \mathbf{r}_{\text{init,i}}^{(1)} \right\rangle > 0 \right) = \mathbf{x}_{\text{init,i}}^{(1)} \mathbb{I} \left( \left\langle \mathbf{w}_{\text{init,i}}, \mathbf{x}_{\text{init,i}}^{(1)} \right\rangle > 0 \right)$ and $\mathbf{r}_{\text{init,i}}^{(2)} \mathbb{I} \left( \left\langle -\mathbf{w}_{\text{init,i}}, \mathbf{r}_{\text{init,i}}^{(2)} \right\rangle > 0 \right) = \mathbf{x}_{\text{init,i}}^{(2)} \mathbb{I} \left( \left\langle -\mathbf{w}_{\text{init,i}}, \mathbf{x}_{\text{init,i}}^{(2)} \right\rangle > 0 \right)$. Also, $\left( \mathbf{x}_{\text{init,i}}^{(1)}, y_{\text{init,i}}^{(1)} \right)$ and $\left( \mathbf{x}_{\text{init,i}}^{(2)}, y_{\text{init,i}}^{(2)} \right)$ are i.i.d. drawn from $\mathcal{D}$, hence,

$$\mathbb{E}\left[ \mathbf{r}_{\text{init,i}}^{(1)} y_{\text{init,i}}^{(1)} \mathbb{I} \left( \left\langle \mathbf{w}_{\text{init,i}}, \mathbf{r}_{\text{init,i}}^{(1)} \right\rangle > 0 \right) + \mathbf{r}_{\text{init,i}}^{(2)} y_{\text{init,i}}^{(2)} \mathbb{I} \left( \left\langle -\mathbf{w}_{\text{init,i}}, \mathbf{r}_{\text{init,i}}^{(2)} \right\rangle > 0 \right) \right]$$
$$= \mathbb{E}\left[ \mathbf{x}_{\text{init,i}}^{(1)} y_{\text{init,i}}^{(1)} \mathbb{I} \left( \left\langle \mathbf{w}_{\text{init,i}}, \mathbf{x}_{\text{init,i}}^{(1)} \right\rangle > 0 \right) + \mathbf{x}_{\text{init,i}}^{(2)} y_{\text{init,i}}^{(2)} \mathbb{I} \left( \left\langle -\mathbf{w}_{\text{init,i}}, \mathbf{x}_{\text{init,i}}^{(2)} \right\rangle > 0 \right) \right]$$
$$= \mathbb{E}\left[ \mathbf{x}y \mathbb{I} \left( \langle \mathbf{w}_{\text{init,i}}, \mathbf{x} \rangle > 0 \right) \right] + \mathbb{E}\left[ \mathbf{x}y \mathbb{I} \left( \langle \mathbf{w}_{\text{init,i}}, \mathbf{x} \rangle < 0 \right) \right]$$
$$= \mathbb{E}\left[ \mathbf{x}y \right].$$

Thus, we have

$$\mathbb{E}\left[ \bar{\mathbf{w}}_0 \right] = \frac{1}{T_{\text{init}}} \sum_{i=1}^{T_{\text{init}}} \mathbb{E}\left[ \mathbf{r}_{\text{init,i}}^{(1)} y_{\text{init,i}}^{(1)} \mathbb{I} \left( \left\langle \mathbf{w}_{\text{init,i}}, \mathbf{r}_{\text{init,i}}^{(1)} \right\rangle > 0 \right) + \mathbf{r}_{\text{init,i}}^{(2)} y_{\text{init,i}}^{(2)} \mathbb{I} \left( \left\langle -\mathbf{w}_{\text{init,i}}, \mathbf{r}_{\text{init,i}}^{(2)} \right\rangle > 0 \right) \right]$$
$$= \mathbb{E}\left[ \mathbf{x}y \right].$$

$\square$

Now we show that $\bar{\mathbf{w}}_0$ constructed by Algorithm 2 has a positive inner product with the optimal coefficient $\mathbf{w}^*$ with high probability.

**Proposition 2.** *For some constants $c_0, c_1 > 0$, when Algorithm 2 runs for $T_{\text{init}} = c_0 \ln T / (1 - 2\bar{\eta})^2$ iterations, its output $\bar{\mathbf{w}}_0$ satisfies $\langle \mathbf{w}^\star, \bar{\mathbf{w}}_0 \rangle > c_1(1 - 2\bar{\eta}) > 0$ and $\theta\left( \mathbf{w}^\star, \bar{\mathbf{w}}_0 \right) \leq \frac{\pi}{2}$ with probability at least $1 - 2/T^2$.*

*Proof.* First, considering the non-strategic classification problem, we establish a lower bound of $\mathbb{E}[\langle \mathbf{w}^\star, \mathbf{x} \rangle y]$ as the following.

$$\begin{aligned}
\mathbb{E}[\langle \mathbf{w}^\star, \mathbf{x} \rangle y] &= \mathbb{E}\left[ \mathbb{E}[y | \mathbf{x}] \langle \mathbf{w}^\star, \mathbf{x} \rangle \right] \\
&= \mathbb{E}\left[ \left[ (1 - \eta(\mathbf{x}))\text{sgn}(\langle \mathbf{w}^\star, \mathbf{x} \rangle) - \eta(\mathbf{x})\text{sgn}(\langle \mathbf{w}^\star, \mathbf{x} \rangle) \right] \langle \mathbf{w}^\star, \mathbf{x} \rangle \right] \\
&= \mathbb{E}\left[ (1 - 2\eta(\mathbf{x})) | \langle \mathbf{w}^\star, \mathbf{x} \rangle | \right] \\
&\geq (1 - 2\bar{\eta})\mathbb{E}\left[ | \langle \mathbf{w}^\star, \mathbf{x} \rangle | \right] \\
&\geq (1 - 2\bar{\eta})L_1 R^2,
\end{aligned} \tag{8}$$

where the first equality holds by the law of iterated expectations. The second equality holds by the definition of Massart Noise. The third equality holds since $\text{sgn}(\langle \mathbf{w}^*, \mathbf{x} \rangle \langle \mathbf{w}^*, \mathbf{x} \rangle) = | \langle \mathbf{w}^*, \mathbf{x} \rangle |$. The first inequality holds because $\eta(\mathbf{x}) \leq \bar{\eta}, \forall \mathbf{x} \in \mathbb{R}^d$. Now we prove $\mathbb{E}\left[ | \langle \mathbf{w}^\star, \mathbf{x} \rangle | \right] \geq L_1 R^2$ for the last inequality as the following: let $\mathbf{x}_{V_1} := \langle \mathbf{w}^\star, \mathbf{x} \rangle$ denote a 1-dimensional projection of $\mathbf{x}$. Then, by Assumption 2, condition 1, $\phi_{V_1}(\mathbf{x}_{V_1}) \geq L_1$ for all $-R \leq \mathbf{x}_{V_1} \leq R$. Hence, we have

$$\mathbb{E}[| \langle \mathbf{w}^\star, \mathbf{x} \rangle |] \geq 2 \int_0^R \mathbf{x}_{V_1} \phi_{V_1}(\mathbf{x}_{V_1}) \, d\mathbf{x}_{V_1} \geq L_1 \times 2 \int_0^R x dx = L_1 R^2.$$

Second, considering agents' strategic response, we find an unbiased estimator of $\mathbb{E}\left[ \langle \mathbf{w}^\star, \mathbf{x} \rangle y \right]$ through samples. By Lemma 14, $\bar{\mathbf{w}}_0 = \frac{1}{T_{\text{init}}} \sum_{i=1}^{T_{\text{init}}} \mathbf{r}_{\text{init},i}^{(1)} y_{\text{init},i}^{(1)} \mathbb{I} \left( \left\langle \mathbf{w}_{\text{init},i}, \mathbf{r}_{\text{init},i}^{(1)} \right\rangle > 0 \right) + \mathbf{r}_{\text{init},i}^{(2)} y_{\text{init},i}^{(2)} \mathbb{I} \left( \left\langle -\mathbf{w}_{\text{init},i}, \mathbf{r}_{\text{init},i}^{(2)} \right\rangle > 0 \right)$ is an unbiased estimator of $\mathbb{E}[\mathbf{x}y]$. Therefore, $\mathbb{E}\left[ \langle \mathbf{w}^\star, \bar{\mathbf{w}}_0 \rangle \right] = \mathbb{E}\left[ \langle \mathbf{w}^\star, \mathbf{x} \rangle y \right] \geq (1 - 2\bar{\eta})L_1 R^2$.

Finally, using the results of concentration inequalities, we establish the high probability bound of $\langle \mathbf{w}^\star, \bar{\mathbf{w}}_0 \rangle$. By Corollary 1, $\left\langle \mathbf{w}^\star, \mathbf{r}_{\text{init},i}^{(1)} \right\rangle \mathbb{I}\left(\left\langle \mathbf{w}_{\text{init},i}, \mathbf{r}_{\text{init},i}^{(1)} \right\rangle > 0 \right) = \left\langle \mathbf{w}^\star, \mathbf{x}_{\text{init},i}^{(1)} \right\rangle \mathbb{I}\left(\left\langle \mathbf{w}_{\text{init},i}, \mathbf{x}_{\text{init},i}^{(1)} \right\rangle > 0 \right)$ and $\left\langle \mathbf{w}^\star, \mathbf{r}_{\text{init},i}^{(2)} \right\rangle \mathbb{I}\left(\left\langle -\mathbf{w}_{\text{init},i}, \mathbf{r}_{\text{init},i}^{(2)} \right\rangle > 0 \right) = \left\langle \mathbf{w}^\star, \mathbf{x}_{\text{init},i}^{(2)} \right\rangle \mathbb{I}\left(\left\langle -\mathbf{w}_{\text{init},i}, \mathbf{x}_{\text{init},i}^{(2)} \right\rangle > 0 \right)$. Then, by Assumption 2 condition 3 and Lemma 9, we get that both $\left\langle \mathbf{w}^\star, \mathbf{r}_{\text{init},i}^{(1)} \right\rangle y_{\text{init},i}^{(1)} \mathbb{I}\left(\left\langle \mathbf{w}_{\text{init},i}, \mathbf{r}_{\text{init},i}^{(1)} \right\rangle > 0 \right)$ and $\left\langle \mathbf{w}^\star, \mathbf{r}_{\text{init},i}^{(2)} \right\rangle y_{\text{init},i}^{(2)} \mathbb{I}\left(\left\langle -\mathbf{w}_{\text{init},i}, \mathbf{r}_{\text{init},i}^{(2)} \right\rangle > 0 \right)$ are $\left(\frac{16}{Q}, \frac{6}{Q}\right)$-subexponential. Thus, by Lemma 6, we have that with probability at least $1 - \frac{1}{T^2}$,

$$\frac{1}{T_{\text{init}}} \sum_{i=1}^{T_{\text{init}}} \left\langle \mathbf{w}^\star, \mathbf{r}_{\text{init},i}^{(1)} \right\rangle y_{\text{init},i}^{(1)} \mathbb{I}\left(\left\langle \mathbf{w}_{\text{init},i}, \mathbf{r}_{\text{init},i}^{(1)} \right\rangle > 0 \right)$$
$$\geq \frac{1}{T_{\text{init}}} \sum_{i=1}^{T_{\text{init}}} \mathbb{E}\left[\langle \mathbf{w}^\star, \mathbf{x} \rangle y \mathbb{I}(\langle \mathbf{w}_{\text{init},i}, \mathbf{x} \rangle > 0)\right] - \frac{32}{Q}\sqrt{\frac{\ln T}{T_{\text{init}}}} - \frac{24}{Q}\frac{\ln T}{T_{\text{init}}},$$

(9)

and with probability $1 - \frac{1}{T^2}$,

$$\frac{1}{T_{\text{init}}} \sum_{i=1}^{T_{\text{init}}} \left\langle \mathbf{w}^\star, \mathbf{r}_{\text{init},i}^{(2)} \right\rangle y_{\text{init},i}^{(2)} \mathbb{I}\left(\left\langle -\mathbf{w}_{\text{init},i}, \mathbf{r}_{\text{init},i}^{(2)} \right\rangle > 0 \right)$$
$$\geq \frac{1}{T_{\text{init}}} \sum_{i=1}^{T_{\text{init}}} \mathbb{E}\left[\langle \mathbf{w}^\star, \mathbf{x} \rangle y \mathbb{I}(\langle \mathbf{w}_{\text{init},i}, \mathbf{x} \rangle < 0)\right] - \frac{32}{Q}\sqrt{\frac{\ln T}{T_{\text{init}}}} - \frac{24}{Q}\frac{\ln T}{T_{\text{init}}}.$$

(10)

Taking the union bound for (9) and (10), then, with probability at least $1 - \frac{2}{T^2}$,

$$\langle \mathbf{w}^\star, \bar{\mathbf{w}}_0 \rangle = \frac{1}{T_{\text{init}}} \sum_{i=1}^{T_{\text{init}}} \left\langle \mathbf{w}^\star, \mathbf{r}_{\text{init},i}^{(1)} \right\rangle y_{\text{init},i}^{(1)} \mathbb{I}\left(\left\langle \mathbf{w}_{\text{init},i}, \mathbf{r}_{\text{init},i}^{(1)} \right\rangle > 0 \right)$$
$$+ \frac{1}{T_{\text{init}}} \sum_{i=1}^{T_{\text{init}}} \left\langle \mathbf{w}^\star, \mathbf{r}_{\text{init},i}^{(2)} \right\rangle y_{\text{init},i}^{(2)} \mathbb{I}\left(\left\langle -\mathbf{w}_{\text{init},i}, \mathbf{r}_{\text{init},i}^{(2)} \right\rangle > 0 \right)$$
$$\geq \frac{1}{T_{\text{init}}} \sum_{i=1}^{T_{\text{init}}} \mathbb{E}\left[\langle \mathbf{w}^\star, \mathbf{x} \rangle y \mathbb{I}(\langle \mathbf{w}_{\text{init},i}, \mathbf{x} \rangle > 0)\right] - \frac{32}{Q}\sqrt{\frac{\ln T}{T_{\text{init}}}} - \frac{24}{Q}\frac{\ln T}{T_{\text{init}}}$$
$$+ \frac{1}{T_{\text{init}}} \sum_{i=1}^{T_{\text{init}}} \mathbb{E}\left[\langle \mathbf{w}^\star, \mathbf{x} \rangle y \mathbb{I}(\langle \mathbf{w}_{\text{init},i}, \mathbf{x} \rangle < 0)\right] - \frac{32}{Q}\sqrt{\frac{\ln T}{T_{\text{init}}}} - \frac{24}{Q}\frac{\ln T}{T_{\text{init}}}$$
$$= \mathbb{E}\left[\langle \mathbf{w}^\star, \mathbf{x} \rangle y\right] - \frac{64}{Q}\sqrt{\frac{\ln T}{T_{\text{init}}}} - \frac{48}{Q}\frac{\ln T}{T_{\text{init}}}$$
$$\geq (1 - 2\bar{\eta})L_1 R^2 - \frac{64}{Q}\sqrt{\frac{\ln T}{T_{\text{init}}}} - \frac{48}{Q}\frac{\ln T}{T_{\text{init}}}$$
$$\geq \frac{1}{2}(1 - 2\bar{\eta})L_1 R^2.$$

The first inequality holds by (9) and (10). The second inequality holds by Lemma 14. The last inequality holds by setting $T_{\text{init}} = \left\lceil \frac{20736 \ln T}{(1-2\bar{\eta})^2 L_1^2 Q^2 R^4} \right\rceil$, then $\frac{64}{Q}\sqrt{\frac{\ln T}{T_{\text{init}}}} \leq \frac{4}{9}(1 - 2\bar{\eta})L_1 R^2$ and $\frac{48}{Q}\frac{\ln T}{T_{\text{init}}} \ll \frac{1}{18}(1 - 2\bar{\eta})L_1 R^2$ for large enough $T$. Thus, we complete the proof of Proposition 2. $\square$

**Theoretical Guarantees of Algorithm 3** Recall that in the $i$'th iteration of Algorithm 3, we declare classifier $\tilde{h}_{0,i}(\mathbf{r}) = \text{sgn}(\langle \mathbf{w}_{0,i}, \mathbf{r} \rangle)$ and construct the gradient as $\tilde{\mathbf{g}}_{0,i} = [-\bar{\eta}\mathbf{r}_{0,i}\mathbb{I}(y_{0,i} = 1) + (1 - \bar{\eta})\mathbf{r}_{0,i}\mathbb{I}(y_{0,i} = -1)]\mathbb{I}(0 < \langle \mathbf{w}_{0,i}, \mathbf{r}_{0,i} \rangle \leq b_0)$. In Lemma 15, we establish the high probability upperbound of $\sum_{i=1}^{T_0} \mathbb{E}[\langle \mathbf{w}^\star, -\tilde{\mathbf{g}}_{0,i} \rangle)| \mathcal{F}_{0,i-1}]$ by $\sum_{i=1}^{T_0} \langle \mathbf{w}^\star, -\tilde{\mathbf{g}}_{0,i} \rangle$.

**Lemma 15.** *In Algorithm 3, with probability at least $1 - 1/T^2$, we have*

$$\sum_{i=1}^{T_0} \mathbb{E}[\langle \mathbf{w}^\star, -\tilde{\mathbf{g}}_{0,i} \rangle) | \mathcal{F}_{0,i-1}] \leq \sum_{i=1}^{T_0} \langle \mathbf{w}^\star, -\tilde{\mathbf{g}}_{0,i} \rangle$$
$$+ \frac{24}{\delta} \sqrt{1 + 2\rho b_0} \sqrt{T_0 \ln T} + \frac{48}{\delta} \ln T,$$

*where $\rho = \max \left\{ U_1 \exp(\delta), \frac{U_2 \exp(\delta)}{\delta} \right\}$.*

*Proof.* Since $\langle \mathbf{w}^\star, -\tilde{\mathbf{g}}_{0,i} \rangle = \langle \mathbf{w}^\star, \mathbf{x}_{0,i} \rangle \mathbb{I}(0 < \langle \mathbf{w}_{0,i}, \mathbf{x}_{0,i} \rangle \leq b_0) \left( \frac{1}{2} y_{0,i} - \left( \frac{1}{2} - \bar{\eta} \right) \right)$, we first establish the probability tail bound of $\langle \mathbf{w}^\star, \mathbf{x}_{0,i} \rangle \mathbb{I}(0 < \langle \mathbf{w}_{0,i}, \mathbf{x}_{0,i} \rangle \leq b_0)$.

We partition $\mathbf{x}_{0,i}$ into two orthonormal vectors, for notational convenience, we omit the subindex $0, i$ of $\mathbf{x}_{0,i}$ and let $\mathbf{x}_{\|\mathbf{w}}$ denote the ingredient of $\mathbf{x}_{0,i}$ that is parallel to $\mathbf{w}_{0,i}$, *i.e.*, $\mathbf{x}_{\|\mathbf{w}} = \langle \mathbf{w}_{0,i}, \mathbf{x}_{0,i} \rangle \mathbf{w}_{0,i}$ and $\mathbf{x}_{\perp \mathbf{w}}$ denote the ingredient of $\mathbf{x}_{0,i}$ that is vertical to $\mathbf{w}_{0,i}$, *i.e.*, $\mathbf{x}_{\perp w} = \mathbf{x}_{0,i} - \mathbf{x}_{\|w}$. Then,

$$\langle \mathbf{w}^\star, \mathbf{x}_{0,i} \rangle \mathbb{I}(0 \leq \langle \mathbf{w}_{0,i}, \mathbf{x}_{0,i} \rangle \leq b_0)$$
$$= \underbrace{\langle \mathbf{w}^\star, \mathbf{x}_{\|\mathbf{w}} \rangle \mathbb{I}(0 \leq \langle \mathbf{w}_{0,i}, \mathbf{x}_{0,i} \rangle \leq b_0)}_{(a)} + \underbrace{\langle \mathbf{w}^\star, \mathbf{x}_{\perp \mathbf{w}} \rangle \mathbb{I}(0 \leq \langle \mathbf{w}_{0,i}, \mathbf{x}_{0,i} \rangle \leq b_0)}_{(b)}. \quad (11)$$

Thus, we have to bound part (a) and (b) in (11), respectively. First, we bound part (a) as

$$\langle \mathbf{w}^\star, \mathbf{x}_{\|\mathbf{w}} \rangle \mathbb{I}(0 \leq \langle \mathbf{w}_{0,i}, \mathbf{x}_{0,i} \rangle \leq b_0) = \langle \mathbf{w}^\star, \langle \mathbf{w}_{0,i}, \mathbf{x}_{0,i} \rangle \mathbf{w}_{0,i} \rangle \mathbb{I}(0 \leq \langle \mathbf{w}_{0,i}, \mathbf{x}_{0,i} \rangle \leq b_0)$$
$$\leq b_0 \langle \mathbf{w}^\star, \mathbf{w}_{0,i} \rangle \quad (12)$$
$$\leq b_0,$$

where the first equality holds since $\mathbf{x}_{\|\mathbf{w}} = \langle \mathbf{w}_{0,i}, \mathbf{x}_{0,i} \rangle \mathbf{w}_{0,i}$. The last inequality holds by the Cauchy-Schwarz Inequality. Next, we bound part (b) in (11). For $\|\mathbf{w}_{0,i} - \mathbf{w}^\star\|_2 \leq r_0$ and $a > b_0$, we have

$$\mathbb{P}(| \langle \mathbf{w}^\star, \mathbf{x}_{\perp \mathbf{w}} \rangle \mathbb{I}(0 \leq \langle \mathbf{w}_{0,i}, \mathbf{x}_{0,i} \rangle \leq b_0)| \geq a - b_0)$$
$$= \mathbb{P}(| \langle \mathbf{w}^\star, \mathbf{x}_{\perp \mathbf{w}} \rangle | \geq a - b_0, \ 0 \leq \langle \mathbf{w}_{0,i}, \mathbf{x}_{0,i} \rangle \leq b_0)$$
$$= \mathbb{P}(| \langle \mathbf{w}^\star_{\perp \mathbf{w}}, \mathbf{x}_{0,i} \rangle | \geq a - b_0, \ 0 \leq \langle \mathbf{w}_{0,i}, \mathbf{x}_{0,i} \rangle \leq b_0) ,$$

where we prove the second equality as follows

$$\langle \mathbf{w}^\star, \mathbf{x}_{\perp \mathbf{w}} \rangle = \langle \mathbf{w}^\star_{\perp \mathbf{w}}, \mathbf{x}_{\perp \mathbf{w}} \rangle + \left\langle \mathbf{w}^\star_{\|\mathbf{w}}, \mathbf{x}_{\perp \mathbf{w}} \right\rangle,$$

$$\langle \mathbf{w}^\star_{\perp \mathbf{w}}, \mathbf{x}_{0,i} \rangle = \langle \mathbf{w}^\star_{\perp \mathbf{w}}, \mathbf{x}_{\perp \mathbf{w}} \rangle + \langle \mathbf{w}^\star_{\perp \mathbf{w}}, \mathbf{x}_{\|\mathbf{w}} \rangle.$$

since $\left\langle \mathbf{w}^\star_{\|\mathbf{w}}, \mathbf{x}_{\perp \mathbf{w}} \right\rangle = \langle \mathbf{w}^\star_{\perp \mathbf{w}}, \mathbf{x}_{\|\mathbf{w}} \rangle = 0$, we have $\langle \mathbf{w}^\star, \mathbf{x}_{\perp \mathbf{w}} \rangle = \langle \mathbf{w}^\star_{\perp \mathbf{w}}, \mathbf{x}_{0,i} \rangle$.

Denote $X := \left\langle \frac{\mathbf{w}^\star_{\perp \mathbf{w}}}{\|\mathbf{w}^\star_{\perp \mathbf{w}}\|_2}, \mathbf{x}_{0,i} \right\rangle$ and $Y := \langle \mathbf{w}_{0,i}, \mathbf{x}_{0,i} \rangle$. Then, $(X, Y)$ forms a projection of $\mathbf{x}_{0,i}$ onto a 2-dimensional subspace $V_2$ spanned by $\frac{\mathbf{w}^\star_{\perp \mathbf{w}}}{\|\mathbf{w}^\star_{\perp \mathbf{w}}\|_2}$ and $\mathbf{w}_{0,i}$. Let $\phi_{V_2}$ denote the density of $(X, Y)$. By Assumption 2 condition 2, we have $\phi_{V_2}(X, Y) \leq U_2 \exp(-\delta \|(X, Y)\|_2) =$

$U_2 \exp(-\delta\sqrt{X^2 + Y^2})$. Hence, we can bound the above probability by

$$\mathbb{P}\left(|\langle \mathbf{w}^\star, \mathbf{x}_{\perp \mathbf{w}} \rangle \mathbb{I}(0 \leq \langle \mathbf{w}_{0,i}, \mathbf{x}_{0,i} \rangle \leq b_0)| \geq a - b_0\right)$$

$$= \mathbb{P}\left(\left|\left\langle \frac{\mathbf{w}^\star_{\perp \mathbf{w}}}{\|\mathbf{w}^\star_{\perp \mathbf{w}}\|_2}, \mathbf{x}_{0,i} \right\rangle\right| \geq \frac{a - b_0}{\|\mathbf{w}^\star_{\perp \mathbf{w}}\|_2}, \ 0 \leq \langle \mathbf{w}_{0,i}, \mathbf{x}_{0,i} \rangle \leq b_0\right)$$

$$= \int_{\frac{a-b_0}{\|\mathbf{w}^\star_{\perp \mathbf{w}}\|_2}}^{+\infty} \int_0^{b_0} \phi_{V_2}(X, Y) dX dY$$

$$\leq U_2 \int_{\frac{a-b_0}{\|\mathbf{w}^\star_{\perp \mathbf{w}}\|_2}}^{+\infty} \int_0^{b_0} \exp\left(-\delta\sqrt{X^2 + Y^2}\right) dX dY \tag{13}$$

$$\leq U_2 b_0 \int_{\frac{a-b_0}{\|\mathbf{w}^\star_{\perp \mathbf{w}}\|_2}}^{+\infty} \exp(-\delta X) dX dY$$

$$= \frac{U_2}{\delta} b_0 \exp\left(-\delta \frac{a - b_0}{\|\mathbf{w}^\star_{\perp \mathbf{w}}\|_2}\right)$$

$$\leq \frac{U_2 \exp(\delta)}{\delta} b_0 \exp\left(-\delta \frac{a}{r_0}\right).$$

The last inequality holds by the fact that $\|\mathbf{w}^\star - \mathbf{w}_{0,i}\|_2 \leq r_0$, which implies $\|\mathbf{w}^\star_{\perp \mathbf{w}}\|_2 \leq r_0$, and that $b_0 < r_0 = 2$, which implies $\exp\left(\delta \frac{b_0}{r_0}\right) < \exp(\delta)$. Combing (12) and (13), for $a > b_0$,

$$\mathbb{P}\left(\langle \mathbf{w}^\star, \mathbf{x}_{0,i} \rangle \mathbb{I}(0 \leq \langle \mathbf{w}_{0,i}, \mathbf{x}_{0,i} \rangle \leq b_0) > a\right) \leq \mathbb{P}\left(|\langle \mathbf{w}^\star, \mathbf{x}_{\perp \mathbf{w}} \rangle \mathbb{I}(0 \leq \langle \mathbf{w}_{0,i}, \mathbf{x}_{0,i} \rangle \leq b_0)| \geq a - b_0\right)$$

$$\leq \frac{U_2 \exp(\delta)}{\delta} b_0 \exp\left(-\delta \frac{a}{r_0}\right). \tag{14}$$

For $0 < a \leq b_0$,

$$\mathbb{P}\left(\langle \mathbf{w}^\star, \mathbf{x}_{0,i} \rangle \mathbb{I}(0 \leq \langle \mathbf{w}_{0,i}, \mathbf{x}_{0,i} \rangle \leq b_0) > a\right) \leq \mathbb{P}\left(|\langle \mathbf{w}^\star, \mathbf{x}_{0,i} \rangle \mathbb{I}(0 \leq \langle \mathbf{w}_{0,i}, \mathbf{x}_{0,i} \rangle \leq b_0)| > 0\right)$$

$$\leq \mathbb{P}\left(0 \leq \langle \mathbf{w}_{0,i}, \mathbf{x}_{0,i} \rangle \leq b_0\right)$$

$$\leq U_1 b_0$$

$$\leq U_1 b_0 \exp\left(\delta \frac{b_0}{r_0}\right) \exp\left(-\delta \frac{a}{r_0}\right) \tag{15}$$

$$\leq U_1 \exp(\delta) b_0 \exp\left(-\delta \frac{a}{r_0}\right).$$

The third inequality holds by Lemma 4 property (a). The fourth inequality holds for $0 < a < b_0$. The last inequality holds since by our construction, $b_0 < r_0 = 2$.

Let $\rho = \max\left\{U_1 \exp(\delta), \frac{U_2 \exp(\delta)}{\delta}\right\}$, by (14) and (15), we conclude that for $\forall a > 0$,

$$\mathbb{P}\left(\mathbb{I}(0 \leq \langle \mathbf{w}_{0,i}, \mathbf{x} \rangle \leq b_0) \langle \mathbf{w}^\star, \mathbf{x} \rangle > a\right) \leq \rho b_0 \exp\left(-\delta \frac{a}{r_0}\right). \tag{16}$$

Thus,

$$\mathbb{P}\left(|\langle \mathbf{w}^\star, -\tilde{\mathbf{g}}_{0,i} \rangle| \geq a\right)$$

$$= \mathbb{P}\left(\left|\langle \mathbf{w}^\star, \mathbf{x}_{0,i} \rangle \mathbb{I}(0 \leq \langle \mathbf{w}_{0,i}, \mathbf{x}_{0,i} \rangle \leq b_0)\left(\frac{1}{2} y_{0,i} - \left(\frac{1}{2} - \bar{\eta}\right)\right)\right| \geq a\right)$$

$$\leq \mathbb{P}\left(|\langle \mathbf{w}^\star, \mathbf{x}_{0,i} \rangle \mathbb{I}(0 \leq \langle \mathbf{w}_{0,i}, \mathbf{x}_{0,i} \rangle \leq b_0)| \geq a\right)$$

$$\leq \rho b_0 \exp\left(-\delta \frac{a}{2}\right),$$

where the first inequality holds since $\left|\frac{1}{2} y_{0,i} - (\frac{1}{2} - \bar{\eta})\right| \leq 1$. In the last inequality, since $\|\mathbf{w}_{0,i} - \mathbf{w}^\star\|_2 \leq \|\mathbf{w}_{0,i}\|_2 + \|\mathbf{w}^\star\|_2 = 2$, we take $r_0 = 2$ in (16) and get the upper bound.

By Lemma 9, $\langle \mathbf{w}^\star, -\tilde{\mathbf{g}}_{0,i} \rangle$ is $\left( \frac{12}{\delta} \sqrt{1 + 2\rho b_0}, \frac{12}{\delta} \right)$-subexponential. By Lemma 7, we can get that with probability at least $(1 - 1/T^2)$,

$$\sum_{i=1}^{T_0} \mathbb{E}[\langle \mathbf{w}^\star, -\tilde{\mathbf{g}}_{0,i} \rangle) | \mathcal{F}_{0,i-1}] \leq \sum_{i=1}^{T_0} \langle \mathbf{w}^\star, -\tilde{\mathbf{g}}_{0,i} \rangle + \frac{24}{\delta} \sqrt{1 + 2\rho b_0} \sqrt{T_0 \ln T} + \frac{48}{\delta} \ln T.$$

$\square$

The following lemma establishes a high-probability upper bound for the average of $f_{0,i}(\mathbf{w}_{0,i})$ over $T_0$ iterations.

**Lemma 16.** *In Algorithm 3, if $\langle \mathbf{w}^\star, \bar{\mathbf{w}}_0 \rangle \geq \frac{1}{2}(1 - 2\eta) L_1 R^2$, then there exist some constants $c_2, c_3, c_4 > 0$, when setting bandwidth $b_0 = c_2(1 - 2\bar{\eta})^2$, iteration number $T_0 = c_3 \frac{1}{(1-2\bar{\eta})^8} d \ln d (\ln T)^2$, step size $\alpha_0 = c_4 \frac{\sqrt{d \ln d}}{\sqrt{T_0} \ln T}$, then with probability at least $1 - 3/T^2$, we have*

$$\frac{1}{T_0} \sum_{i=1}^{T_0} f_{0,i}(\mathbf{w}_{0,i}) \leq \min \left\{ 1, L_1 R^2 \right\} \frac{\pi(1 - 2\bar{\eta}) L_2 R^2}{2880 U_1}.$$

*Proof.* Recall that in the non-strategic setting, $\mathbf{g}_{k,i} = [-\bar{\eta} \mathbf{x}_{0,i} \mathbb{I}(y_{0,i} = 1) + (1 - \bar{\eta}) \mathbf{x}_{0,i} \mathbb{I}(y_{0,i} = -1)] \mathbb{I}(\mathbf{x}_{0,i} \in D_{0,i})$, where $D_{0,i} = \{\mathbf{x} \mid 0 \leq \langle \mathbf{w}_{0,i}, \mathbf{x} \rangle \leq b_0\}$ is just the localization region. Since the announced classifier is $\tilde{h}_{0,i}(\mathbf{r}) = \text{sgn}(\langle \mathbf{w}_{0,i}, \mathbf{r} \rangle)$, by the construction of $\tilde{\mathbf{g}}_{0,i}$ in Algorithm 3 and Corollary 4, we have

$$\tilde{\mathbf{g}}_{0,i} = [-\bar{\eta} \mathbf{r}_{0,i} \mathbb{I}(y_{0,i} = 1) + (1 - \bar{\eta}) \mathbf{r}_{0,i} \mathbb{I}(y_{0,i} = -1)] \mathbb{I}(0 < \langle \mathbf{w}_{0,i}, \mathbf{r}_{0,i} \rangle \leq b_0)$$
$$= [-\bar{\eta} \mathbf{x}_{0,i} \mathbb{I}(y_{0,i} = 1) + (1 - \bar{\eta}) \mathbf{x}_{0,i} \mathbb{I}(y_{0,i} = -1)] \mathbb{I}(\mathbf{x}_{0,i} \in D_{0,i})$$
$$= \mathbf{g}_{0,i}.$$

Then, by Lemma 10, we have

$$\mathbb{E}[\langle \mathbf{w}^\star, -\tilde{\mathbf{g}}_{0,i} \rangle] = \mathbb{E}[\langle \mathbf{w}^\star, -\mathbf{g}_{0,i} \rangle] \geq (1 - 2\bar{\eta}) f_{0,i}(\mathbf{w}_{0,i}) \mathbb{P}(\mathbf{x} \in D_{0,i}). \tag{17}$$

We proceed to establish the high probability bound of $\sum_{i=1}^{T_0} \mathbb{E}[\langle \mathbf{w}^\star, -\tilde{\mathbf{g}}_{0,i} \rangle]$. By Lemma 15, with probability at least $1 - 1/T^2$, we have

$$\sum_{i=1}^{T_0} \mathbb{E}[\langle \mathbf{w}^\star, -\tilde{\mathbf{g}}_{0,i} \rangle) | \mathcal{F}_{0,i-1}] \leq \sum_{i=1}^{T_0} \langle \mathbf{w}^\star, -\tilde{\mathbf{g}}_{0,i} \rangle + \frac{24}{\delta} \sqrt{1 + 2\rho b_0} \sqrt{T_0 \ln T} + \frac{48}{\delta} \ln T, \tag{18}$$

where $\rho = \max \left\{ U_1 \exp(\delta), \frac{U_2 \exp(\delta)}{\delta} \right\}$.

Next, we move on to upper bound $\sum_{i=1}^{T_0} \langle \mathbf{w}^\star, -\tilde{\mathbf{g}}_{0,i} \rangle$ through a nonstandard regret analysis of online mirror decent.

Let $B(\mathbf{v}_1, \mathbf{v}_2) := \frac{1}{2(p-1)} \|\mathbf{v}_1 - \mathbf{v}_2\|_p^2$ denote the Bregman divergence w.r.t. $\frac{1}{2(p-1)} \| \cdot \|_p^2$, where $p = \frac{\ln(8d)}{\ln(8d)-1}$. In each iteration $i$, the regularizer, $B(\cdot, \mathbf{w}_{0,i-1})$ is 1-strongly convex with respect to $\| \cdot \|_p$ [see Shalev-Shwartz [2007]]. From the analysis of online mirror descent [see Orabona [2023], Lemma 6.9], with step size $\alpha_0$, we have

$$\langle \alpha_0 \hat{\mathbf{g}}_{0,i}, \mathbf{w}_{0,i} - \mathbf{w}^\star \rangle \leq B(\mathbf{w}^\star, \mathbf{w}_{0,i}) - B(\mathbf{w}^\star, \mathbf{w}_{0,i+1}) + \frac{\alpha_0^2}{2} \|\tilde{\mathbf{g}}_{0,i}\|_q^2,$$

where $q = \ln(8d) > 2$. Summing the above equality over $i \in [T_0]$, we get

$$\sum_{i=1}^{T_0} \langle \alpha_0 \tilde{\mathbf{g}}_{0,i}, \mathbf{w}_{0,i} - \mathbf{w}^\star \rangle \leq B(\mathbf{w}^\star, \mathbf{w}_{0,1}) - B(\mathbf{w}^\star, \mathbf{w}_{0,T_0+1}) + \frac{\alpha_0^2}{2} \sum_{i=1}^{T_0} \|\tilde{\mathbf{g}}_{0,i}\|_q^2.$$

Dividing both sides by $\alpha_0$, and moving $\sum_{i=1}^{T_0} \langle \mathbf{w}_{0,i}, \tilde{\mathbf{g}}_{0,i} \rangle$ to RHS, we get

$$\sum_{i=1}^{T_0} \langle \mathbf{w}^\star, -\tilde{\mathbf{g}}_{0,i} \rangle \leq \frac{1}{\alpha_0} [B(\mathbf{w}^\star, \mathbf{w}_{0,1}) - B(\mathbf{w}^\star, \mathbf{w}_{0,T_0+1})] + \sum_{i=1}^{T_0} \langle \mathbf{w}_{0,i}, -\tilde{\mathbf{g}}_{0,i} \rangle + \frac{\alpha_0}{2} \sum_{i=1}^{T_0} \|\tilde{\mathbf{g}}_{0,i}\|_q^2$$

$$\leq \frac{1}{\alpha_0} B(\mathbf{w}^\star, \mathbf{w}_{0,1}) + \sum_{i=1}^{T_0} \langle \mathbf{w}_{0,i}, -\tilde{\mathbf{g}}_{0,i} \rangle + \frac{\alpha_0}{2} \sum_{i=1}^{T_0} \|\tilde{\mathbf{g}}_{0,i}\|_q^2.$$

$$\tag{19}$$

Now we need to bound the three terms in the RHS of (19) respectively.

First, we bound $B(\mathbf{w}^\star, \mathbf{w}_{0,1}) = B(\mathbf{w}^\star, \mathbf{w}_0)$

$$B(\mathbf{w}^\star, \mathbf{w}_{0,1}) = \frac{\|\mathbf{w}^\star - \mathbf{w}_0\|_p^2}{2(p-1)} \overset{(a)}{\leq} \frac{\|\mathbf{w}^\star - \mathbf{w}_0\|_1^2}{2(p-1)} \overset{(b)}{\leq} \frac{d\|\mathbf{w}^\star - \mathbf{w}_0\|_2^2}{2(p-1)} \overset{(c)}{\leq} 2d\ln(8d), \qquad (20)$$

where inequality (a) holds by the fact that $\|\mathbf{x}\|_p \leq \|\mathbf{x}\|_1$ for all $p > 1$ and $\mathbf{x} \in \mathbb{R}^d$. Inequality (b) holds since $\|\mathbf{x}\|_1 \leq \sqrt{d}\|\mathbf{x}\|_2$ for all $\mathbf{x} \in \mathbb{R}^d$. Inequality (c) holds since $\|\mathbf{w}^\star - \mathbf{w}_0\|_2 \leq \|\mathbf{w}^\star\|_2 + \|\mathbf{w}_0\|_2 = 2$ and $\frac{1}{p-1} \leq \ln(8d) - 1 < \ln(8d)$.

Next, we bound $\sum_{i=1}^{T_0} \langle -\mathbf{w}_{0,i}, \tilde{\mathbf{g}}_{0,i} \rangle$.
Since $\langle -\mathbf{w}_{0,i}, \tilde{\mathbf{g}}_{0,i} \rangle = \langle \mathbf{w}_{0,i}, \mathbf{x}_{0,i} \rangle \left( \frac{1}{2} y_{0,i} - \left( \frac{1}{2} - \eta \right) \right) \mathbb{I} \left( 0 < \langle \mathbf{w}_{0,i}, \mathbf{x}_{0,i} \rangle \leq b_0 \right)$, then $|\langle -\mathbf{w}_{0,i}, \tilde{\mathbf{g}}_{0,i} \rangle| \leq b_0$ and $\mathbb{E}\left[ \langle -\mathbf{w}_{0,i}, \tilde{\mathbf{g}}_{0,i} \rangle \right] \leq b_0 \mathbb{P}(\mathbf{x} \in D_{0,i})$, by Lemma 8, with probability at least $1 - 1/T^2$, we have

$$\sum_{i=1}^{T_0} \langle -\mathbf{w}_{0,i}, \tilde{\mathbf{g}}_{0,i} \rangle \leq \sum_{i=1}^{T_0} \mathbb{E}\left[ \langle -\mathbf{w}_{0,i}, \tilde{\mathbf{g}}_{0,i} \rangle \mid \mathcal{F}_{0,i-1} \right] + b_0 \sqrt{T_0 \ln T}$$

$$\leq b_0 \sum_{i=1}^{T_0} \Pr(\mathbf{x} \in D_{0,i}) + b_0 \sqrt{T_0 \ln T}.$$

(21)

Finally, we bound $\sum_{i=1}^{T_0} \|\tilde{\mathbf{g}}_{0,i}\|_q^2$. Since $\|\mathbf{g}_{0,i}\|_q \leq 2\|\mathbf{g}_{0,i}\|_\infty$, we only need to upper bound $\sum_{i=1}^{T_0} \|\mathbf{g}_{0,i}\|_\infty^2$, which satisfies

$$\|\mathbf{g}_{0,i}\|_\infty = \left\| \mathbb{I}(0 \leq \langle \mathbf{w}_{0,i}, \mathbf{x}_{0,i} \rangle \leq b_0) \left( -\frac{1}{2} y_{0,i} + \left( \frac{1}{2} - \eta \right) \right) \mathbf{x}_{0,i} \right\|_\infty \leq \|\mathbf{x}_{0,i}\|_\infty.$$

By Lemma 13, we have with probability at least $1 - 1/T_0 T^2$,

$$\|\mathbf{x}_{0,i}\|_\infty \leq \frac{1}{Q} \left( 1 + \ln(dT_0 T^2) \right) \leq \frac{3}{Q} \ln T.$$

Thus, taking the union bound over $i \in [T_0]$, we have with probability at least $1 - 1/T^2$,

$$\sum_{i=1}^{T_0} \|\mathbf{g}_{0,i}\|_\infty^2 \leq T_0 \times \left( \frac{3}{Q} \ln T \right)^2 = \frac{9}{Q^2} T_0 (\ln T)^2.$$

Hence, with probability at least $1 - 1/T^2$,

$$\sum_{i=1}^{T_0} \|\tilde{\mathbf{g}}_{0,i}\|_q^2 \leq 4 \sum_{i=1}^{T_0} \|\mathbf{g}_{0,i}\|_\infty^2 \leq \frac{36}{Q^2} T_0 (\ln T)^2. \qquad (22)$$

Combining (18), (20), (21) and (22) together, and taking the union bound, we get with probability at least $1 - 3/T^2$,

$$(1 - 2\eta) \sum_{i=1}^{T_0} f_{0,i}(\mathbf{w}_{0,i}) \mathbb{P}(\mathbf{x} \in D_{0,i})$$

$$\leq \sum_{i=1}^{T_0} \mathbb{E}\left[ \langle \mathbf{w}^\star, -\tilde{\mathbf{g}}_{0,i} \rangle \right]$$

$$\leq \sum_{i=1}^{T_0} \langle \mathbf{w}^\star, -\tilde{\mathbf{g}}_{0,i} \rangle + \frac{24}{\delta} \sqrt{1 + 2\rho b_0} \sqrt{T_0 \ln T} + \frac{48}{\delta} \ln T \qquad (23)$$

$$\leq \frac{2d \ln(8d)}{\alpha_0} + b_0 \sum_{i=1}^{T_0} \Pr(\mathbf{x} \in D_{0,i}) + b_0 \sqrt{T_0 \ln T} + \frac{18\alpha_0}{Q^2} T_0 (\ln T)^2$$

$$+ \frac{24}{\delta} \sqrt{1 + 2\rho b_0} \sqrt{T_0 \ln T} + \frac{48}{\delta} \ln T,$$

where $\rho = \max\left\{ U_1 \exp(\delta), \frac{U_2 \exp(\delta)}{\delta} \right\}$. By Lemma 4, property (a), and the fact that $b_0 < R$, we have $L_1 b_0 \leq \mathbb{P}(\mathbf{x} \in D_{0,i}) \leq U_1 b_0$, dividing both sides of (23) by $(1 - 2\bar{\eta})L_1 b_0 T_0$, we get

$$\frac{1}{T_0} \sum_{i=1}^{T_0} f_{0,i}(\mathbf{w}_{0,i})$$

$$\leq \frac{1}{\alpha_0} \frac{2d \ln(8d)}{(1 - 2\bar{\eta})L_1 b_0 T_0} + \frac{U_1}{(1 - 2\bar{\eta})L_1} b_0 + \frac{1}{(1 - 2\bar{\eta})L_1} \sqrt{\frac{\ln T}{T_0}} + \frac{18\alpha_0}{(1 - 2\bar{\eta})Q^2 L_1 b_0} (\ln T)^2$$

$$+ \frac{24}{(1 - 2\bar{\eta})\delta L_1 b_0} \sqrt{1 + 2\rho b_0} \sqrt{\frac{\ln T}{T_0}} + \frac{48}{(1 - 2\bar{\eta})\delta L_1 b_0} \frac{\ln T}{T_0}.$$

By our setting, $b_0 = \min\left\{1, L_1 R^2\right\} \frac{(1 - 2\bar{\eta})^2 L_1 L_2 R^2}{2880 U_1^2} = c_2(1 - 2\bar{\eta})^2$, $T_0 = \frac{576(1 + 2\rho b_0)d \ln(8d)(\ln T)^2}{U_1^2 \delta^2 b_0^4} = c_3 \frac{1}{(1 - 2\bar{\eta})^8} d \ln d (\ln T)^2$, $\alpha_0 = \frac{Q\sqrt{d \ln(8d)}}{3\sqrt{T_0} \ln T} = c_4 \frac{\sqrt{d \ln d}}{\sqrt{T_0} \ln T}$, then we finish the proof. $\qquad \square$

The following lemma established by Zhang et al. [2020] indicates that by the construction of the constraint set $\mathcal{W}_0$, any two vectors in $\mathcal{W}_0$ form an angle that is no bigger than $\pi - \frac{1}{2}(1 - 2\bar{\eta})L_1 R^2$.

**Lemma 17.** *(Zhang et al. [2020], Lemma 19) For any two vectors* $\mathbf{u}, \mathbf{v} \in \mathcal{W}_0 = \left\{\mathbf{w} \mid \|w\|_2 \leq 1, \langle \mathbf{w}, \bar{\mathbf{w}}_0 \rangle \geq \frac{1}{2}(1 - 2\bar{\eta})L_1 R^2\right\}$, *we have* $\theta(\mathbf{u}, \mathbf{v}) \leq \pi - \frac{1}{2}(1 - 2\bar{\eta})L_1 R^2$.

We use the following corollary to show that, in the $i$'th iteration, a small value of $f_{0,i}(\mathbf{w}_{0,i})$ indicates that $\mathbf{w}_{0,i}$ and $\mathbf{w}^*$ are close.

**Corollary 5.** *If* $\mathbf{w}^* \in \mathcal{W}_0$ *and* $f_{0,i}(\mathbf{w}_{0,i}) < \min\left\{1, L_1 R^2\right\} \frac{\pi(1 - 2\bar{\eta})L_2 R^2}{320 U_1}$, *then* $\theta(\mathbf{w}^\star, \mathbf{w}_{0,i}) \leq \frac{\pi}{10}$.

*Proof.* We first exclude the case that $\theta(\mathbf{w}^\star, \mathbf{w}_{0,i}) > \frac{\pi}{2}$, which we prove by contradiction. Suppose $\theta(\mathbf{w}^\star, \mathbf{w}_{0,i}) > \frac{\pi}{2}$. By Lemma 17 and our choice of $b_0$, $\theta(\mathbf{w}^\star, \mathbf{w}_{0,i}) \leq \pi - \frac{1}{2}(1 - 2\eta)L_1 R^2 < \pi - b_0$. From Lemma 11, we get $f_{k,i}(\mathbf{w}_{k,i}) \geq \frac{L_2}{32 U_1} R^2 (\pi - \theta(\mathbf{w}^\star, \mathbf{w}_{k,i}))$. Together with the condition that $f_{0,i}(\mathbf{w}_{0,i}) < \frac{\pi(1 - 2\bar{\eta})L_1 L_2 R^4}{320 U_1}$, we have $\theta(\mathbf{w}^\star, \mathbf{w}_{0,i}) > \pi - \frac{\pi}{10}(1 - 2\bar{\eta})L_1 R^2 > \pi - \frac{1}{2}(1 - 2\bar{\eta})L_1 R^2$, which is a contradiction. Thus, we conclude that $\theta(\mathbf{w}^\star, \mathbf{w}_{0,i}) \leq \frac{\pi}{2}$.

Next, since $f_{0,i}(\mathbf{w}_{0,i}) < \frac{\pi(1 - 2\bar{\eta})L_2 R^2}{320 U_1} \leq \frac{L_2 R^2}{160 U_1} \cdot \frac{\pi}{2}$, by Corollary 3, setting $\theta_0 = \frac{\pi}{2}$, then $\theta(\mathbf{w}^\star, \mathbf{w}_{0,i}) \leq \frac{\theta_0}{5} = \frac{\pi}{10}$. $\qquad \square$

Putting all pieces together, now we are able to show the main theoretical guarantee of Algorithm 3.

**Proposition 3.** *For the constant* $c_1$ *in Proposition 2 and some constants* $c_2, c_3, c_4 > 0$*, when the initial vector* $\bar{\mathbf{w}}_0$ *satisfies* $\langle \mathbf{w}^\star, \bar{\mathbf{w}}_0 \rangle \geq c_1(1 - 2\bar{\eta})$ *and Algorithm 3 runs with bandwidth* $b_0 = c_2(1 - 2\bar{\eta})^2$ *for* $T_0 = c_3 d \ln d (\ln T)^2/(1 - 2\bar{\eta})^8$ *iterations with step size* $\alpha_0 = c_4 \sqrt{d \ln(d)}/(\sqrt{T_0} \ln T)$*, then its output* $\mathbf{w}_1$ *satisfies* $\theta(\mathbf{w}^\star, \mathbf{w}_1) \leq \pi/4$ *with probability at least* $1 - 3/T^2$.

*Proof.* In Algorithm 3, the constraint set we choose for gradient update is $\mathcal{W}_0 = \{\mathbf{w} \mid \|\mathbf{w}\|_2 \leq 1, \langle \mathbf{w}, \bar{\mathbf{w}}_0 \rangle \geq c_1(1 - 2\bar{\eta})\}$, where $c_1 = \frac{1}{2} L_1 R^2$, since $\langle \mathbf{w}^\star, \bar{\mathbf{w}}_0 \rangle \geq c_1(1 - 2\bar{\eta})$, we can conclude that $\mathbf{w}^\star \in \mathcal{W}_0$.

Next, Lemma 16 shows that with probability at least $1 - 3/T^2$,

$$\frac{1}{T_0} \sum_{i=1}^{T_0} f_{0,i}(\mathbf{w}_{0,i}) \leq \min\left\{1, L_1 R^2\right\} \frac{\pi(1 - 2\bar{\eta})L_2 R^2}{2880 U_1}. \tag{24}$$

Let $A_0$ denote the set $\left\{i \mid f_{0,i}(\mathbf{w}_{0,i}) > \min\left\{1, L_1 R^2\right\} \frac{\pi(1 - 2\bar{\eta})L_2 R^2}{320 U_1}\right\}$. Combing (24), we have:

$$\min\left\{1, L_1 R^2\right\} \frac{\pi(1 - 2\bar{\eta})L_2 R^2}{2880 U_1} \geq \frac{1}{T_0} \sum_{i=1}^{T_0} f_{0,i}(\mathbf{w}_{0,i}) \geq \frac{|A_0|}{T_0} \min\left\{1, L_1 R^2\right\} \frac{\pi(1 - 2\bar{\eta})L_2 R^2}{320 U_1}.$$

Solve the above inequality and we get $\frac{|A_0|}{T_0} \leq \frac{1}{9}$.

From Corollary 3, set $\theta_0 = \frac{\pi}{2}$, we know that when $i \in \bar{A}_0$, $\theta(\mathbf{w}^\star, \mathbf{w}_{0,i}) \leq \frac{\pi}{10}$. Thus,

$$
\begin{aligned}
\frac{1}{T_0} \sum_{i=1}^{T_0} \cos(\theta\left(\mathbf{w}^\star, \mathbf{w}_{0,i}\right)) &\geq \frac{|\bar{A}_0|}{T_0} \cos\left(\frac{\pi}{10}\right) - \frac{|A_0|}{T_0} \\
&\geq \left(1 - \frac{|A_0|}{T_0}\right)\left(1 - \frac{1}{2}\left(\frac{\pi}{10}\right)^2\right) - \frac{|A_0|}{T_0} \\
&\geq \left(1 - \frac{1}{9}\right)\left(1 - \frac{1}{2}\left(\frac{\pi}{10}\right)^2\right) - \frac{1}{9} \\
&\geq \cos\left(\frac{\pi}{4}\right),
\end{aligned}
$$

where the first inequality holds since $\cos x$ is decreasing in $x \in [0, \pi]$ and $\cos x \geq -1$. The second inequality holds since $\cos x \geq 1 - \frac{1}{2}x^2$ for all $x \in [0, \pi]$. The last inequality holds since $\left(1 - \frac{1}{9}\right)\left(1 - \frac{1}{2}\left(\frac{\pi}{10}\right)^2\right) - \frac{1}{9} \approx 0.73 > 0.71 \approx \cos\left(\frac{\pi}{4}\right)$. By the concavity of $\cos(\theta\left(\mathbf{w}^\star, \cdot\right))$, using Jensen's inequality, we conclude that when the above inequality holds, we have

$$
\cos\left(\theta\left(\mathbf{w}^\star, \frac{1}{T_0}\sum_{i=1}^{T_0}\mathbf{w}_{0,i}\right)\right) \geq \frac{1}{T_0}\sum_{i=1}^{T_0}\cos(\theta\left(\mathbf{w}^\star, \mathbf{w}_{0,i}\right)) \geq \cos\left(\frac{\pi}{4}\right).
$$

Thus, we can get that with probability at least $1 - 3/T^2$, Algorithm 3 returns a vector $\mathbf{w}_1$ such that $\theta(\mathbf{w}^\star, \mathbf{w}_1) \leq \frac{\pi}{4}$. $\qquad\square$

**Theoretical Guarantees of Algorithm 4** The following lemma shows that if, in batch $k$, iteration $i$, we can identify agents whose true features lie in the localization region $D_{k,i}$, and we use proxy features to construct a proxy gradient $\tilde{\mathbf{g}}_{k,i}$, then $\mathbb{E}\left[\langle\mathbf{w}^\star, -\tilde{\mathbf{g}}_{k,i}\rangle\right]$ upper bounds $\mathbb{E}\left[\langle\mathbf{w}^\star, -\mathbf{g}_{k,i}\rangle\right]$ (and hence upper bounds $\theta\left(\mathbf{w}^\star, \mathbf{w}_{k,i}\right)$).

**Lemma 18.** *Given a classification rule $\tilde{h}(\mathbf{r}) = sgn(\langle\mathbf{w}_{k,i}, \mathbf{r}\rangle + m_{k,i})$ with fixed $\mathbf{w}_{k,i} \in \mathbb{S}^d$ and arbitrary $m_{k,i} < 0$, an agent $(\mathbf{x}_{k,i}, y_{k,i})$ reports his feature as $\mathbf{r}_{k,i}$ according to Lemma 1. Construct proxy data as*

$$
\tilde{\mathbf{x}}_{k,i}^+ := \left(\mathbf{r}_{k,i} + (b_k - \langle\mathbf{w}_{k,i}, \mathbf{r}_{k,i}\rangle)\mathbf{w}_{k,i}\right)\mathbb{I}\left(y_{k,i} = 1, \mathbf{x}_{k,i} \in D_{k,i}\right),
$$
$$
\tilde{\mathbf{x}}_{k,i}^- := \left(\mathbf{r}_{k,i} - \langle\mathbf{w}_{k,i}, \mathbf{r}_{k,i}\rangle \mathbf{w}_{k,i}\right)\mathbb{I}\left(y_{k,i} = -1, \mathbf{x}_{k,i} \in D_{k,i}\right),
$$

*and define the gradient as*

$$
\tilde{\mathbf{g}}_{k,i} := \left[-\bar{\eta}\tilde{\mathbf{x}}_{k,i}^+ + (1 - \bar{\eta})\tilde{\mathbf{x}}_{k,i}^-\right]\mathbb{I}\left(\mathbf{x}_{k,i} \in D_{k,i}\right).
$$

*Then, we have*

$$
\mathbb{E}\left[\langle\mathbf{w}^\star, -\tilde{\mathbf{g}}_{k,i}\rangle\right] \geq \mathbb{E}\left[\langle\mathbf{w}^\star, -\mathbf{g}_{k,i}\rangle\right].
$$

*Proof.* First, by Lemma 1, given $\mathbf{w}_{k,i}, m_{k,i}$, for $\forall\, \mathbf{x}_{k,i} \in \mathbb{R}^d$, we have

$$
\mathbf{r}_{k,i} + (b_k - \langle\mathbf{w}_{k,i}, \mathbf{r}_{k,i}\rangle)\mathbf{w}_{k,i} = \mathbf{x}_{k,i} + (b_k - \langle\mathbf{w}_{k,i}, \mathbf{x}_{k,i}\rangle)\mathbf{w}_{k,i},
$$

and

$$
\mathbf{r}_{k,i} - \langle\mathbf{w}_{k,i}, \mathbf{r}_{k,i}\rangle \mathbf{w}_{k,i} = \mathbf{x}_{k,i} - \langle\mathbf{w}_{k,i}, \mathbf{x}_{k,i}\rangle \mathbf{w}_{k,i}.
$$

Therefore, when $\mathbf{x}_{k,i} \in D_{k,i} = \{\mathbf{x}|\ 0 \leq \langle\mathbf{w}_{k,i}, \mathbf{x}\rangle \leq b_k\}$, we have the following

$$
\begin{aligned}
\left\langle\mathbf{w}^\star, \tilde{\mathbf{x}}_{k,i}^+\right\rangle &= \langle\mathbf{w}^\star, \mathbf{r}_{k,i} + (b_k - \langle\mathbf{w}_{k,i}, \mathbf{r}_{k,i}\rangle)\mathbf{w}_{k,i}\rangle\,\mathbb{I}\left(y_{k,i} = 1,\ \mathbf{x}_{k,i} \in D_{k,i}\right) \\
&= \langle\mathbf{w}^\star, \mathbf{x}_{k,i} + (b_k - \langle\mathbf{w}_{k,i}, \mathbf{x}_{k,i}\rangle)\mathbf{w}_{k,i}\rangle\,\mathbb{I}\left(y_{k,i} = 1,\ \mathbf{x}_{k,i} \in D_{k,i}\right) \\
&= \left[\langle\mathbf{w}^\star, \mathbf{x}_{k,i}\rangle + (b_k - \langle\mathbf{w}_{k,i}, \mathbf{x}_{k,i}\rangle)\langle\mathbf{w}^\star, \mathbf{w}_{k,i}\rangle\right]\mathbb{I}\left(y_{k,i} = 1,\ \mathbf{x}_{k,i} \in D_{k,i}\right) \\
&\geq \langle\mathbf{w}^\star, \mathbf{x}_{k,i}\rangle\,\mathbb{I}\left(y_{k,i} = 1,\ \mathbf{x}_{k,i} \in D_{k,i}\right),
\end{aligned}
\tag{25}
$$

where the inequality holds since $\langle\mathbf{w}^\star, \mathbf{w}_{k,i}\rangle > 0$ and $\langle\mathbf{w}_{k,i}, \mathbf{x}_{k,i}\rangle\,\mathbb{I}\left(\mathbf{x}_{k,i} \in D_{k,i}\right) \leq b_k$.

Similarly,

$$
\begin{aligned}
\left\langle \mathbf{w}^\star, \tilde{\mathbf{x}}_{k,i}^- \right\rangle &= \langle \mathbf{w}^\star, \mathbf{r}_{k,i} - \langle \mathbf{w}_{k,i}, \mathbf{r}_{k,i} \rangle \, \mathbf{w}_{k,i} \rangle \, \mathbb{I} \left( y_{k,i} = -1, \, \mathbf{x}_{k,i} \in D_{k,i} \right) \\
&= \langle \mathbf{w}^\star, \mathbf{x}_{k,i} - \langle \mathbf{w}_{k,i}, \mathbf{x}_{k,i} \rangle \, \mathbf{w}_{k,i} \rangle \, \mathbb{I} \left( y_{k,i} = -1, \, \mathbf{x}_{k,i} \in D_{k,i} \right) \\
&= \left[ \langle \mathbf{w}^\star, \mathbf{x}_{k,i} \rangle - \langle \mathbf{w}_{k,i}, \mathbf{x}_{k,i} \rangle \langle \mathbf{w}^\star, \mathbf{w}_{k,i} \rangle \right] \mathbb{I} \left( y_{k,i} = -1, \, \mathbf{x}_{k,i} \in D_{k,i} \right) \\
&\le \langle \mathbf{w}^\star, \mathbf{x}_{k,i} \rangle \, \mathbb{I} \left( y_{k,i} = -1, \, \mathbf{x}_{k,i} \in D_{k,i} \right).
\end{aligned}
\tag{26}
$$

where the inequality holds since $\langle \mathbf{w}^\star, \mathbf{w}_{k,i} \rangle > 0$ and $\langle \mathbf{w}_{k,i}, \mathbf{x}_{k,i} \rangle \mathbb{I} \left( \mathbf{x}_{k,i} \in D_{k,i} \right) \ge 0$. Combing (25) and (26), we have

$$
\begin{aligned}
&\mathbb{E} \left[ \langle \mathbf{w}^\star, -\tilde{\mathbf{g}}_{k,i} \rangle \right] \\
&= \bar{\eta} \mathbb{E} \left[ \left\langle \mathbf{w}^\star, \tilde{\mathbf{x}}_{k,i}^+ \right\rangle \right] - (1 - \bar{\eta}) \mathbb{E} \left[ \left\langle \mathbf{w}^\star, \tilde{\mathbf{x}}_{k,i}^- \right\rangle \right] \\
&\ge \bar{\eta} \mathbb{E} \left[ \langle \mathbf{w}^\star, \mathbf{x}_{k,i} \rangle \, \mathbb{I} \left( y_{k,i} = 1, \, \mathbf{x}_{k,i} \in D_{k,i} \right) \right] - (1 - \bar{\eta}) \mathbb{E} \left[ \langle \mathbf{w}^\star, \mathbf{x}_{k,i} \rangle \, \mathbb{I} \left( y_{k,i} = -1, \, \mathbf{x}_{k,i} \in D_{k,i} \right) \right] \\
&= \mathbb{E} \left[ \langle \mathbf{w}^\star, -\mathbf{g}_{k,i} \rangle \right].
\end{aligned}
$$

$\square$

The following lemma shows that by pairwise comparing agents' responses under two different declared classifiers designed in Algorithm 4, we can unbiasedly estimate the proxy data desired by Lemma 18 and hence construct a gradient estimator $\hat{\mathbf{g}}_{k,i}$, accordingly.

**Lemma 19.** *Given $\tilde{\mathbf{x}}_{k,i}^+$, $\tilde{\mathbf{x}}_{k,i}^-$ and $\tilde{\mathbf{g}}_{k,i}$ defined in Lemma 18, for the proxy data $\hat{\mathbf{x}}_{k,i}^{(1,+)}$, $\hat{\mathbf{x}}_{k,i}^{(1,-)}$, $\hat{\mathbf{x}}_{k,i}^{(2,+)}$, $\hat{\mathbf{x}}_{k,i}^{(2,-)}$ and gradient estimator $\hat{\mathbf{g}}_{k,i}$ defined in Algorithm 4, we have*

$$
\mathbb{E} \left[ \hat{\mathbf{x}}_{k,i}^{(1,+)} - \hat{\mathbf{x}}_{k,i}^{(2,+)} \right] = \mathbb{E} \left[ \tilde{\mathbf{x}}_{k,i}^+ \right],
$$

*and*

$$
\mathbb{E} \left[ \hat{\mathbf{x}}_{k,i}^{(1,-)} - \hat{\mathbf{x}}_{k,i}^{(2,-)} \right] = \mathbb{E} \left[ \tilde{\mathbf{x}}_{k,i}^- \right].
$$

*Moreover,*

$$
\mathbb{E} \left[ \langle \mathbf{w}^\star, -\hat{\mathbf{g}}_{k,i} \rangle \right] = \mathbb{E} \left[ \langle \mathbf{w}^\star, -\tilde{\mathbf{g}}_{k,i} \rangle \right].
$$

*Proof.* For fixed normal vector $\mathbf{w}_{k,i}$ and bandwidth $b_k$, recall that $D_{k,i}^{(1)} = \{ \mathbf{x} \, | \, 0 \le \langle \mathbf{w}_{k,i}, \mathbf{x} \rangle \le \gamma + b_k \}$ and $D_{k,i}^{(2)} = \{ \mathbf{x} \, | \, b_k \le \langle \mathbf{w}_{k,i}, \mathbf{x} \rangle \le \gamma + b_k \}$. Then, we can verify that $D_{k,i} = D_{k,i}^{(1)} / D_{k,i}^{(2)}$. By Corollary 4, we have

$$
\begin{aligned}
\tilde{\mathbf{x}}_{k,i}^+ &= [\mathbf{r}_{k,i} + (b_k - \langle \mathbf{w}_{k,i}, \mathbf{r}_{k,i} \rangle) \mathbf{w}_{k,i}] \mathbb{I}(y_{k,i} = 1, \, \mathbf{x}_{k,i} \in D_{k,i}) \\
&= [\mathbf{x}_{k,i} + (b_k - \langle \mathbf{w}_{k,i}, \mathbf{x}_{k,i} \rangle) \mathbf{w}_{k,i}] \mathbb{I}(y_{k,i} = 1, \, \mathbf{x}_{k,i} \in D_{k,i}), \\
\tilde{\mathbf{x}}_{k,i}^- &= [\mathbf{r}_{k,i} - \langle \mathbf{w}_{k,i}, \mathbf{r}_{k,i} \rangle \, \mathbf{w}_{k,i}] \mathbb{I}(y_{k,i} = -1, \, \mathbf{x}_{k,i} \in D_{k,i}) \\
&= [\mathbf{x}_{k,i} - \langle \mathbf{w}_{k,i}, \mathbf{x}_{k,i} \rangle \, \mathbf{w}_{k,i}] \mathbb{I}(y_{k,i} = -1, \, \mathbf{x}_{k,i} \in D_{k,i}), \\
\hat{\mathbf{x}}_{k,i}^{(1,+)} &= \left[ \mathbf{r}_{k,i}^{(1)} + \left( b_k - \left\langle \mathbf{w}_{k,i}, \mathbf{r}_{k,i}^{(1)} \right\rangle \right) \mathbf{w}_{k,i} \right] \mathbb{I} \left( y_{k,i}^{(1)} = 1, \, \gamma \le \left\langle \mathbf{w}_{k,i}, \mathbf{r}_{k,i}^{(1)} \right\rangle \le \gamma + b_k \right) \\
&= \left[ \mathbf{x}_{k,i}^{(1)} + \left( b_k - \left\langle \mathbf{w}_{k,i}, \mathbf{x}_{k,i}^{(1)} \right\rangle \right) \mathbf{w}_{k,i} \right] \mathbb{I} \left( y_{k,i}^{(1)} = 1, \, \mathbf{x}_{k,i}^{(1)} \in D_{k,i}^{(1)} \right), \\
\hat{\mathbf{x}}_{k,i}^{(1,-)} &= \left[ \mathbf{r}_{k,i}^{(1)} - \left\langle \mathbf{w}_{k,i}, \mathbf{r}_{k,i}^{(1)} \right\rangle \mathbf{w}_{k,i} \right] \mathbb{I} \left( y_{k,i}^{(1)} = -1, \, \gamma \le \left\langle \mathbf{w}_{k,i}, \mathbf{r}_{k,i}^{(1)} \right\rangle \le \gamma + b_k \right) \\
&= \left[ \mathbf{x}_{k,i}^{(1)} - \left\langle \mathbf{w}_{k,i}, \mathbf{x}_{k,i}^{(1)} \right\rangle \mathbf{w}_{k,i} \right] \mathbb{I} \left( y_{k,i}^{(1)} = -1, \, \mathbf{x}_{k,i}^{(1)} \in D_{k,i}^{(1)} \right), \\
\hat{\mathbf{x}}_{k,i}^{(2,+)} &= \left[ \mathbf{r}_{k,i}^{(2)} + \left( b_k - \left\langle \mathbf{w}_{k,i}, \mathbf{r}_{k,i}^{(2)} \right\rangle \right) \mathbf{w}_{k,i} \right] \mathbb{I} \left( y_{k,i}^{(2)} = 1, \, \left\langle \mathbf{w}_{k,i}, \mathbf{r}_{k,i}^{(2)} \right\rangle = \gamma + b_k \right) \\
&= \left[ \mathbf{x}_{k,i}^{(2)} + \left( b_k - \left\langle \mathbf{w}_{k,i}, \mathbf{x}_{k,i}^{(2)} \right\rangle \right) \mathbf{w}_{k,i} \right] \mathbb{I} \left( y_{k,i}^{(2)} = 1, \, \mathbf{x}_{k,i}^{(2)} \in D_{k,i}^{(2)} \right), \\
\hat{\mathbf{x}}_{k,i}^{(2,-)} &= \left[ \mathbf{r}_{k,i}^{(2)} - \left\langle \mathbf{w}_{k,i}, \mathbf{r}_{k,i}^{(2)} \right\rangle \mathbf{w}_{k,i} \right] \mathbb{I} \left( y_{k,i}^{(2)} = -1, \, \left\langle \mathbf{w}_{k,i}, \mathbf{r}_{k,i}^{(2)} \right\rangle = \gamma + b_k \right) \\
&= \left[ \mathbf{x}_{k,i}^{(2)} - \left\langle \mathbf{w}_{k,i}, \mathbf{x}_{k,i}^{(2)} \right\rangle \mathbf{w}_{k,i} \right] \mathbb{I} \left( y_{k,i}^{(2)} = -1, \, \mathbf{x}_{k,i}^{(2)} \in D_{k,i}^{(2)} \right).
\end{aligned}
$$

Also, $(\mathbf{x}_{k,i}^{(1)}, y_{k,i}^{(1)})$ and $(\mathbf{x}_{k,i}^{(2)}, y_{k,i}^{(2)})$ are drawn i.i.d. from $\mathcal{D}$, thus,

$$
\begin{aligned}
\mathbb{E}\left[\hat{\mathbf{x}}_{k,i}^{(1,+)} - \hat{\mathbf{x}}_{k,i}^{(2,+)}\right] &= \mathbb{E}\left[\left[\mathbf{x}_{k,i}^{(1)} + \left(b_k - \left\langle\mathbf{w}_{k,i}, \mathbf{x}_{k,i}^{(1)}\right\rangle\right)\mathbf{w}_{k,i}\right]\mathbb{I}\left(y_{k,i}^{(1)} = 1,\ \mathbf{x}_{k,i}^{(1)} \in D_{k,i}^{(1)}\right)\right] \\
&\quad - \mathbb{E}\left[\left[\mathbf{x}_{k,i}^{(2)} + \left(b_k - \left\langle\mathbf{w}_{k,i}, \mathbf{x}_{k,i}^{(2)}\right\rangle\right)\mathbf{w}_{k,i}\right]\mathbb{I}\left(y_{k,i}^{(2)} = 1,\ \mathbf{x}_{k,i}^{(2)} \in D_{k,i}^{(2)}\right)\right] \\
&= \mathbb{E}\left[[\mathbf{x} + (b_k - \langle\mathbf{w}_{k,i}, \mathbf{x}\rangle)\mathbf{w}_{k,i}]\mathbb{I}\left(y = 1,\ \mathbf{x} \in D_{k,i}^{(1)}\right)\right] \\
&\quad - \mathbb{E}\left[[\mathbf{x} + (b_k - \langle\mathbf{w}_{k,i}, \mathbf{x}\rangle)\mathbf{w}_{k,i}]\mathbb{I}\left(y = 1,\ \mathbf{x} \in D_{k,i}^{(2)}\right)\right] \\
&= \mathbb{E}[[\mathbf{x} + (b_k - \langle\mathbf{w}_{k,i}, \mathbf{x}\rangle)\mathbf{w}_{k,i}]\mathbb{I}(y = 1,\ \mathbf{x} \in D_{k,i})] \\
&= \mathbb{E}\left[\tilde{\mathbf{x}}_{k,i}^{+}\right].
\end{aligned}
$$

Similarly, we can show that $\mathbb{E}\left[\hat{\mathbf{x}}_{k,i}^{(1,-)} - \hat{\mathbf{x}}_{k,i}^{(2,-)}\right] = \mathbb{E}\left[\tilde{\mathbf{x}}_{k,i}^{-}\right]$. Therefore,

$$
\begin{aligned}
\mathbb{E}[\langle\mathbf{w}^{\star}, -\hat{\mathbf{g}}_{k,i}\rangle] &= \bar{\eta}\mathbb{E}\left[\left\langle\mathbf{w}^{\star}, \hat{\mathbf{x}}_{k,i}^{(1,+)} - \hat{\mathbf{x}}_{k,i}^{(2,+)}\right\rangle\right] - (1-\bar{\eta})\mathbb{E}\left[\left\langle\mathbf{w}^{\star}, \hat{\mathbf{x}}_{k,i}^{(1,-)} - \hat{\mathbf{x}}_{k,i}^{(2,-)}\right\rangle\right] \\
&= \bar{\eta}\mathbb{E}\left[\left\langle\mathbf{w}^{\star}, \tilde{\mathbf{x}}_{k,i}^{(+)}\right\rangle\right] - (1-\bar{\eta})\mathbb{E}\left[\left\langle\mathbf{w}^{\star}, \tilde{\mathbf{x}}_{k,i}^{(-)}\right\rangle\right] \\
&= \mathbb{E}[\langle\mathbf{w}^{\star}, -\tilde{\mathbf{g}}_{k,i}\rangle].
\end{aligned}
$$

$\square$

Next, we establish the high probability bound of $\sum_{i=1}^{T_k}\mathbb{E}[\langle\mathbf{w}^{\star}, -\hat{\mathbf{g}}_{k,i}\rangle \mid \mathcal{F}_{k,i-1}]$ by $\sum_{i=1}^{T_k}\langle\mathbf{w}^{\star}, -\hat{\mathbf{g}}_{k,i}\rangle$.

**Lemma 20.** *At batch $k$ of Algorithm 4, when $\|\mathbf{w}^{\star} - \mathbf{w}_{k,i}\|_2 \leq r_k$ for $\forall i \in [T_k]$, then with probability at least $1 - 2/T^2$, we have*

$$
\sum_{i=1}^{T_k}\mathbb{E}[\langle\mathbf{w}^{\star}, -\hat{\mathbf{g}}_{k,i}\rangle \mid \mathcal{F}_{k,i-1}] \leq \sum_{i=1}^{T_k}\langle\mathbf{w}^{\star}, -\hat{\mathbf{g}}_{k,i}\rangle
$$
$$
+ \frac{24}{\delta}\sqrt{1 + 2\rho(\gamma + b_k)}r_k\sqrt{T_k \ln T} + \frac{48}{\delta}r_k \ln T,
$$

*where $\rho = \max\left\{U_1\exp(\delta), \frac{U_2\exp(\delta)}{\delta}\right\}$.*

*Proof.* First, we partition $\langle\mathbf{w}^{\star}, -\hat{\mathbf{g}}_{k,i}\rangle$ into two parts, as the following

$$
\begin{aligned}
&\langle\mathbf{w}^{\star}, -\hat{\mathbf{g}}_{k,i}\rangle \\
&= \left\langle\mathbf{w}^{\star}, \bar{\eta}(\hat{\mathbf{x}}_{k,i}^{(1,+)} - \hat{\mathbf{x}}_{k,i}^{(2,+)}) - (1-\bar{\eta})(\hat{\mathbf{x}}_{k,i}^{(1,-)} - \hat{\mathbf{x}}_{k,i}^{(2,-)})\right\rangle \\
&= \underbrace{\left\langle\mathbf{w}^{\star}, \bar{\eta}\hat{\mathbf{x}}_{k,i}^{(1,+)} - (1-\bar{\eta})\hat{\mathbf{x}}_{k,i}^{(1,-)}\right\rangle}_{(a)} - \underbrace{\left\langle\mathbf{w}^{\star}, \bar{\eta}\hat{\mathbf{x}}_{k,i}^{(2,+)} - (1-\bar{\eta})\hat{\mathbf{x}}_{k,i}^{(2,-)}\right\rangle}_{(b)}.
\end{aligned}
\tag{27}
$$

Since the randomness in part (a) arises from one sample, while the randomness in part (b) arises from another independent sample, parts (a) and (b) are independent. Therefore, we can bound them separately. We first discuss the high-probability tail bound of part (a).

We partition $\bar{\eta}\hat{\mathbf{x}}_{k,i}^{(1,+)} - (1-\bar{\eta})\hat{\mathbf{x}}_{k,i}^{(1,-)}$ into two orthonormal vectors. For notational convenience, we omit the subindex $k, i$ and let $\mathbf{x}_{\|\mathbf{w}}$ denote the component of $\mathbf{x}$ that is parallel to $\mathbf{w}_{k,i}$, *i.e.*, $\mathbf{x}_{\|\mathbf{w}} = \langle\mathbf{w}_{k,i}, \mathbf{x}\rangle\mathbf{w}_{k,i}$, and $\mathbf{x}_{\perp\mathbf{w}}$ denote the component of $\mathbf{x}$ that is orthogonal to $\mathbf{w}_{k,i}$, *i.e.*, $\mathbf{x}_{\perp\mathbf{w}} = \mathbf{x} - \mathbf{x}_{\|\mathbf{w}}$. Then,

$$
\begin{aligned}
&\left\langle\mathbf{w}^{\star}, \bar{\eta}\hat{\mathbf{x}}_{k,i}^{(1,+)} - (1-\bar{\eta})\hat{\mathbf{x}}_{k,i}^{(1,-)}\right\rangle \\
&= \underbrace{\left\langle\mathbf{w}^{\star}, \left(\bar{\eta}\hat{\mathbf{x}}_{k,i}^{(1,+)} - (1-\bar{\eta})\hat{\mathbf{x}}_{k,i}^{(1,-)}\right)_{\|\mathbf{w}}\right\rangle}_{(a1)} + \underbrace{\left\langle\mathbf{w}^{\star}, \left(\bar{\eta}\hat{\mathbf{x}}_{k,i}^{(1,+)} - (1-\bar{\eta})\hat{\mathbf{x}}_{k,i}^{(1,-)}\right)_{\perp\mathbf{w}}\right\rangle}_{(a2)}.
\end{aligned}
\tag{28}
$$

Hence, we have to bound part (a1) and (a2) in (28), respectively.

To bound part (a1) in (28), we have

$$
\begin{aligned}
\left\langle \mathbf{w}^\star, \hat{\mathbf{x}}_{\|\mathbf{w}}^{(1,+)} \right\rangle &= \left\langle \mathbf{w}^\star, \mathbf{r}_{\|\mathbf{w}}^{(1)} + \left( b_k - \left\langle \mathbf{w}_{k,i}, \mathbf{r}_{k,i}^{(1)} \right\rangle \right) \mathbf{w}_{k,i} \right\rangle \mathbb{I}\left( y_{k,i}^{(1)} = 1, \gamma \le \left\langle \mathbf{w}_{k,i}, \mathbf{r}_{k,i}^{(1)} \right\rangle \le \gamma + b_k \right) \\
&= \left\langle \mathbf{w}^\star, \mathbf{r}_{\|\mathbf{w}}^{(1)} + \left( b_k \mathbf{w}_{k,i} - \mathbf{r}_{\|\mathbf{w}}^{(1)} \right) \right\rangle \mathbb{I}\left( y_{k,i}^{(1)} = 1, \gamma \le \left\langle \mathbf{w}_{k,i}, \mathbf{r}_{k,i}^{(1)} \right\rangle \le \gamma + b_k \right) \\
&= \langle \mathbf{w}^\star, b_k \mathbf{w}_{k,i} \rangle \, \mathbb{I}\left( y_{k,i}^{(1)} = 1, \gamma \le \left\langle \mathbf{w}_{k,i}, \mathbf{r}_{k,i}^{(1)} \right\rangle \le \gamma + b_k \right) \\
&\le b_k,
\end{aligned}
\tag{29}
$$

where the last inequality holds because $\langle \mathbf{w}^\star, \mathbf{w}_{k,i} \rangle \le \|\mathbf{w}^\star\|_2 \|\mathbf{w}_{k,i}\|_2 \le 1$. Similarly,

$$
\begin{aligned}
\left\langle \mathbf{w}^\star, \hat{\mathbf{x}}_{\|\mathbf{w}}^{(1,-)} \right\rangle &= \left\langle \mathbf{w}^\star, \mathbf{r}_{\|\mathbf{w}}^{(1)} - \left\langle \mathbf{w}_{k,i}, \mathbf{r}_{k,i}^{(1)} \right\rangle \mathbf{w}_{k,i} \right\rangle \mathbb{I}\left( y_{k,i}^{(1)} = -1, \left\langle \mathbf{w}_{k,i}, \mathbf{r}_{k,i}^{(1)} \right\rangle = \gamma + b_k \right) \\
&= \left\langle \mathbf{w}^\star, \mathbf{r}_{\|\mathbf{w}}^{(1)} - \mathbf{r}_{\|\mathbf{w}}^{(1)} \right\rangle \mathbb{I}\left( y_{k,i}^{(1)} = -1, \left\langle \mathbf{w}_{k,i}, \mathbf{r}_{k,i}^{(1)} \right\rangle = \gamma + b_k \right) \\
&= 0.
\end{aligned}
\tag{30}
$$

Combing (29) and (30), we can bound part (a1) as

$$
\left\langle \mathbf{w}^\star, \left( \bar{\eta} \hat{\mathbf{x}}_{k,i}^{(1,+)} - (1 - \bar{\eta}) \hat{\mathbf{x}}_{k,i}^{(1,-)} \right)_{\|\mathbf{w}} \right\rangle \le b_k.
\tag{31}
$$

Next, we bound part (a2) in (28). From Lemma 1, we get that

$$
\mathbf{r}_{\perp \mathbf{w}}^{(1)} = \left( \mathbf{r}_{k,i}^{(1)} + \left( b_k - \left\langle \mathbf{w}_{k,i}, \mathbf{r}_{k,i}^{(1)} \right\rangle \right) \mathbf{w}_{k,i} \right)_{\perp \mathbf{w}} = \left( \mathbf{r}_{k,i}^{(1)} - \left\langle \mathbf{w}_{k,i}, \mathbf{r}_{k,i}^{(1)} \right\rangle \mathbf{w}_{k,i} \right)_{\perp \mathbf{w}} = \mathbf{x}_{\perp \mathbf{w}}^{(1)},
$$

hence,

$$
\begin{aligned}
&\underbrace{\left\langle \mathbf{w}^\star, \left( \bar{\eta} \hat{\mathbf{x}}_{k,i}^{(1,+)} - (1 - \bar{\eta}) \hat{\mathbf{x}}_{k,i}^{(1,-)} \right)_{\perp \mathbf{w}} \right\rangle}_{(a2)} \\
&= \bar{\eta} \left\langle \mathbf{w}^\star, \left( \mathbf{r}_{k,i}^{(1)} + \left( b_k - \left\langle \mathbf{w}_{k,i}, \mathbf{r}_{k,i}^{(1)} \right\rangle \right) \mathbf{w}_{k,i} \right)_{\perp \mathbf{w}} \right\rangle \mathbb{I}\left( y_{k,i}^{(1)} = 1, \; \gamma \le \left\langle \mathbf{w}_{k,i}, \mathbf{r}_{k,i}^{(1)} \right\rangle \le \gamma + b_k \right) \\
&\quad - (1 - \bar{\eta}) \left\langle \mathbf{w}^\star, \left( \mathbf{r}_{k,i}^{(1)} - \left\langle \mathbf{w}_{k,i}, \mathbf{r}_{k,i}^{(1)} \right\rangle \mathbf{w}_{k,i} \right)_{\perp \mathbf{w}} \right\rangle \mathbb{I}\left( y_{k,i}^{(1)} = -1, \; \gamma \le \left\langle \mathbf{w}_{k,i}, \mathbf{r}_{k,i}^{(1)} \right\rangle \le \gamma + b_k \right) \\
&= \bar{\eta} \left\langle \mathbf{w}^\star, \mathbf{x}_{\perp \mathbf{w}}^{(1)} \right\rangle \mathbb{I}\left( y_{k,i}^{(1)} = 1, \; \mathbf{x}_{k,i}^{(1)} \in D_{k,i}^{(1)} \right) - (1 - \bar{\eta}) \left\langle \mathbf{w}^\star, \mathbf{x}_{\perp \mathbf{w}}^{(1)} \right\rangle \mathbb{I}\left( y_{k,i}^{(1)} = -1, \; \mathbf{x}_{k,i}^{(1)} \in D_{k,i}^{(1)} \right) \\
&= \left\langle \mathbf{w}^\star, \mathbf{x}_{\perp \mathbf{w}}^{(1)} \right\rangle \left( \frac{1}{2} y_{k,i}^{(1)} - \left( \frac{1}{2} - \bar{\eta} \right) \right) \mathbb{I}\left( \mathbf{x}_{k,i}^{(1)} \in D_{k,i}^{(1)} \right),
\end{aligned}
$$

where the second equality holds by Corollary 4.

Since $\left| \frac{1}{2} y_{k,i}^{(1)} - \left( \frac{1}{2} - \bar{\eta} \right) \right| \le 1$, we only need to establish the high probability bound of $\left\langle \mathbf{w}^\star, \mathbf{x}_{\perp \mathbf{w}}^{(1)} \right\rangle \mathbb{I}\left( \mathbf{x}_{k,i}^{(1)} \in D_{k,i}^{(1)} \right)$, for $a > b_k$ we have

$$
\begin{aligned}
&\mathbb{P}\left( \left| \left\langle \mathbf{w}^\star, \mathbf{x}_{\perp \mathbf{w}}^{(1)} \right\rangle \mathbb{I}\left( \mathbf{x}_{k,i}^{(1)} \in D_{k,i}^{(1)} \right) \right| \ge a - b_k \right) \\
&= \mathbb{P}\left( \left| \left\langle \mathbf{w}^\star, \mathbf{x}_{\perp \mathbf{w}}^{(1)} \right\rangle \right| \ge a - b_k, \; 0 \le \left\langle \mathbf{w}_{k,i}, \mathbf{x}_{k,i}^{(1)} \right\rangle \le \gamma + b_k \right) \\
&= \mathbb{P}\left( \left| \left\langle \mathbf{w}_{\perp \mathbf{w}}^\star, \mathbf{x}_{k,i}^{(1)} \right\rangle \right| \ge a - b_k, \; 0 \le \left\langle \mathbf{w}_{k,i}, \mathbf{x}_{k,i}^{(1)} \right\rangle \le \gamma + b_k \right),
\end{aligned}
$$

where the last inequality holds because $\left\langle \mathbf{w}^\star, \mathbf{x}_{\perp \mathbf{w}}^{(1)} \right\rangle = \left\langle \mathbf{w}_{\perp \mathbf{w}}^\star, \mathbf{x}_{\perp \mathbf{w}}^{(1)} \right\rangle + \left\langle \mathbf{w}_{\|\mathbf{w}}^\star, \mathbf{x}_{\perp \mathbf{w}}^{(1)} \right\rangle = \left\langle \mathbf{w}_{\perp \mathbf{w}}^\star, \mathbf{x}_{\perp \mathbf{w}}^{(1)} \right\rangle = \left\langle \mathbf{w}_{\perp \mathbf{w}}^\star, \mathbf{x}_{\perp \mathbf{w}}^{(1)} \right\rangle + \left\langle \mathbf{w}_{\perp \mathbf{w}}^\star, \mathbf{x}_{\|\mathbf{w}}^{(1)} \right\rangle = \left\langle \mathbf{w}_{\perp \mathbf{w}}^\star, \mathbf{x}_{k,i}^{(1)} \right\rangle$.

Let $X := \left\langle \frac{\mathbf{w}_{\perp \mathbf{w}}^\star}{\|\mathbf{w}_{\perp \mathbf{w}}^\star\|_2}, \mathbf{x}_{k,i} \right\rangle$ and $Y := \langle \mathbf{w}_{k,i}, \mathbf{x}_{k,i} \rangle$. Then, $(X, Y)$ forms a projection of $\mathbf{x}_{k,i}$ onto a 2-dimensional subspace $V_2$ spanned by $\frac{\mathbf{w}_{\perp \mathbf{w}}^\star}{\|\mathbf{w}_{\perp \mathbf{w}}^\star\|_2}$ and $\mathbf{w}_{k,i}$. Let $\phi_{V_2}$ denote the density of $(X, Y)$. By

condition 2 of Assumption 2 $\phi_{V_2}(X, Y) \leq U_2 \exp(-\delta\|(X, Y)\|_2)$, thus, we can bound the above probability by

$$\mathbb{P}\left(\left|\left\langle\mathbf{w}^\star, \mathbf{x}_{\perp\mathbf{w}}^{(1)}\right\rangle \mathbb{I}\left(\mathbf{x} \in D_{k,i}^{(1)}\right)\right| \geq a - b_k\right) = \int_{\frac{a-b_k}{\|\mathbf{w}_{\perp\mathbf{w}}^\star\|_2}}^{+\infty} \int_0^{\gamma+b_k} \phi(X, Y) dX dY$$

$$\leq U_2 \int_{\frac{a-b_k}{\|\mathbf{w}_{\perp w}^\star\|_2}}^{+\infty} \int_0^{\gamma+b_k} \exp(-\delta\sqrt{X^2 + Y^2}) dX dY$$

$$\leq U_2(\gamma + b_k) \int_{\frac{a-b_k}{\|\mathbf{w}_{\perp\mathbf{w}}^\star\|_2}}^{+\infty} \exp(-\delta X) dX dY$$

$$= \frac{U_2(\gamma + b_k)}{\delta} \exp\left(-\delta\frac{a - b_k}{\|\mathbf{w}_{\perp\mathbf{w}}^\star\|_2}\right)$$

$$\leq \frac{U_2(\gamma + b_k)}{\delta} \exp\left(-\delta\frac{a - b_k}{r_k}\right)$$

$$\leq \frac{U_2 \exp(\delta)(\gamma + b_k)}{\delta} \exp\left(-\delta\frac{a}{r_k}\right),$$

where the third inequality holds since $\|\mathbf{w}^\star - \mathbf{w}_{k,i}\| \leq r_k < \frac{\pi}{2}$, which implies $\|\mathbf{w}_{\perp\mathbf{w}}^\star\|_2 \leq r_k$. The last inequality holds since $b_k < r_k$ by our setting.

Since $\left|\left(\frac{1}{2}y_{k,i}^{(1)} - \left(\frac{1}{2} - \bar{\eta}\right)\right)\right| \leq 1$, for $a \geq b_k$, we have

$$\mathbb{P}\left(\left|\left\langle\mathbf{w}^\star, \mathbf{x}_{\perp\mathbf{w}}^{(1)}\right\rangle\left(\frac{1}{2} - \left(\frac{1}{2} - \bar{\eta}\right)y_{k,i}^{(1)}\right)\mathbb{I}\left(\mathbf{x}_{k,i}^{(1)} \in D^{(1)}\right)\right| \geq a - b_k\right)$$

$$\leq \mathbb{P}\left(\left|\left\langle\mathbf{w}^\star, \mathbf{x}_{\perp\mathbf{w}}^{(1)}\right\rangle\mathbb{I}\left(\mathbf{x}_{k,i}^{(1)} \in D_{k,i}^{(1)}\right)\right| \geq a - b_k\right) \leq \frac{U_2 \exp(\delta)(\gamma + b_k)}{\delta} \exp\left(-\delta\frac{a}{r_k}\right). \tag{32}$$

Combing (32) and (31), we get that for $a \geq b_k$,

$$\mathbb{P}\left(\left|\left\langle\mathbf{w}^\star, \bar{\eta}\tilde{\mathbf{x}}_{k,i}^{(1,+)} - (1 - \bar{\eta})\tilde{\mathbf{x}}_{k,i}^{(1,-)}\right\rangle\right| \geq a\right) \leq \mathbb{P}\left(\left|\left\langle\mathbf{w}^*, \left(\bar{\eta}\tilde{\mathbf{x}}_{k,i}^{(1,+)} - (1 - \bar{\eta})\tilde{\mathbf{x}}_{k,i}^{(1,-)}\right)_{\perp\mathbf{w}}\right\rangle\right| \geq a - b_k\right)$$

$$\leq \frac{U_2 \exp(\delta)(\gamma + b_k)}{\delta} \exp\left(-\delta\frac{a}{r_k}\right). \tag{33}$$

For $0 < a < b_k$,

$$\mathbb{P}\left(\left|\left\langle\mathbf{w}^\star, \bar{\eta}\hat{\mathbf{x}}_{k,i}^{(1,+)} - (1 - \bar{\eta})\hat{\mathbf{x}}_{k,i}^{(1,-)}\right\rangle\right| \geq a\right) \leq \mathbb{P}\left(\left|\left\langle\mathbf{w}^\star, \bar{\eta}\hat{\mathbf{x}}_{k,i}^{(1,+)} - (1 - \bar{\eta})\hat{\mathbf{x}}_{k,i}^{(1,-)}\right\rangle\right| > 0\right)$$

$$\leq \mathbb{P}\left(\mathbf{x}_{k,i}^{(1)} \in D_{k,i}^{(1)}\right)$$

$$\leq U_1(\gamma + b_k) \tag{34}$$

$$\leq U_1 \exp(\delta)(\gamma + b_k) \exp\left(-\delta\frac{a}{r_k}\right) .$$

Where the third inequality holds by Lemma 4 property (a) and the last equality holds since $a < b_k < r_k$.

Combing (33) and (34), we establish probability tail bound of (a) in (27): for $\forall a > 0$ and $\rho = \max\left\{U_1 \exp(\delta), \frac{U_2 \exp(\delta)}{\delta}\right\}$, we have

$$\mathbb{P}\left(\left|\left\langle\mathbf{w}^\star, \bar{\eta}\hat{\mathbf{x}}_{k,i}^{(1,+)} - (1 - \bar{\eta})\hat{\mathbf{x}}_{k,i}^{(1,-)}\right\rangle\right| \geq a\right) \leq \rho(\gamma + b_k) \exp\left(-\delta\frac{a}{r_k}\right).$$

Following the same technique, we can bound part (b) of (27) for $a > 0$ by:

$$\mathbb{P}\left(\left|\left\langle\mathbf{w}^\star, \bar{\eta}\hat{\mathbf{x}}_{k,i}^{(2,+)} - (1 - \bar{\eta})\hat{x}_{k,i}^{(2,-)}\right\rangle\right| \geq a\right) \leq \rho\gamma \exp\left(-\delta\frac{a}{r_k}\right).$$

By Lemma 9, part (a) and part (b) in (27) are $\left(\frac{6}{\delta}\sqrt{1+2\rho(\gamma+b_k)}r_k, \frac{6}{\delta}r_k\right)$-subexponential, thus, by Lemma 7, we get that with probability at least $1-1/T^2$,

$$
\sum_{i=1}^{T_k} \mathbb{E}\left[\left\langle \mathbf{w}^\star, \bar{\eta}\tilde{\mathbf{x}}_{k,i}^{(1,+)} - (1-\bar{\eta})\tilde{\mathbf{x}}_{k,i}^{(1,-)}\right\rangle \middle| \mathcal{F}_{k,i-1}\right]
$$
$$
\leq \sum_{i=1}^{T_k} \left\langle \mathbf{w}^\star, \bar{\eta}\tilde{\mathbf{x}}_{k,i}^{(1,+)} - (1-\bar{\eta})\tilde{\mathbf{x}}_{k,i}^{(1,-)}\right\rangle + \frac{12}{\delta}\sqrt{1+2\rho(\gamma+b_k)}r_k\sqrt{T_k\ln T} + \frac{24}{\delta}r_k\ln T, \tag{35}
$$

and with probability at least $1-1/T^2$,

$$
\sum_{i=1}^{T_k} \mathbb{E}\left[\left\langle \mathbf{w}^\star, \bar{\eta}\tilde{\mathbf{x}}_{k,i}^{(2,+)} - (1-\bar{\eta})\tilde{\mathbf{x}}_{k,i}^{(2,-)}\right\rangle \middle| \mathcal{F}_{k,i-1}\right]
$$
$$
\geq \sum_{i=1}^{T_k} \left\langle \mathbf{w}^\star, \bar{\eta}\tilde{\mathbf{x}}_{k,i}^{(2,+)} - (1-\bar{\eta})\tilde{\mathbf{x}}_{k,i}^{(2,-)}\right\rangle - \frac{12}{\delta}\sqrt{1+2\rho(\gamma+b_k)}r_k\sqrt{T_k\ln T} - \frac{24}{\delta}r_k\ln T. \tag{36}
$$

Taking the union bound of (35) and (36), we get that with probability at least $1-2/T^2$,

$$
\sum_{i=1}^{T_k} \mathbb{E}[\langle \mathbf{w}^\star, -\hat{\mathbf{g}}_{k,i}\rangle)| \, \mathcal{F}_{k,i-1}] \leq \sum_{i=1}^{T_k} \langle \mathbf{w}^\star, -\hat{\mathbf{g}}_{k,i}\rangle + \frac{24}{\delta}\sqrt{1+2\rho(\gamma+b_k)}r_k\sqrt{T_k\ln T} + \frac{48}{\delta}r_k\ln T.
$$

$\square$

We then show that by our construction of $\hat{\mathbf{g}}_{k,i}$, $\sum_{i=1}^{T_k}\langle -\mathbf{w}_{k,i}, \hat{\mathbf{g}}_{k,i}\rangle$ also has a high probability upper bound.

**Lemma 21** (High probability bound of $\sum_{i=1}^{T_k}\langle -\mathbf{w}_{k,i}, \hat{\mathbf{g}}_{k,i}\rangle$). *At batch k, with probability at least* $1-2/T^2$, *we have*

$$
\sum_{i=1}^{T_k} \langle -\mathbf{w}_{k,i}, \hat{\mathbf{g}}_{k,i}\rangle \leq \bar{\eta}b_k\sum_{i=1}^{T_k}\mathbb{P}\left(\mathbf{x}\in D_{k,i}\right) + 2\bar{\eta}b_k\sqrt{T_k\ln T}.
$$

*Proof.* Since $\hat{\mathbf{g}}_{k,i} = -\bar{\eta}(\hat{\mathbf{x}}_{k,i}^{(1,+)} - \hat{\mathbf{x}}_{k,i}^{(2,+)}) + (1-\bar{\eta})(\hat{\mathbf{x}}_{k,i}^{(1,-)} - \hat{\mathbf{x}}_{k,i}^{(2,-)})$, and we have the

$$
\left\langle \mathbf{w}_{k,i}, \hat{\mathbf{x}}_{k,i}^{(1,+)}\right\rangle = b_k\mathbb{I}(y_{k,i}^{(1)}=1)\mathbb{I}(\mathbf{x}_{k,i}^{(1)}\in D_{k,i}^{(1)}),
$$
$$
\left\langle \mathbf{w}_{k,i}, \hat{\mathbf{x}}_{k,i}^{(2,+)}\right\rangle = b_k\mathbb{I}(y_{k,i}^{(2)}=1)\mathbb{I}(\mathbf{x}_{k,i}^{(2)}\in D_{k,i}^{(2)}),
$$
$$
\left\langle \mathbf{w}_{k,i}, \hat{\mathbf{x}}_{k,i}^{(1,-)}\right\rangle = \left\langle \mathbf{w}_{k,i}, \hat{\mathbf{x}}_{k,i}^{(2,-)}\right\rangle = 0.
$$

Thus,

$$
\mathbb{E}[\langle \mathbf{w}_{k,i}, -\hat{\mathbf{g}}_{k,i}\rangle] = \bar{\eta}b_k\mathbb{E}\left[\mathbb{I}\left(y=1,\, \mathbf{x}\in D_{k,i}^{(1)}\right) - \mathbb{I}\left(y=1,\, \mathbf{x}\in D_{k,i}^{(2)}\right)\right]
$$
$$
= \bar{\eta}b_k\mathbb{P}\left(y=1,\, \mathbf{x}\in D_{k,i}\right)
$$
$$
= \bar{\eta}b_k\mathbb{P}\left(y=1\mid \mathbf{x}\in D_{k,i}\right)\mathbb{P}\left(\mathbf{x}\in D_{k,i}\right).
$$

Also, since $\hat{\mathbf{x}}_{k,i}^{(1,+)}$ and $\hat{\mathbf{x}}_{k,i}^{(1,-)}$ are calculated by one sample while $\hat{\mathbf{x}}_{k,i}^{(2,+)}$ and $\hat{\mathbf{x}}_{k,i}^{(2,-)}$ are calculated by another independent sample, we can reformulate $\langle \mathbf{w}_{k,i}, -\hat{\mathbf{g}}_{k,i}\rangle$ as the following:

$$
\langle \mathbf{w}_{k,i}, -\hat{\mathbf{g}}_{k,i}\rangle = \underbrace{\left\langle \mathbf{w}_{k,i}, \bar{\eta}\hat{\mathbf{x}}_{k,i}^{(1,+)} - (1-\bar{\eta})\hat{\mathbf{x}}_{k,i}^{(1,-)}\right\rangle}_{(a)} - \underbrace{\left\langle \mathbf{w}_{k,i}, \bar{\eta}\hat{\mathbf{x}}_{k,i}^{(2,+)} - (1-\bar{\eta})\hat{\mathbf{x}}_{k,i}^{(2,-)}\right\rangle}_{(b)}, \tag{37}
$$

where (a) and (b) in (37) are independent. Thus, we establish the high probability bound of parts (a) and (b), respectively.

For part (a), $\left| \left\langle \mathbf{w}_{k,i}, \bar{\eta}\hat{\mathbf{x}}_{k,i}^{(1,+)} - (1-\bar{\eta})\hat{\mathbf{x}}_{k,i}^{(1,-)} \right\rangle \right| \leq \bar{\eta}b_k$, so by Lemma 8, with probability at least $1 - 1/T^2$,

$$\sum_{i=1}^{T_k} \left\langle \mathbf{w}_{k,i}, \bar{\eta}\hat{\mathbf{x}}_{k,i}^{(1,+)} - (1-\bar{\eta})\hat{\mathbf{x}}_{k,i}^{(1,-)} \right\rangle \leq \sum_{i=1}^{T_k} \mathbb{E}\left[ \left\langle \mathbf{w}_{k,i}, \bar{\eta}\hat{\mathbf{x}}_{k,i}^{(1,+)} - (1-\bar{\eta})\hat{\mathbf{x}}_{k,i}^{(1,-)} \right\rangle \middle| \mathcal{F}_{k,i-1} \right]$$
$$+ \bar{\eta}b_k\sqrt{T_k\ln T}.$$

Similarly, for part (b), with probability at least $1 - 1/T^2$,

$$\sum_{i=1}^{T_k} \left\langle \mathbf{w}_{k,i}, \bar{\eta}\hat{\mathbf{x}}_{k,i}^{(2,+)} - (1-\bar{\eta})\hat{\mathbf{x}}_{k,i}^{(2,-)} \right\rangle \geq \sum_{i=1}^{T_k} \mathbb{E}\left[ \left\langle \mathbf{w}_{k,i}, \bar{\eta}\hat{\mathbf{x}}_{k,i}^{(2,+)} - (1-\bar{\eta})\hat{\mathbf{x}}_{k,i}^{(2,-)} \right\rangle \middle| \mathcal{F}_{k,i-1} \right] \quad (38)$$
$$- \bar{\eta}b_k\sqrt{T_k\ln T}.$$

Combing part (a) and (b) above, and taking the union bound, we get that with probability at least $1 - 2/T^2$,

$$\sum_{i=1}^{T_k} \left\langle -\mathbf{w}_{k,i}, \hat{\mathbf{g}}_{k,i} \right\rangle \leq \mathbb{E}\left[ \sum_{i=1}^{T_k} \left\langle -\mathbf{w}_{k,i}, \hat{\mathbf{g}}_{k,i} \right\rangle \right] + 2\bar{\eta}b_k\sqrt{T_k\ln T}$$
$$= \bar{\eta}b_k \sum_{i=1}^{T_k} \Pr(y=1 \mid \mathbf{x} \in D_{k,i})\Pr(\mathbf{x} \in D_{k,i}) + 2\bar{\eta}b_k\sqrt{T_k\ln T} \quad (39)$$
$$\leq \bar{\eta}b_k \sum_{i=1}^{T_k} \Pr(\mathbf{x} \in D_{k,i}) + 2\bar{\eta}b_k\sqrt{T_k\ln T}.$$

$\square$

The following lemma shows a high probability upper bound of $\sum_{i=1}^{T_k}\|\hat{\mathbf{g}}_{k,i}\|_q^2$.

**Lemma 22** (High probability bound of $\sum_{i=1}^{T_k}\|\hat{\mathbf{g}}_{k,i}\|_q^2$). *In Algorithm 4, with probability at least $1 - 2/T^2$,*

$$\sum_{i=1}^{T_k} \|\hat{\mathbf{g}}_{k,i}\|_\infty^2 \leq \frac{144}{Q^2}T_k(\ln T)^2 \,.$$

*Proof.* For $q = \ln(8d) > 2$, $\|\hat{\mathbf{g}}_{k,i}\|_q \leq 2\|\hat{\mathbf{g}}_{k,i}\|_\infty$, hence, we only need to establish the high probability bound of $\|\hat{\mathbf{g}}_{k,i}\|_\infty$. By our construction of $\hat{\mathbf{g}}_{k,i}$,

$$\|\hat{\mathbf{g}}_{k,i}\|_\infty = \left\| -\bar{\eta}(\hat{\mathbf{x}}_{k,i}^{(1,+)} - \hat{\mathbf{x}}_{k,i}^{(2,+)}) + (1-\bar{\eta})(\hat{\mathbf{x}}_{k,i}^{(1,-)} - \hat{\mathbf{x}}_{k,i}^{(2,-)}) \right\|_\infty$$
$$\leq \bar{\eta}\left\|\hat{\mathbf{x}}_{k,i}^{(1,+)}\right\|_\infty + (1-\bar{\eta})\left\|\hat{\mathbf{x}}_{k,i}^{(1,-)}\right\|_\infty + \bar{\eta}\left\|\hat{\mathbf{x}}_{k,i}^{(2,+)}\right\|_\infty + (1-\bar{\eta})\left\|\hat{\mathbf{x}}_{k,i}^{(2,-)}\right\|_\infty$$
$$= \bar{\eta}\left\|[\mathbf{x}_{k,i}^{(1)} + (b_k - \left\langle \mathbf{w}_{k,i}, \mathbf{x}_{k,i}^{(1)} \right\rangle)\mathbf{w}_{k,i}]\mathbb{I}(y_{k,i}=1)\mathbb{I}(\mathbf{x}_{k,i}^{(1)} \in D_{k,i}^{(1)})\right\|_\infty$$
$$+ (1-\bar{\eta})\left\|(\mathbf{x}_{k,i}^{(1)} - \left\langle \mathbf{w}_{k,i}, \mathbf{x}_{k,i}^{(1)} \right\rangle\mathbf{w}_{k,i})\mathbb{I}(y_{k,i}=-1)\mathbb{I}(\mathbf{x}_{k,i}^{(1)} \in D_{k,i}^{(1)})\right\|_\infty$$
$$+ \bar{\eta}\left\|[\mathbf{x}_{k,i}^{(2)} + (b_k - \left\langle \mathbf{w}_{k,i}, \mathbf{x}_{k,i}^{(2)} \right\rangle)\mathbf{w}_{k,i}]\mathbb{I}(y_{k,i}=1)\mathbb{I}(\mathbf{x}_{k,i}^{(2)} \in D_{k,i}^{(2)})\right\|_\infty$$
$$+ (1-\bar{\eta})\left\|(\mathbf{x}_{k,i}^{(2)} - \left\langle \mathbf{w}_{k,i}, \mathbf{x}_{k,i}^{(2)} \right\rangle\mathbf{w}_{k,i})\mathbb{I}(y_{k,i}=-1)\mathbb{I}(\mathbf{x}_{k,i}^{(2)} \in D_{k,i})\right\|_\infty$$
$$\leq \left\|\mathbf{x}_{k,i}^{(1)}\right\|_\infty + \left\|\mathbf{x}_{k,i}^{(2)}\right\|_\infty + 2(\gamma + b_k).$$

From Lemma 13, we get that ,with probability at least $1 - \frac{1}{T_kT^2}$,

$$\left\|\mathbf{x}_{k,i}^{(1)}\right\|_\infty \leq \frac{1}{Q}\left(1 + \ln(dT_kT^2)\right), \quad (40)$$

and with probability at least $1 - \frac{1}{T_k T^2}$,

$$\left\| \mathbf{x}_{k,i}^{(2)} \right\|_\infty \leq \frac{1}{Q} \left( 1 + \ln(dT_k T^2) \right). \tag{41}$$

Taking the union bound of (40),(41), we get that with probability at least $1 - \frac{2}{T_k T^2}$,

$$\|\hat{\mathbf{g}}_{k,i}\|_\infty \leq \frac{2}{Q} + \frac{2}{Q} \ln(dT_k T^2) + 2(\gamma + b_k).$$

Taking union bound over all iterations $i \in [T_k]$, we have that with probability at least $1 - 2/T^2$,

$$\sum_{i=1}^{T_k} \|\hat{\mathbf{g}}_{k,i}\|_\infty^2 \leq T_k \left( \frac{2}{Q} + \frac{2}{Q} \ln(dT_k T^2) + 2(\gamma + b_k) \right)^2 \leq T_k \left( \frac{2}{Q} \ln(T^3) \right)^2 = \frac{36}{Q^2} T_k (\ln T)^2.$$

Thus, we conclude that with probability at least $1 - 2/T^2$,

$$\sum_{i=1}^{T_k} \|\hat{\mathbf{g}}_{k,i}\|_q^2 \leq 4 \sum_{i=1}^{T_k} \|\hat{\mathbf{g}}_{k,i}\|_\infty^2 \leq \frac{144}{Q^2} T_k (\ln T)^2.$$

$\square$

Given the starting angle of batch $k$ as $\theta_k = \frac{\pi}{2^{k+1}}$, the following lemma establishes the high probability upper bound of the average of $f_{k,i}(\mathbf{w}_{k,i})$.

**Lemma 23.** *In Algorithm 4, at every batch $k \in \{1, 2, \cdots, K\}$, if $\theta(\mathbf{w}^\star, \mathbf{w}_k) \leq \theta_k$, there exists some constants $c_5, c_6, c_7 > 0$, when setting bandwidth $b_k = c_5 \frac{1 - 2\bar\eta}{2^k}$, iteration number $T_k = c_6 \frac{(\gamma+1)d \ln d \ln T}{(1 - 2\bar\eta)^4} \cdot 4^k$, step size $\alpha_k = c_7 \frac{\sqrt{d}\theta_k}{T_k \ln T}$, then with probability at least $1 - \frac{6}{T^2}$, the following holds*

$$\frac{1}{T_k} \sum_{i=1}^{T_k} f_{k,i}(\mathbf{w}_{k,i}) \leq \frac{L_2 R^2 \theta_k}{12800 U_1}. \tag{42}$$

*Proof.* Combing Lemma 10, Lemma 18 and Lemma 19, we have $(1 - 2\bar\eta) f_{k,i}(\mathbf{w}_{k,i}) \mathbb{P}(\mathbf{x} \in D_{k,i}) \leq \mathbb{E}[\langle \mathbf{w}^\star, -\hat{\mathbf{g}}_{k,i}\rangle]$, hence, it suffices to upper bound $\sum_{i=1}^{T_k} \mathbb{E}[\langle \mathbf{w}^\star, -\hat{\mathbf{g}}_{k,i}\rangle]$.

First, we upper bound $\sum_{i=1}^{T_k} \mathbb{E}[\langle \mathbf{w}^\star, -\hat{\mathbf{g}}_{k,i}\rangle]$ by $\sum_{i=1}^{T_k} \langle \mathbf{w}^\star, -\hat{\mathbf{g}}_{k,i}\rangle$.

By our setting of constraint set, $\cos(\theta(\mathbf{w}_{k,i}, \mathbf{w}_k)) = \langle \mathbf{w}_{k,i}, \mathbf{w}_k \rangle \geq \cos\theta_k$, since $\cos\theta$ is decreasing in $\theta \in [0, \pi]$, hence, we have $\theta(\mathbf{w}_{k,i}, \mathbf{w}_k) \leq \theta_k$, thus, $\|\mathbf{w}_{k,i} - \mathbf{w}_k\|_2 \leq \theta(\mathbf{w}_{k,i}, \mathbf{w}_k) \leq \theta_k$. Also, $\|\mathbf{w}_k - \mathbf{w}^\star\|_2 \leq \theta(\mathbf{w}_k, \mathbf{w}^\star) \leq \theta_k$, by Lemma 12, $\|\mathbf{w}_{k,i} - \mathbf{w}^\star\|_2 \leq 2\theta_k$. According to Lemma 20, set $r_k = 2\theta_k$, then with probability at least $1 - 2/T^2$, the following holds:

$$\sum_{i=1}^{T_k} \mathbb{E}[\langle \mathbf{w}^\star, -\hat{\mathbf{g}}_{k,i}\rangle)|\ \mathcal{F}_{k,i-1}] \leq \sum_{i=1}^{T_k} \langle \mathbf{w}^\star, -\hat{\mathbf{g}}_{k,i}\rangle \frac{48}{\delta} \sqrt{1 + 2\rho(\gamma + b_k)} \theta_k \sqrt{T_k \ln T} + \frac{96}{\delta} \theta_k \ln T. \tag{43}$$

Next, we move on to upper bound $\sum_{i=1}^{T_k} \langle \mathbf{w}^\star, -\hat{\mathbf{g}}_{k,i}\rangle$ through a nonstandard regret analysis of online mirror decent. Let $B(\mathbf{v}_1, \mathbf{v}_2) := \frac{1}{2(p-1)} \|\mathbf{v}_1 - \mathbf{v}_2\|_p^2$ denotes the Bregman divergence w.r.t. $\frac{1}{2(p-1)} \| \cdot \|_p^2$, where $p = \frac{\ln(8d)}{\ln(8d)-1}$. In each iteration $i$, the regularizer, $B(\cdot, \mathbf{w}_{k,i-1})$ is 1-strongly convex with respect to $\| \cdot \|_p$ [see Shalev-Shwartz [2007]]. From the analysis of online mirror descent [see Orabona [2023], Lemma 6.9], with step size $\alpha_k$, we have

$$\langle \alpha_k \hat{\mathbf{g}}_{k,i}, \mathbf{w}_{k,i} - \mathbf{w}^\star\rangle \leq B(\mathbf{w}^\star, \mathbf{w}_{k,i}) - B(\mathbf{w}^\star, \mathbf{w}_{k,i+1}) + \frac{\alpha_k^2}{2} \|\hat{\mathbf{g}}_{k,i}\|_q^2.$$

Where $q = \ln(8d) > 2$. Summing the above equality over $i \in [T_k]$, we get

$$\sum_{i=1}^{T_k} \langle \alpha_k \hat{\mathbf{g}}_{k,i}, \mathbf{w}_{k,i} - \mathbf{w}^\star\rangle \leq B(\mathbf{w}^\star, \mathbf{w}_{k,1}) - B(\mathbf{w}^\star, \mathbf{w}_{k,T_k+1}) + \frac{\alpha_k^2}{2} \sum_{i=1}^{T_k} \|\hat{\mathbf{g}}_{k,i}\|_q^2.$$

Dividing both sides by $\alpha_k$, and moving $\sum_{i=1}^{T_k} \langle \mathbf{w}_{k,i}, \hat{\mathbf{g}}_{k,i} \rangle$ to RHS, we get

$$
\begin{aligned}
\sum_{i=1}^{T_k} \langle \mathbf{w}^\star, -\hat{\mathbf{g}}_k \rangle &\leq \frac{1}{\alpha_k} [B(\mathbf{w}^\star, \mathbf{w}_{k,1}) - B(\mathbf{w}^\star, \mathbf{w}_{k,T_k+1})] + \sum_{i=1}^{T_k} \langle \mathbf{w}_{k,i}, -\hat{\mathbf{g}}_{k,i} \rangle + \frac{\alpha_k}{2} \sum_{i=1}^{T_k} \|\hat{\mathbf{g}}_{k,i}\|_q^2 \\
&\leq \frac{1}{\alpha_k} B(\mathbf{w}^\star, \mathbf{w}_{k,1}) + \sum_{i=1}^{T_k} \langle \mathbf{w}_{k,i}, -\hat{\mathbf{g}}_{k,i} \rangle + \frac{\alpha_k}{2} \sum_{i=1}^{T_k} \|\hat{\mathbf{g}}_{k,i}\|_q^2.
\end{aligned}
$$
(44)

Now we need to bound the three terms in the RHS of (44) respectively.

First, we bound $B(\mathbf{w}^\star, \mathbf{w}_{k,1}) = B(\mathbf{w}^\star, \mathbf{w}_k)$.

$$
B(\mathbf{w}^\star, \mathbf{w}_{k,1}) = \frac{\|\mathbf{w}^\star - \mathbf{w}_k\|_p^2}{2(p-1)} \overset{(a)}{\leq} \frac{\|\mathbf{w}^\star - \mathbf{w}_k\|_1^2}{2(p-1)} \overset{(b)}{\leq} \frac{d\|\mathbf{w}^\star - \mathbf{w}_k\|_2^2}{2(p-1)} \overset{(c)}{\leq} \frac{d\ln(8d)\theta_k^2}{2}.
$$
(45)

Where inequality (a) holds by the fact that $\|\mathbf{x}\|_p \leq \|\mathbf{x}\|_1$ for all $p > 1$ and $\mathbf{x} \in \mathbb{R}^d$. Inequality (b) holds since $\|\mathbf{x}\|_1 \leq \sqrt{d}\|\mathbf{x}\|_2$ for all $\mathbf{x} \in \mathbb{R}^d$. Inequality (c) holds since $\|\mathbf{w}^\star - \mathbf{w}_k\|_2 \leq \theta(\mathbf{w}^\star, \mathbf{w}_k) \leq \theta_k$.

Next, we bound $\sum_{i=1}^{T_k} \langle -\mathbf{w}_{k,i}, \hat{\mathbf{g}}_{k,i} \rangle$. By Lemma 21, we have that with probability at least $1 - 2/T^2$,

$$
\sum_{i=1}^{T_k} \langle -\mathbf{w}_{k,i}, \hat{\mathbf{g}}_{k,i} \rangle \leq \bar{\eta} b_k \sum_{i=1}^{T_k} \mathbb{P}(\mathbf{x} \in D_{k,i}) + 2\bar{\eta} b_k \sqrt{T_k \ln T}.
$$
(46)

Finally, we bound $\sum_{i=1}^{T_k} \|\hat{\mathbf{g}}_{k,i}\|_q^2$. By Lemma 22, we have with probability at least $1 - 2/T^2$,

$$
\sum_{i=1}^{T_k} \|\hat{\mathbf{g}}_{k,i}\|_\infty^2 \leq \frac{144}{Q^2} T_k (\ln T)^2.
$$
(47)

Combining (43), (45), (46) and (47), and taking the union bound, we get that with probability at least $1 - 6/T^2$,

$$
\begin{aligned}
(1 - 2\bar{\eta}) &\sum_{i=1}^{T_k} f_{k,i}(\mathbf{w}_{k,i}) \mathbb{P}(\mathbf{x} \in D_{k,i}) \\
&\leq \sum_{i=1}^{T_k} \mathbb{E}[\langle \mathbf{w}^\star, -\hat{\mathbf{g}}_{k,i} \rangle] \\
&\leq \frac{1}{\alpha_k} \frac{d\ln(8d)\theta_k^2}{2} + \bar{\eta} b_k \sum_{i=1}^{T_k} \mathbb{P}(\mathbf{x} \in D_{k,i}) + 2\bar{\eta} b_k \sqrt{T_k \ln T} + \frac{72}{Q^2} \alpha_k T_k (\ln T)^2 \\
&\quad + \frac{48}{\delta} \sqrt{1 + 2\rho(\gamma + b_k)} \theta_k \sqrt{T_k \ln T} + \frac{96}{\delta} \theta_k \ln T.
\end{aligned}
$$
(48)

By Lemma 4, property (a), and the fact that $b_k \leq b_1 < R$, we have $L_1 b_k \leq \mathbb{P}(\mathbf{x} \in D_{k,i}) \leq U_1 b_k$, dividing both sides of (48) by $(1 - 2\bar{\eta}) L_1 T_k b_k$, we get

$$
\begin{aligned}
\frac{1}{T_k} &\sum_{i=1}^{T_k} f_{k,i}(\mathbf{w}_{k,i}) \\
&\leq \frac{1}{\alpha_k} \frac{d\ln(8d)\theta_k^2}{2(1 - 2\bar{\eta}) L_1 T_k b_k} + \frac{U_1 \bar{\eta}}{(1 - 2\bar{\eta}) L_1} b_k + \frac{2\bar{\eta}}{(1 - 2\bar{\eta}) L_1} \sqrt{\frac{\ln T}{T_k}} + \frac{72\alpha_k}{Q^2(1 - 2\bar{\eta}) L_1 b_k} (\ln T)^2 \quad (49) \\
&\quad + \frac{48}{\delta(1 - 2\bar{\eta}) L_1} \sqrt{1 + 2\rho(\gamma + b_k)} \frac{\theta_k}{b_k} \sqrt{\frac{\ln T}{T_k}} + \frac{96}{\delta(1 - 2\bar{\eta}) L_1} \frac{\theta_k}{b_k} \frac{\ln T}{T_k}.
\end{aligned}
$$

By our setting, $b_k = \frac{(1 - 2\bar{\eta}) L_1 L_2 R^2}{38400 U_1^2} \theta_k = c_5 \frac{1 - 2\bar{\eta}}{2^k}$, $T_k = \frac{576\pi^2 (1 + 2\rho(\gamma + b_1)) d\ln(8d)(\ln T)^2}{\delta^2 c_5^2 (1 - 2\bar{\eta})^2 b_k^2} = c_6 \frac{(\gamma + 1) d\ln d(\ln T)^2}{(1 - 2\bar{\eta})^4} 4^k$, $\alpha_k = \frac{Q\sqrt{d\ln(8d)}\theta_k}{12\sqrt{T_k \ln T}} = c_7 \frac{\sqrt{d\ln d}\theta_k}{\sqrt{T_k \ln T}}$, then we get our proof. $\qquad \square$

Based on Lemma 23, we establish the main theoretical guarantee of Algorithm 4 in Proposition 4.

**Proposition 4.** *For some constants $c_5, c_6, c_7 > 0$, when Algorithm 4 runs with an initial vector $\mathbf{w}_k$ satisfying $\theta(\mathbf{w}^\star, \mathbf{w}_k) \leq \theta_k = \pi/2^{k+1}$, bandwidth $b_k = c_5(1 - 2\bar{\eta})2^{-k}$ for $T_k = c_6 4^k(\gamma + 1)d \ln d(\ln T)^2/(1 - 2\bar{\eta})^4$ iterations with step size $\alpha_k = c_7\sqrt{d \ln d}\theta_k/(\sqrt{T_k}\ln T)$, its output $\mathbf{w}_{k+1}$ satisfies $\theta(\mathbf{w}^\star, \mathbf{w}_{k+1}) \leq \theta_{k+1} = \frac{\theta_k}{2}$ with probability at least $1 - 6/T^2$.*

*Proof.* For the given unit vector $\mathbf{w}_k$ that satisfy $\theta(\mathbf{w}^\star, \mathbf{w}_k) \leq \theta_k \leq \frac{\pi}{4}$, we have

$$\|\mathbf{w}^\star - \mathbf{w}_k\|_2 \leq 2\sin\left(\frac{\theta(\mathbf{w}^\star, \mathbf{w}_k)}{2}\right) \leq 2\sin\left(\frac{\theta_k}{2}\right) \leq \theta_k.$$

The first inequality holds since $\|\mathbf{w}^\star\|_2 = \|\mathbf{w}_k\|_2 = 1$. The second inequality holds since $\sin x$ is increasing in $x \in [0, \frac{\pi}{2}]$, the last inequality holds since $\sin x \leq x$ for all $0 \leq x \leq \frac{\pi}{2}$.

By our choice of $\mathcal{W}_k$, for every iteration $i$, we have $\cos(\theta(\mathbf{w}_{k,i}, \mathbf{w}_k)) = \langle \mathbf{w}_{k,i}, \mathbf{w}_k \rangle \geq \cos(\theta_k)$, thus, since $\cos x$ is decreasing for $0 \leq x \leq \pi$, then $\theta(\mathbf{w}_{k,i}, \mathbf{w}_k) \leq \theta_k$, hence,

$$\theta(\mathbf{w}^\star, \mathbf{w}_{k,i}) \leq \theta(\mathbf{w}^\star, \mathbf{w}_k) + \theta(\mathbf{w}_k, \mathbf{w}_{k,i}) \leq 2\theta_k, \tag{50}$$

where the first inequality holds by Lemma 12. By Lemma 10, with probability at least $1 - \frac{6}{T^2}$, $\frac{1}{T_k}\sum_{i=1}^{T_k} f_{k,i}(\mathbf{w}_{k,i}) \leq \frac{L_2 R^2 \theta_k}{12800 U_1}$. Let $A_k := \{i \in [T_k] \mid f_{k,i}(\mathbf{w}_{k,i}) \geq \frac{L_2}{160 U_1}R^2\theta_k\}$. Thus,

$$\frac{L_2 R^2 \theta_k}{12800 U_1} \geq \frac{1}{T_k}\sum_{i=1}^{T_k} f_{k,i}(\mathbf{w}_{k,i}) \geq \frac{L_2 R^2 \theta_k}{160 U_1}\frac{|A_k|}{T_k}.$$

From the above inequality and we get $\frac{|A_k|}{T_k} \leq \frac{1}{80}$, and thus $\frac{|\bar{A}_k|}{T_k} \geq \frac{79}{80}$. By Corollary 3, the iterations $i' \in \bar{A}_k$ satisfy $\theta(\mathbf{w}^\star, \mathbf{w}_{k,i'}) \leq \frac{\theta_k}{5}$. Other iterations $i' \in A_k$ satisfy $\theta(\mathbf{w}^\star, \mathbf{w}_{k,i}) \leq 2\theta_k$ by (50). Therefore,

$$\begin{aligned}
\frac{1}{T_k}\sum_{i=1}^{T_k}\cos(\theta(\mathbf{w}^\star, \mathbf{w}_{k,i})) &\geq \cos\left(\frac{1}{5}\theta\right) \times \frac{|\bar{A}_k|}{T_k} + \cos(2\theta) \times \frac{|A_k|}{T_k} \\
&\geq \left(1 - \frac{1}{50}\theta^2\right) \times \frac{79}{80} + \left(1 - \frac{1}{2} \times 4\theta^2\right) \times \frac{1}{80} \\
&\geq 1 - \frac{1}{50}\theta^2 - \frac{1}{2} \times \frac{1}{20}\theta^2 \\
&\geq 1 - \frac{1}{20}\theta^2 \\
&= 1 - \frac{1}{5} \times \left(\frac{1}{2}\theta\right)^2 \\
&\geq \cos\left(\frac{\theta}{2}\right),
\end{aligned}$$

where the second inequality utilizes the fact that $\cos x \geq 1 - \frac{1}{2}x^2$ and the last inequality holds since $\cos x \leq 1 - \frac{1}{5}x^2$ for $0 \leq x \leq \frac{\pi}{2}$. By the concavity of $\cos(\theta(\mathbf{w}^\star, \cdot))$ when $\theta(\mathbf{w}^\star, \mathbf{w}_{k,i}) \leq \frac{\pi}{2}$, we have

$$\cos(\theta(\mathbf{w}^\star, \mathbf{w}_{k+1})) = \cos\left(\theta\left(\mathbf{w}^\star, \frac{1}{T_k}\sum_{i=1}^{T_k}\mathbf{w}_{k,i}\right)\right) \geq \frac{1}{T_k}\sum_{i=1}^{T_k}\cos(\theta(\mathbf{w}^\star, \mathbf{w}_{k,i})) \geq \cos\left(\frac{\theta_k}{2}\right).$$

Since $\cos x$ is decreasing in $x \in [0, \pi]$, we have $\theta(\mathbf{w}^\star, \mathbf{w}_{k+1}) \leq \frac{\theta_k}{2}$. $\qquad\square$

### A.6 Proofs for Section 4

In this section, we outline the proof of Theorem 1, which is the key theorem of this paper.

**Theorem 1.** *For any instance of our online strategic classification problem with noise level $\bar{\eta}$, maximum manipulation distance $\gamma$, and feature dimension $d$, the expected regret of classifiers $\tilde{\mathbf{h}}$ from Algorithm 1 over $T$ cycles satisfies*

$$\mathbb{E}[\text{Reg}(\tilde{\mathbf{h}}; T)] = O\left(d\ln d \times (\ln T)^2/(1 - 2\bar{\eta})^8 + \sqrt{(\gamma + 1)d\ln d \times T}\ln T/(1 - 2\bar{\eta})^2\right).$$

*Proof.* To derive the regret bound in Theorem 1, we decompose the total regret $\text{Reg}(\tilde{\mathbf{h}}; T)$ as defined in (1) into two parts according to pure exploration phase versus exploration-exploitation phase, and then we move on to decompose the regret in the exploration-exploitation phase according to certain events. We upper bound each of these parts separately. First, we define the two phases and the events used in the regret decomposition.

**Definition 4.** *Define the set $\mathcal{T}_{PE} := \{t \in [T] \mid 0 < t \leq 2T_{init} + T_0\}$ as the pure exploration phase, where $T_{init}$ and $T_0$ are number of iterations in Algorithm 2 and Algorithm 3. Define the set $\mathcal{T}_{EE} := \{t \in [T] \mid 2T_{init} + T_0 < t \leq T\}$ as the exploration-exploitation phase. Define the event $\varepsilon_{init} := \{\langle \bar{\mathbf{w}}_0, \mathbf{w}^\star \rangle \geq c_0(1 - 2\bar{\eta})\}$, where $c_0$ is a constant defined in Proposition 2. For $k \in \{0, 1, 2, \cdots, K\}$, define the event $\varepsilon_k := \{\theta(\mathbf{w}_{k+1}, \mathbf{w}^\star) \leq \frac{\pi}{2^{k+2}}\}$. Define the event $\varepsilon := \varepsilon_{init} \bigcap \varepsilon_0 \bigcap_{k \in [K]} \varepsilon_k$ as the "clean event".*

In Definition 4, the pure exploration phase $\mathcal{T}_{PE}$ corresponds to all cycles in the Initialization and Refinement Algorithm, and the exploration-exploitation phase $\mathcal{T}_{EE}$ corresponds to all cycles in the Enhancement Algorithm. By Proposition 2, Proposition 3 and Proposition 4, the events defined in Definition 4 satisfy the following properties:

$$\mathbb{P}(\varepsilon_{init}) \geq 1 - \frac{2}{T^2}, \tag{51}$$

$$\mathbb{P}(\varepsilon_0 \mid \varepsilon_{init}) \geq 1 - \frac{3}{T^2}, \tag{52}$$

$$\mathbb{P}(\varepsilon_k \mid \varepsilon_{k-1}, \varepsilon_{k-2}, \cdots \varepsilon_0, \varepsilon_{init}) = \mathbb{P}(\varepsilon_k \mid \varepsilon_{k-1}) \geq 1 - \frac{6}{T^2}, \forall k \in [K]. \tag{53}$$

Hence, taking the union bound by (51),(52) and (53), we get that the probability of clean event satisfy

$$\mathbb{P}(\varepsilon) = \mathbb{P}\left(\varepsilon_{init} \bigcap \varepsilon_0 \bigcap_{k \in [K]} \varepsilon_k\right) \geq \left(1 - \frac{2}{T^2}\right) \times \left(1 - \frac{3}{T^2}\right) \times \left(1 - \frac{6}{T^2}\right)^K \geq 1 - \frac{6}{T}.$$

Then, we decompose the total regret as

$$\text{Reg}(\tilde{\mathbf{h}}; T) = \text{Reg}(\tilde{\mathbf{h}}; \mathcal{T}_{PE}) + \text{Reg}(\tilde{\mathbf{h}}; \mathcal{T}_{EE}, \varepsilon) + \text{Reg}(\tilde{\mathbf{h}}; \mathcal{T}_{EE}, \bar{\varepsilon}), \tag{54}$$

where $\bar{\varepsilon}$ denotes the complement of $\varepsilon$, and the three parts of (54) is represented as the following:

$$\text{Reg}(\tilde{\mathbf{h}}; \mathcal{T}_{PE}) = \sum_{t \in \mathcal{T}_{PE}} \text{Err}(\tilde{h}_t) - |\mathcal{T}_{PE}| \times \text{Err}(\tilde{h}^\star),$$

$$\text{Reg}(\tilde{\mathbf{h}}; \mathcal{T}_{EE}, \varepsilon) = \sum_{t \in \mathcal{T}_{EE}} \left(\text{Err}(\tilde{h}_t) - \text{Err}(\tilde{h}^\star)\right) \mathbb{I}(\varepsilon),$$

$$\text{Reg}(\tilde{\mathbf{h}}; \mathcal{T}_{EE}, \bar{\varepsilon}) = \sum_{t \in \mathcal{T}_{EE}} \left(\text{Err}(\tilde{h}_t) - \text{Err}(\tilde{h}^\star)\right) \mathbb{I}(\bar{\varepsilon}).$$

The first term in (54) denotes the expected regret incurred during the pure exploration phase. The second term captures the expected regret incurred during the exploration-exploitation phase, given that the clean event holds. The last term characterizes the expected regret incurred during the exploration-exploitation phase, given that the clean event does not hold.

Now we upper bound the three parts of (54) respectively. For the first term $\text{Reg}(\tilde{\mathbf{h}}; \mathcal{T}_{PE})$, the regret incurred in a single cycle is at most 1, and the length of pure exploration phase is $|\mathcal{T}_{PE}| = 2T_{init} + T_0$, then, the expected total regret during these time can be upper bounded by

$$\mathbb{E}\left[\text{Reg}(\tilde{\mathbf{h}}; \mathcal{T}_{PE})\right] \leq \sum_{t \in \mathcal{T}_{PE}} 1 \leq 2T_{init} + T_0 = O\left(\frac{1}{(1 - 2\eta)^8} d\ln d(\ln T)^2\right), \tag{55}$$

where the last equality holds by our setting that $T_{\text{init}} = O\left(\frac{1}{(1-2\eta)^2}\ln T\right)$ and $T_0 = O\left(\frac{1}{(1-2\eta)^8}d\ln d(\ln T)^2\right)$.

Then, we upper bound the expectation of the second term $\text{Reg}(\tilde{\mathbf{h}}; \mathcal{T}_{EE}, \varepsilon)$, which is the cumulative regret incurred under the clean event during the Enhancement procedure. Let $\text{Reg}_k(\tilde{\mathbf{h}}; \mathcal{T}_{EE}, \varepsilon)$ denote the regret in each batch $k \in \{1, 2, \cdots, K\}$ during this procedure under "clean event" *i.e.*, $\text{Reg}(\tilde{\mathbf{h}}; \mathcal{T}_{EE}, \varepsilon) = \sum_{k=1}^{K}\text{Reg}_k(\tilde{\mathbf{h}}; \mathcal{T}_{EE}, \varepsilon)$, then we only need to upper bound $\mathbb{E}\left[\text{Reg}_k(\tilde{\mathbf{h}}; \mathcal{T}_{EE}, \varepsilon)\right]$ for each batch $k \in \{1, 2, \cdots, K\}$, which is characterized as

$$
\begin{aligned}
&\mathbb{E}\left[\text{Reg}_k(\tilde{\mathbf{h}}; \mathcal{T}_{EE}, \varepsilon)\right] \\
&= \sum_{i=1}^{T_k}\sum_{j=1}^{2}\mathbb{E}\left[\text{Err}(\tilde{h}_{k,i}^{(j)}) - \text{Err}(\tilde{h}^\star)\,\Big|\,\varepsilon\right]\mathbb{P}(\varepsilon) \\
&\leq \sum_{i=1}^{T_k}\sum_{j=1}^{2}\mathbb{E}\left[\text{Err}(\tilde{h}_{k,i}^{(j)}) - \text{Err}(\tilde{h}^\star)\,\Big|\,\varepsilon\right] \\
&\leq \sum_{i=1}^{T_k}\mathbb{E}\left[\mathbb{I}\left(\text{sgn}\left(\left\langle\mathbf{w}_{k,i}, \mathbf{r}_{k,i}^{(1)}\right\rangle - \gamma\right) \neq y_{k,i}^{(1)}\right) - \mathbb{I}\left(\text{sgn}\left(\left\langle\mathbf{w}^\star, \mathbf{r}_{k,i}^{(1,*)}\right\rangle - \gamma\right) \neq y_{k,i}^{(1)}\right)\,\Big|\,\varepsilon\right] \\
&\quad + \sum_{i=1}^{T_k}\mathbb{E}\left[\mathbb{I}\left(\text{sgn}\left(\left\langle\mathbf{w}_{k,i}, \mathbf{r}_{k,i}^{(2)}\right\rangle - \gamma - b_k\right) \neq y_{k,i}^{(2)}\right) - \mathbb{I}\left(\text{sgn}\left(\left\langle\mathbf{w}^\star, \mathbf{r}_{k,i}^{(2,*)}\right\rangle - \gamma\right) \neq y_{k,i}^{(2)}\right)\,\Big|\,\varepsilon\right] \\
&= \sum_{i=1}^{T_k}\mathbb{E}\left[\mathbb{I}\left(\text{sgn}\left(\left\langle\mathbf{w}_{k,i}, \mathbf{x}_{k,i}^{(1)}\right\rangle\right) \neq y_{k,i}^{(1)}\right) - \mathbb{I}\left(\text{sgn}\left(\left\langle\mathbf{w}^\star, \mathbf{x}_{k,i}^{(1)}\right\rangle\right) \neq y_{k,i}^{(1)}\right)\,\Big|\,\varepsilon\right] \\
&\quad + \sum_{i=1}^{T_k}\mathbb{E}\left[\mathbb{I}\left(\text{sgn}\left(\left\langle\mathbf{w}_{k,i}, \mathbf{x}_{k,i}^{(2)}\right\rangle - b_k\right) \neq y_{k,i}^{(2)}\right) - \mathbb{I}\left(\text{sgn}\left(\left\langle\mathbf{w}^\star, \mathbf{x}_{k,i}^{(2)}\right\rangle\right) \neq y_{k,i}^{(2)}\right)\,\Big|\,\varepsilon\right].
\end{aligned}
$$

(56)

Where $\mathbf{r}_{k,i}^{(1,*)}$ and $\mathbf{r}_{k,i}^{(2,*)}$ are the counterfactual agent responses under the optimal classifier. The first equality holds by the fact that $\mathbb{E}[X\mathbb{I}(\varepsilon)] = \mathbb{E}[X \mid \mathbb{I}(\varepsilon) = 1]\mathbb{P}(\varepsilon) + \mathbb{E}[0 \mid \mathbb{I}(\varepsilon) = 0]\mathbb{P}(\bar{\varepsilon}) = \mathbb{E}[X \mid \varepsilon]\mathbb{P}(\varepsilon)$. The inequality holds since $0 \leq \mathbb{P}(\varepsilon) \leq 1$. The last equality holds by Proposition 1.

For the first term within the summation in the RHS of (56), we have

$$
\begin{aligned}
&\mathbb{E}\left[\mathbb{I}\left(\text{sgn}\left(\left\langle\mathbf{w}_{k,i}, \mathbf{x}_{k,i}^{(1)}\right\rangle\right) \neq y_{k,i}^{(1)}\right) - \mathbb{I}\left(\text{sgn}\left(\left\langle\mathbf{w}^\star, \mathbf{x}_{k,i}^{(1)}\right\rangle\right) \neq y_{k,i}^{(1)}\right)\,\Big|\,\varepsilon\right] \\
&\leq \mathbb{P}\left(\text{sgn}\left(\langle\mathbf{w}^\star, \mathbf{x}\rangle\right) \neq \text{sgn}\left(\langle\mathbf{w}_{k,i}, \mathbf{x}\rangle\right)\,\Big|\,\varepsilon\right) \\
&\leq \frac{c_{10}}{2^k},
\end{aligned}
$$

(57)

where $c_{10}$ is a positive constant. The first inequality holds by the triangular inequality, the last equality holds since by Lemma 4 property (b), and Proposition 4.

For the second term, we have

$$\mathbb{E}\left[\mathbb{I}\left(\text{sgn}\left(\left\langle \mathbf{w}_{k,i}, \mathbf{x}_{k,i}^{(2)}\right\rangle - b_k\right) \neq y_{k,i}^{(2)}\right) - \mathbb{I}\left(\text{sgn}\left(\left\langle \mathbf{w}^\star, \mathbf{x}_{k,i}^{(2)}\right\rangle\right) \neq y_{k,i}^{(2)}\right) \Big| \varepsilon\right]$$

$$= \mathbb{E}\left[\mathbb{I}\left(\text{sgn}\left(\left\langle \mathbf{w}_{k,i}, \mathbf{x}_{k,i}^{(2)}\right\rangle - b_k\right) \neq y_{k,i}^{(2)}\right) - \mathbb{I}\left(\text{sgn}\left(\left\langle \mathbf{w}_{k,i}, \mathbf{x}_{k,i}^{(2)}\right\rangle\right) \neq y_{k,i}^{(2)}\right) \Big| \varepsilon\right]$$

$$+ \mathbb{E}\left[\mathbb{I}\left(\text{sgn}\left(\left\langle \mathbf{w}_{k,i}, \mathbf{x}_{k,i}^{(2)}\right\rangle\right) \neq y_{k,i}^{(2)}\right) - \mathbb{I}\left(\text{sgn}\left(\left\langle \mathbf{w}^\star, \mathbf{x}_{k,i}^{(2)}\right\rangle\right) \neq y_{k,i}^{(2)}\right) \Big| \varepsilon\right]$$

$$\leq \mathbb{E}\left[\mathbb{I}\left(\text{sgn}\left(\left\langle \mathbf{w}_{k,i}, \mathbf{x}_{k,i}^{(2)}\right\rangle - b_k\right) \neq \text{sgn}\left(\left\langle \mathbf{w}_{k,i}, \mathbf{x}_{k,i}^{(2)}\right\rangle\right)\right) \Big| \varepsilon\right] \tag{58}$$

$$+ \mathbb{E}\left[\mathbb{I}\left(\text{sgn}\left(\left\langle \mathbf{w}_{k,i}, \mathbf{x}_{k,i}^{(2)}\right\rangle\right) \neq \text{sgn}\left(\left\langle \mathbf{w}^\star, \mathbf{x}_{k,i}^{(2)}\right\rangle\right)\right) \Big| \varepsilon\right]$$

$$= \mathbb{P}\left(0 \leq \left\langle \mathbf{w}_{k,i}, \mathbf{x}_{k,i}^{(2)}\right\rangle < b_k\right) + \mathbb{P}\left(\text{sgn}\left(\left\langle \mathbf{w}^\star, \mathbf{x}\right\rangle\right) \neq \text{sgn}\left(\left\langle \mathbf{w}_{k,i}, \mathbf{x}\right\rangle\right) \Big| \varepsilon\right)$$

$$\leq U_1 b_k + \frac{c_{10}}{2^k} \leq \frac{c_{11}(1 - 2\bar\eta)}{2^k} + \frac{c_{10}}{2^k},$$

where $c_{11} > 0$ is a positive constant. The first inequality holds by the triangular inequality, the last inequality holds by Lemma 4 property (a) and (b), and Proposition 4.

Summing (57) and (58) over $[T_k]$, and then we can upper bound (56) by

$$\mathbb{E}\left[\text{Reg}_k(\tilde{\mathbf{h}}; \mathcal{T}_{EE}, \varepsilon)\right] \leq \sum_{i=1}^{T_k} O(1) \cdot \frac{1}{2^k} + O(1) \cdot \frac{1 - 2\eta}{2^k}$$

$$= O(1) \cdot \frac{T_k}{2^k} + O(1) \cdot \frac{(1 - 2\bar\eta)T_k}{2^k}$$

$$= 2^k O\left(\frac{1}{(1 - 2\eta)^4}(\gamma + 1)d\ln d(\ln T)^2\right).$$

Since $T = |\mathcal{T}_{PE}| + |\mathcal{T}_{EE}|$ and $|\mathcal{T}_{PE}| = O(d\ln d(\ln T)^2)$, then $|\mathcal{T}_{EE}| = O(T)$. Also, the exploration-exploitation phase corresponds to all cycles run in Algorithm 4, hence $2\sum_{k=1}^K T_k = |\mathcal{T}_{EE}| = O(T)$. By $T_k = 4^k O\left(\frac{1}{(1-2\eta)^4}(\gamma + 1)d\ln d(\ln T)^2\right)$, we get the total number of batches as $K = \log_4\left(O\left(\frac{(1-2\eta)^4 T}{(\gamma+1)d\ln d(\ln T)^2}\right)\right)$. Then, we can upper bound the cumulative regret during the exploration-exploitation phase under "clean event" as

$$\mathbb{E}\left[\text{Reg}(\tilde{\mathbf{h}}; \mathcal{T}_{EE}, \varepsilon)\right] = \sum_{k=1}^K \mathbb{E}\left[\text{Reg}_k(\tilde{\mathbf{h}}; \mathcal{T}_{EE}, \varepsilon)\right]$$

$$= \sum_{k=1}^{\log_4\left(O\left(\frac{(1-2\eta)^4 T}{(\gamma+1)d\ln d(\ln T)^2}\right)\right)} 2^k O\left(\frac{1}{(1 - 2\eta)^4}(\gamma + 1)d\ln d(\ln T)^2\right) \tag{59}$$

$$= O\left(\frac{1}{(1 - 2\eta)^2}\sqrt{(\gamma + 1)d\ln dT}\ln T\right).$$

Finally, we upper bound the third term in (54) as

$$\mathbb{E}\left[\text{Reg}\left(\tilde{\mathbf{h}}; \mathcal{T}_{EE}, \bar\varepsilon\right)\right] = \sum_{t \in \mathcal{T}_{EE}} \mathbb{E}\left[\left(\text{Err}(\tilde{h}_t) - \text{Err}(\tilde{h}^\star)\right)\mathbb{I}\left(\bar\varepsilon\right)\right]$$

$$\leq \mathbb{E}\left[\sum_{t \in \mathcal{T}_{EE}} \mathbb{I}\left(\bar\varepsilon\right)\right] = |\mathcal{T}_{EE}|\mathbb{P}\left(\bar\varepsilon\right) \leq T \cdot \frac{6}{T} = 6. \tag{60}$$

By combining the upper bounds of (55), (59), and (60), we finish the proof. $\qquad \square$

## A.7 Proofs for Appendix A.1

Before proving Lemma 3, we need to first prove some intermediate lemmas. In the following lemma, we show that any log-concave distributed random vector with zero mean and positive definite covariance matrix can be linearly transformed into a new random vector with an isotropic log-concave distribution.

**Lemma 24.** *For any random vector $\mathbf{x} \in \mathbb{R}^d$ that follows a log-concave distribution with $\mathbb{E}[\mathbf{x}] = \mathbf{0}$ and $\mathbb{E}[\mathbf{x}\mathbf{x}^T] = \Sigma$, where $\Sigma$ is positive definite, the transformed random vector $\mathbf{z} = \Sigma^{-\frac{1}{2}}\mathbf{x}$ that follows an isotropic log-concave distribution.*

*Proof.* We first prove that $\mathbf{z} = \Sigma^{-\frac{1}{2}}\mathbf{x}$ is isotropic. Since $\mathbb{E}[\mathbf{x}\mathbf{x}^T] = \Sigma$, $\Sigma$ is positive definite, then $\mathbb{E}[\mathbf{z}\mathbf{z}^T] = \Sigma^{-\frac{1}{2}}\mathbb{E}[\mathbf{x}\mathbf{x}^T]\left(\Sigma^{-\frac{1}{2}}\right)^T = \Sigma^{-\frac{1}{2}}\Sigma\left(\Sigma^{-\frac{1}{2}}\right)^T = I$. Also, since $\mathbb{E}[\mathbf{x}] = \mathbf{0}$, then $\mathbb{E}[\mathbf{z}] = \Sigma^{-\frac{1}{2}}\mathbb{E}[\mathbf{x}] = \mathbf{0}$.

Next, we show that the probability density function of $\mathbf{z}$ is log-concave. Suppose the corresponding probability density functions of $\mathbf{x}$ and $\mathbf{z}$ are $\phi_{\mathbf{x}}(\cdot)$ are $\phi_{\mathbf{z}}(\cdot)$, respectively. Since $\phi_{\mathbf{x}}(\cdot)$ is log-concave, then for $\forall \alpha \in [0,1]$ and $\forall \mathbf{x}_1, \mathbf{x}_2 \in \mathbb{R}^d$,

$$\alpha \ln(\phi_{\mathbf{x}}(\mathbf{x}_1)) + (1-\alpha)\ln(\phi_{\mathbf{x}}(\mathbf{x}_2)) \leq \ln(\phi_{\mathbf{x}}(\alpha\mathbf{x}_1 + (1-\alpha)\mathbf{x}_2)).$$

Then, for $\forall \alpha \in [0,1], \mathbf{z}_1, \mathbf{z}_2 \in \mathbb{R}^d$,

$$
\begin{aligned}
&\alpha \ln((\phi_{\mathbf{z}}(\mathbf{z}_1))) + (1-\alpha)\ln((\phi_{\mathbf{z}}(\mathbf{z}_2)))\\
&= \alpha \ln\left(\det\left(\Sigma^{\frac{1}{2}}\right)\phi_{\mathbf{x}}\left(\Sigma^{\frac{1}{2}}\mathbf{z}_1\right)\right) + (1-\alpha)\ln\left(\det\left(\Sigma^{\frac{1}{2}}\right)\phi_{\mathbf{x}}\left(\Sigma^{\frac{1}{2}}\mathbf{z}_2\right)\right)\\
&= \alpha \ln\left(\phi_{\mathbf{x}}\left(\Sigma^{\frac{1}{2}}\mathbf{z}_1\right)\right) + (1-\alpha)\ln\left(\phi_{\mathbf{x}}\left(\Sigma^{\frac{1}{2}}\mathbf{z}_2\right)\right) + \ln\left(\det\left(\Sigma^{\frac{1}{2}}\right)\right)\\
&\leq \ln\left(\phi_{\mathbf{x}}\left(\alpha\Sigma^{\frac{1}{2}}\mathbf{z}_1 + (1-\alpha)\Sigma^{\frac{1}{2}}\mathbf{z}_2\right)\right) + \ln\left(\det\left(\Sigma^{\frac{1}{2}}\right)\right)\\
&= \ln(\phi_{\mathbf{z}}(\alpha\mathbf{z}_1 + (1-\alpha)\mathbf{z}_2)).
\end{aligned}
$$

Thus, $\phi_{\mathbf{z}}(\cdot)$ is isotropic log-concave. $\qquad\square$

The next lemma outlines the relationship between the eigenvalues of the covariance matrices for a high-dimensional random variable before and after it is projected onto a lower-dimensional subspace.

**Lemma 25.** *Let $\mathbf{x}$ be an arbitrary $d$-dimensional random variable with a positive definite covariance matrix $\Sigma$, whose maximum and minimum eigenvalues are $\bar{\lambda}$ and $\underline{\lambda}$, respectively. Let $V_{d'}$ an arbitrary $d'$-dimensional subspace with $d' \leq d$. Let $\mathbf{x}_{V_{d'}}$ denote the projection of $\mathbf{x}$ onto $V_{d'}$ with covariance matrix $\Sigma_{V_{d'}}$ whose maximum and minimum eigenvalues are $\bar{\lambda}_{V_{d'}}$ and $\underline{\lambda}_{V_{d'}}$, respectively. Then, $\bar{\lambda}_{V_{d'}} \leq \bar{\lambda}$ and $\underline{\lambda}_{V_{d'}} \geq \underline{\lambda}$.*

*Proof.* Let $P \in \mathbb{R}^{d \times d'}$ denote the projection matrix of $\mathbf{x}$, *i.e.*, $P^T P = I$ and $P^T \mathbf{x} = \mathbf{x}_{V_{d'}}$, then, $\Sigma_{V_{d'}} = P^T \Sigma P$. Hence, by definition of maximum eigenvalue, we have

$$
\begin{aligned}
\bar{\lambda}_{V_{d'}} &= \max_{\mathbf{v}\neq\mathbf{0},\,\mathbf{v}\in\mathbb{R}^{d'}} \frac{\mathbf{v}^T \Sigma_{V_{d'}}\mathbf{v}}{\mathbf{v}^T\mathbf{v}} = \max_{\mathbf{v}\neq\mathbf{0},\,\mathbf{v}\in\mathbb{R}^{d'}} \frac{\mathbf{v}^T P^T \Sigma P\mathbf{v}}{\mathbf{v}^T\mathbf{v}}\\
&= \max_{\mathbf{v}\neq\mathbf{0},\,\mathbf{v}\in\mathbb{R}^{d'}} \frac{\mathbf{v}^T P^T \Sigma P\mathbf{v}}{\mathbf{v}^T P^T P\mathbf{v}} \leq \max_{\mathbf{u}\neq\mathbf{0},\,\mathbf{u}\in\mathbb{R}^{d}} \frac{\mathbf{u}^T \Sigma\mathbf{u}}{\mathbf{u}^T\mathbf{u}} = \bar{\lambda},
\end{aligned}
$$

where $\mathbf{u} = P\mathbf{v}$.

$$
\begin{aligned}
\underline{\lambda}_{V_{d'}} &= \min_{\mathbf{v}\neq\mathbf{0},\,\mathbf{v}\in\mathbb{R}^{d'}} \frac{\mathbf{v}^T \Sigma_{V_{d'}}\mathbf{v}}{\mathbf{v}^T\mathbf{v}} = \min_{\mathbf{v}\neq\mathbf{0},\,\mathbf{v}\in\mathbb{R}^{d'}} \frac{\mathbf{v}^T P^T \Sigma P\mathbf{v}}{\mathbf{v}^T\mathbf{v}}\\
&= \min_{\mathbf{v}\neq\mathbf{0},\,\mathbf{v}\in\mathbb{R}^{d'}} \frac{\mathbf{v}^T P^T \Sigma P\mathbf{v}}{\mathbf{v}^T P^T P\mathbf{v}} \geq \min_{\mathbf{u}\neq\mathbf{0},\,\mathbf{u}\in\mathbb{R}^{d}} \frac{\mathbf{u}^T \Sigma\mathbf{u}}{\mathbf{u}^T\mathbf{u}} = \underline{\lambda},
\end{aligned}
$$

where $\mathbf{u} = P\mathbf{v}$. $\qquad\square$

Now we are ready to prove Lemma 3.

**Lemma 3.** *Let $\mathbf{x} \in \mathbb{R}^d$ ($d \geq 2$) have zero mean and a log-concave distribution. Suppose the eigenvalues of its covariance matrix $\Sigma = \mathbb{E}\left[\mathbf{x}\mathbf{x}^\top\right]$ are all bounded within $[\underline{\lambda}, \overline{\lambda}]$ for some positive constants $\underline{\lambda}, \overline{\lambda}$, then the distribution of $\mathbf{x}$ satisfies the regularity conditions in assumption 2, with parameters $L_1 = \frac{\beta_1(1)}{\sqrt{\overline{\lambda}}}$, $L_2 = \frac{\beta_1(2)}{\overline{\lambda}}$, $R = \frac{1}{9}\sqrt{\underline{\lambda}}$, $U_1 = \frac{1}{\sqrt{\underline{\lambda}}}$, $U_2 = \frac{\beta_2(2)}{\underline{\lambda}}$, $\delta = \frac{\beta_3(2)}{\sqrt{\overline{\lambda}}}$, $Q = \sqrt{\overline{\lambda}}$ for $\beta_1(1), \beta_1(2), \beta_2(2), \beta_3(2)$ given in Lemma 2.*

*Proof for Lemma 3.* For arbitrary 1-dimensional subspace $V_1 \subseteq \mathbb{R}^1$ and 2-dimensional subspace $V_2 \subseteq \mathbb{R}^2$, let $\mathbf{x}_{V_1}$ and $\mathbf{x}_{V_2}$ denote the projected vectors on $V_1$ and $V_2$ with covariance matrices $\Sigma_{V_1}$ and $\Sigma_{V_2}$, respectively. Let $\overline{\lambda}_{V_1}$ and $\overline{\lambda}_{V_2}$ denote the maximum eigenvalues of $\Sigma_{V_1}$, $\Sigma_{V_2}$ and $\underline{\lambda}_{V_1}$ and $\underline{\lambda}_{V_2}$ denote the minimum eigenvalues of $\Sigma_{V_1}$, $\Sigma_{V_2}$, respectively. Then by Lemma 25, we have $\overline{\lambda} \geq \overline{\lambda}_{V_1}, \overline{\lambda} \geq \overline{\lambda}_{V_2}, \underline{\lambda} \leq \underline{\lambda}_{V_1}$ and $\underline{\lambda} \leq \underline{\lambda}_{V_2}$. Let $\mathbf{z}_{V_1} = \Sigma_{V_1}^{-\frac{1}{2}}\mathbf{x}_{V_1}$, $\mathbf{z}_{V_2} = \Sigma_{V_2}^{-\frac{1}{2}}\mathbf{x}_{V_2}$. Then, $\mathbf{z}_{V_1}$ and $\mathbf{z}_{V_2}$ have isotropic log-concave densities, denoted as $\phi_{\mathbf{z}_{V_1}}$ and $\phi_{\mathbf{z}_{V_2}}$, respectively.

We first determine $L_1$, $L_2$ and $R$ prescribed in Assumption 2, Condition 1. For all $\|\mathbf{x}_{V_1}\|_2 \leq \frac{1}{9}\sqrt{\underline{\lambda}}$, $\mathbf{x}_{V_1}$'s probability density function $\phi_{\mathbf{x}_{V_1}}(\cdot)$ satisfies

$$\phi_{\mathbf{x}_{V_1}}(\mathbf{x}_{V_1}) = \phi_{z_{V_1}}\left(\Sigma_{V_1}^{-\frac{1}{2}}\mathbf{x}_{V_1}\right) \det\left(\Sigma_{V_1}^{-\frac{1}{2}}\right) \geq \frac{\beta_1(1)}{\sqrt{\overline{\lambda}_{V_1}}} \geq \frac{\beta_1(1)}{\sqrt{\overline{\lambda}}}. \tag{61}$$

Now $\|\mathbf{z}_{V_1}\|_2 = \left\|\Sigma_{V_1}^{-\frac{1}{2}}\mathbf{x}_{V_1}\right\|_2 \leq \frac{1}{\sqrt{\underline{\lambda}_{V_1}}}\|\mathbf{x}_{V_1}\|_2 \leq \frac{1}{\sqrt{\underline{\lambda}}} \cdot \frac{1}{9}\sqrt{\underline{\lambda}} = \frac{1}{9}$. Then, the first inequality in (61) holds by Lemma 2, property (b) and $\det\left(\Sigma_{V_1}^{-\frac{1}{2}}\right) \geq \frac{1}{\sqrt{\overline{\lambda}_{V_1}}}$. The second inequality holds by Lemma 25.

Similarly, for all $\|\mathbf{x}_{V_2}\|_2 \leq \frac{1}{9}\sqrt{\underline{\lambda}}$, $\mathbf{x}_{V_2}$'s probability density function $\phi_{V_2}(\cdot)$ satisfies

$$\phi_{\mathbf{x}_{V_2}}(\mathbf{x}_{V_2}) = \phi_{z_{V_2}}\left(\Sigma_{V_2}^{-\frac{1}{2}}\mathbf{x}_{V_2}\right) \det\left(\Sigma_{V_2}^{-\frac{1}{2}}\right) \geq \frac{\beta_1(2)}{\overline{\lambda}_{V_2}} \geq \frac{\beta_1(2)}{\overline{\lambda}}. \tag{62}$$

Now $\|\mathbf{z}_{V_2}\|_2 = \left\|\Sigma_{V_2}^{-\frac{1}{2}}\mathbf{x}_{V_2}\right\|_2 \leq \frac{1}{\sqrt{\underline{\lambda}_{V_2}}}\|\mathbf{x}_{V_2}\|_2 \leq \frac{1}{\sqrt{\underline{\lambda}}} \cdot \frac{1}{9}\sqrt{\underline{\lambda}} = \frac{1}{9}$. Then, the first inequality in (62) holds by Lemma 2, property (b) and $\det\left(\Sigma_{V_2}^{-\frac{1}{2}}\right) \geq \frac{1}{\overline{\lambda}_{V_2}}$. The second inequality holds by Lemma 25.

Combining (61) and (62), we have $L_1 = \frac{\beta_1(1)}{\sqrt{\overline{\lambda}}}$, $L_2 = \frac{\beta_1(2)}{\overline{\lambda}}$, $R = \frac{1}{9}\sqrt{\underline{\lambda}}$.

Then, we determine $U_1, U_2$ and $\delta$ prescribed in Assumption 2, Condition 2. For $\phi_{\mathbf{x}_{V_1}}(\cdot)$, we have

$$\phi_{\mathbf{x}_{V_1}}(\mathbf{x}_{V_1}) = \phi_{z_{V_1}}\left(\Sigma_{V_1}^{-\frac{1}{2}}\mathbf{x}_{V_1}\right) \det\left(\Sigma_{V_1}^{-\frac{1}{2}}\right) \leq \frac{1}{\sqrt{\underline{\lambda}_{V_1}}} \leq \frac{1}{\sqrt{\underline{\lambda}}},$$

where the first the inequality holds by Lemma 2, property (c) and $\det\left(\Sigma_{V_1}^{-\frac{1}{2}}\right) \leq \frac{1}{\sqrt{\underline{\lambda}_{V_1}}}$. The second inequality holds by Lemma 25. Thus, $U_1 = \frac{1}{\sqrt{\underline{\lambda}}}$.

For $\phi_{\mathbf{x}_{V_2}}(\cdot)$, we have

$$\phi_{\mathbf{x}_{V_2}}(\mathbf{x}_{V_2}) = \phi_{z_{V_2}}\left(\Sigma_{V_2}^{-\frac{1}{2}}\mathbf{x}_{V_2}\right) \det\left(\Sigma_{V_2}^{-\frac{1}{2}}\right) \leq \frac{\beta_2(2)\exp\left(-\beta_3(2)\left\|\Sigma_{V_2}^{-\frac{1}{2}}\mathbf{x}_{V_2}\right\|_2\right)}{\underline{\lambda}_{V_2}}$$

$$\leq \frac{\beta_2(2)\exp\left(-\frac{\beta_3(2)}{\sqrt{\overline{\lambda}_{V_2}}}\|\mathbf{x}_{V_2}\|_2\right)}{\underline{\lambda}_{V_2}}$$

$$\leq \frac{\beta_2(2)\exp\left(-\frac{\beta_3(2)}{\sqrt{\overline{\lambda}}}\|\mathbf{x}_{V_1}\|_2\right)}{\underline{\lambda}},$$

where the first inequality holds by Lemma 2, property (d) and $\det\left(\Sigma_{V_2}^{-\frac{1}{2}}\right) \leq \frac{1}{\underline{\lambda}_{V_2}}$. The second inequality holds by $\left\|\Sigma_{V_2}^{-\frac{1}{2}}\mathbf{x}_{V_2}\right\|_2 \geq \frac{1}{\sqrt{\overline{\lambda}_{V_2}}}\|\mathbf{x}_{V_2}\|_2$. The last inequality holds by Lemma 25.

Thus, we have $U_2 = \frac{\beta_2(2)}{\underline{\lambda}}, \delta = \frac{\beta_3(2)}{\sqrt{\overline{\lambda}}}$.

Finally, we determine $Q$ prescribed in Assumption 2, Condition 3. For arbitrary $\mathbf{w} \in \mathbb{B}^d$, $\langle \mathbf{w}, \mathbf{x} \rangle$ forms a 1-dimensional random variable whose probability density function is log-concave. Let $\mathbf{x}_{V_1'} := \langle \mathbf{w}, \mathbf{x} \rangle$ denote the projected random variable and $\sigma_{V_1'}^2 = \mathbb{E}\left[\mathbf{x}_{V_1'}^T \mathbf{x}_{V_1'}\right]$ denote its variance. Then, by Lemma 2 property (a), for every $t > 0$,

$$\mathbb{P}\left(|\mathbf{z}| > t\right) \leq e^{-\sigma_{V_1'} t + 1} \leq e^{-\sqrt{\underline{\lambda}} t + 1},$$

where the second inequality holds by $\sigma_{V_1'}^2 \geq \underline{\lambda}_{V_1'} \geq \underline{\lambda}$. Thus $Q = \sqrt{\underline{\lambda}}$. $\qquad\square$

