# OpenReview forum: "Online Strategic Classification With Noise and Partial Feedback"
_NeurIPS.cc/2025/Conference — NeurIPS 2025 spotlight_

### Official Review · Reviewer_SQmo · 2025-06-30

**Clarity:** 2
**Significance:** 3
**Originality:** 2
**Rating:** 5
**Confidence:** 3

**Summary:**

This paper considers the strategic classification model with following extension: candidates arrive online and the principal only observes the manipulated features and the label only if the candidate was classified positively. Candidate features originate from an unknown distribution and are not exactly separable by a linear hyperplane - rather, the labels can be noisy wrt to the plane (Massert noise that is correlated with features). Like other works in the online strategic setting [Chen, 2020], they assume agents can manipulate within some radius. The key departure from existing online model is the observability of the labels conditioned on acceptance.

The key technical contribution of the work is providing an online algorithm that achieves $\sqrt{T}$ regret with respect to the offline optimal, which can be informally characterized as shifting the optimal non-strategic classifier by the width of the manipulation radius. The algorithm composed of three parts: an initialization phase where the principal positively classifies both sides of the plane (on every other step) to collect data. A refinement stage that explores a boundary near the initialization phase - both of these can incur large regret and thus runs for log(T) time. Lastly, the large part of the algo runs where the enhancement is over batches and proxy features are used to approximate the gradient (in lieu of costly exploration).

**Questions:**

- Do you have any thoughts on the setting where each agent's manipulation radius $\gamma$ is different? So one can either think of this as being an unknown (perhaps too hard) or belonging to some range. I know prior literature also makes this assumption -- so not a major criticism -- but I am curious about it since afaik the clairvoyant classifier in this setting can no longer be easily characterized.

- Could you explain why in the refinement stage, why we need to consider strategic manipulation and form a linear optimization problem. My understanding is that we set a threshold b and agents within b margin (according to true features) of the current classifier are positively classified. Can this not be achieved by shifting the boundary by \gamma?

**Ethical Concerns:**

["NO or VERY MINOR ethics concerns only"]

**Final Justification:**

I had a positive view of this work and the author rebuttal maintained this perspective.

**Limitations:**

I don't see any negative social impact of this work.

**Quality:**

3

**Strengths And Weaknesses:**

With strategic classification paper, I find the model to be crucial and I find the model here to be fairly reasonable and well justified. Indeed in many settings only positive labels are observed for their true label. While this is the main difference from existing works, I don't think it's a negative as it does lead to a meaningful/challenging problem. I also found the paper to be focused in answering the crucial question of sub-linear regret. Overall, I think it's a good paper.

My main criticism would be in the exposition of the paper - especially in the theory section. For example, in the initialization part of the algorithm, it is not clear what the significance of the acute angle perspective is. Some diagrams or high-level description of the main ideas would help for the first and second components of the algorithm.

---

> ### Author Rebuttal · Authors · 2025-07-31
>
> We sincerely thank you for finding our model crucial, reasonable and well-justified and acknowledging that our paper is good. We also appreciate your thoughtful comments. We provide our feedback as follows.
>
> - Explanation for the significance of each algorithm:
>
> We agree that we should explain the sub-algorithms in more detail. Here let us provide further explanations for the significance of each sub-algorithm below, which we also plan to add to our manuscript.
>
> Initialization Algorithm. The Initialization Algorithm aims to output an initial vector $\bar w_0$ such that the inner product of $\bar w_0$ and the coefficient vector of the optimal classifier $w^\star$ is lower bounded by a positive constant ($\langle w^\star, \bar w_0 \rangle\ge c_1(1-2\bar\eta)>0$) with high probability (notably, we want to clarify that the angle between $\bar w_0$ and $w^\star$ should be strictly less than $\pi/2$, not just less than or equal to $\pi/2$). This guarantee is crucial for the subsequent Refinement stage. The Refinement Algorithm relies on the localization technique, which in turn requires that the hyperplane deployed in every round remains close to the optimal one. However, since $w^\star$ is actually unavailable, for each coefficient vector $w_{0,i}$ in Refinement, we cannot measure $\langle w^\star , w_{0,i}\rangle$ directly. Instead, we constrain every $w_{0,i}$ to lie in the set $\mathcal{K}_0 = \{w|\ \langle w, \bar w_0 \rangle\ge c_1(1-2\bar\eta)\}$, which, by construction, contains $w^\star$ with high probability. Thus, every hyperplane used in Refinement remains provably close to the optimal one, enabling effective localization without ever seeing $w^\star$.
>
> Refinement Algorithm. The Refinement Algorithm aims to output a vector $w_1$ such that the angle between $w_1$ and the coefficient vector of the optimal classifier $w^\star$ is less than or equal to $\pi/4$ with high probability. This guarantee is important for the subsequent Enhancement stage. The Enhancement Algorithm relies on the effective construction of proxy features, which in turn requires that the angle between the coefficient vectors $w_{k,i}$ deployed in each iteration always forms an acute angle with $w^\star$ ($\theta(w_{k,i},w^\star)\le \pi/2$). Still, since $w^\star$ is actually unavailable, we cannot directly measure $\theta(w_{1,i},w^\star)$ in batch 1. Instead, we constrain every $w_{1,i}$ to lie in the set $\mathcal{K} _ 1 = \{ w |\ \langle w,  w _ 1 \rangle \ge \cos (\pi / 4) \}$, which is equivalent to $\theta(w_{1,i},w_1)\le \pi/4$ for all $i\in[T_1]$. Then, by the triangular inequality of angles as we have proved in Appendix A.3, Lemma 12, if $\theta(w_1,w^\star)\le \pi/4$ and $\theta(w_1,w_{1, i})\le \pi/4$, then $\theta(w_{1,i},w^\star)\le \pi/2$ for all  $i\in[T_1]$, enabling the construction of proxy feature.
>
> Enhancement Algorithm. During the Batched Enhancement phase, each batch $k$ receives an initial vector $w_{k}$ from the last batch, which is required to satisfy $\theta(w^\star,w_{k})\le \pi/2^{k+1}$. Next, in each iteration $i$ of batch $k$, we constrain the deployed coefficient vector $w_{k,i}$ to lie in the set  $\mathcal{K} _ k = \{ w \mid \langle w, w _ k \rangle \ge \cos \left( \pi / 2^{k+1} \right) \}$, which means $\theta(w_{k,i},w_k)\le \pi/2^{k+1}$ for all $i\in[T_k]$. Then, by the triangular inequality of angles as we have proved in Appendix A.3, Lemma 12, we can show that $\theta(w^\star,w_{k,i})\le \theta(w^\star,w_{k})+\theta(w_{k,i},w_{k})\le \pi/2^{k}$. This statement is quite critical: On one hand, it controls the expected cumulative error in batch $k$ to be $O(\frac{1}{2^k}\cdot T_k)$. On the other hand, the condition that $\theta(w_{k,i},w_k)\le \pi/2^{k}$, together with our localized online mirror descent method, ensures that batch $k$ outputs a vector $w_{k+1}$ that satisfies $\theta(w^\star,w_{k+1})\le \pi/2^{k+2}$ with high probability, which is required by the next batch.
>
> - Discussions on unknown or heterogeneous maximum manipulation radius $\gamma$:
>
> In our paper we consider the setting that the manipulation radius (or manipulation budget/maximum manipulation distance) $\gamma$ is known, which is a common practice in the literature (Chen et al. (2020), Shen et al. (2024)). However, considering unknown or heterogeneous $\gamma$ is a quite interesting direction of future study.
>
>
> For a homogeneous unknown $\gamma$,
> Ahmadi et al. (2021) proposed a strategic perceptron algorithm with an unknown manipulation radius. Their key idea is running a binary search to find a proper estimate on this radius. However, this idea relies heavily on their assumption that the positive and negative classes are strictly separated by a margin without any noise. Their algorithm design requires either knowing the value of the margin or having an estimation process for it. These requirements cannot be fulfilled in the noisy setting without any margin. Therefore, it is hard to extend Ahmadi et al. (2021)'s binary search approach to our setting with unknown $\gamma$.
>
>
> If the maximum manipulation distance is heterogeneous but known, then adapting our algorithm seems straightforward. That is, each agent arriving at time $t$ has a maximum manipulation distance $\gamma_t$ that potentially differs from other agents' manipulation distances, but every $\gamma_t$ is known to the principal prior to decision-making. In this case, our construction of proxy feature and pairwise contrastive inference can directly use the known $\gamma_t$   for each agent, in place of the previous homogeneous $\gamma$.
>
> - Explanations on the Refinement stage:
>
> As we have explained at the beginning, we need the Refinement Algorithm to output a vector $w_1$ such that $\theta(w_1,w^\star)\le \pi/4$ with high probability, which is crucial for the subsequent Enhancement Algorithm.
> To achieve this, we form an online linear optimization problem with localization following Zhang et al. (2020). Zhang et al. (2020) show that in the non-strategic setting, this approach can effectively learn halfspaces under Massart noise. Our refinement stage applies classifiers as if there is no manipulation and focuses on the localization region right above the classification hyperplane. Agents within the localization region are naturally positively classified so they would truthfully report their features. Therefore, we can  collect true data within the localization region, enabling the approach in Zhang et al. (2020) with theoretical guarantees. However, although this can collect true data within the localization region, it is fairly costly. This is because unqualified agents below the classification hyperplanes are incentivized to manipulate their features to get positively classified, resulting in high instantaneous regret. Fortunately, the total regret in the Refinement stage is controlled because this stage lasts for only $O(\log T)$ cycles.
>
> However, this cannot be achieved by simply shifting the boundary by $\gamma$. Although shifting the boundary by $\gamma$ can prevent excessive manipulation from unqualified agents, it also causes challenges for targeting the localization region in terms of the true features. In this case, all agents with true features within $\gamma$ distance below the announced classification  hyperplane will strategically manipulate their features onto the classification hyperplane (see Figure 1(a) in our paper). These manipulating agents consist of both agents whose true features are within the localization band and agents whose true features are outside the band, and their reported features all lie on classification hyperplane. So it is difficult to tell whose true features are indeed within the localization band. Our Enhancement algorithm manages to resolve this issue by combining proxy features and pairwise contrastive inference.
>
> We appreciate your insightful questions and suggestions. We will integrate our discussions into the camera-ready version to clarify the points you have raised.
>
> Chen, Y., Liu, Y., & Podimata, C. (2020). Learning strategy-aware linear classifiers. Advances in Neural Information Processing Systems, 33, 15265-15276.
>
> Shen, L., Ho-Nguyen, N., Giang-Tran, K. H., & Kılınç-Karzan, F. (2024). Mistake, manipulation and margin guarantees in online strategic classification. arXiv preprint arXiv:2403.18176.
>
> Ahmadi, S., Beyhaghi, H., Blum, A., & Naggita, K. (2021). The strategic perceptron. In Proceedings of the 22nd ACM Conference on Economics and Computation (pp. 6-25).
>
> Zhang, C., Shen, J., & Awasthi, P. (2020). Efficient active learning of sparse halfspaces with arbitrary bounded noise. Advances in Neural Information Processing Systems, 33, 7184-7197.

---

> > ### Comment · Reviewer_SQmo · 2025-08-03
> >
> > Thank your for your detailed response and I remain convinced this is a good paper.

---

> > > ### Author Response · Authors · 2025-08-04
> > >
> > > Thank you for your insightful questions and suggestions. We will integrate our discussions into the camera-ready version to clarify the points you have raised. We are particularly grateful for your inspiring question regarding the unknown or heterogeneous maximum manipulation radius $\gamma$, as it has prompted us to explore an exciting new future research direction.

---

### Official Review · Reviewer_pYbv · 2025-06-30

**Clarity:** 4
**Significance:** 3
**Originality:** 3
**Rating:** 5
**Confidence:** 3

**Summary:**

The paper studies online binary linear classification when agents strategically manipulate their features to achieve positive prediction at quadratic cost, and the learner only sees labels of positively classified agents. The authors assume that the true feature-label relationship is a halfspace model subject to arbitrary but bounded (Massart) noise. Additionally, the distribution of features is assumed to satisfy regularity conditions for 1-dimensional and 2-dimensional projections, including lower and upper density bounds and a sub-exponential tail bound for the inner product of features with any unit vector. This assumption appears in prior works on learning halfspaces with noise.

The proposed algorithm consists of three phases: initialization, refinement, and batched enhancement, utilizing proxy features and pairwise contrastive inference. A regret bound of $\tilde O(\sqrt T​)$ is provided. Numerical experiments support these theoretical findings.

**Questions:**

1. Do you have any intuition for how your algorithm or regret bounds might change under other noise settings?
2. Do you have an idea on how to extend the partial feedback setting to more complex classes, e.g., finite VC classes (even without noise)?
3. Are there any guarantees if the budget parameter $\gamma$ is unknown or must be estimated online?
4. Any guarantees on more general cost functions (e.g., linear, or other convex costs)?

**Ethical Concerns:**

["NO or VERY MINOR ethics concerns only"]

**Final Justification:**

I have read the rebuttal and still think the paper should be accepted.

**Limitations:**

yes

**Paper Formatting Concerns:**

The title of the paper in openreview [Online Strategic Classification with Noise] is not the same as the pdf [Online Strategic Classification with Noise and Partial Feedback], but this could be easily fixed.

**Quality:**

4

**Strengths And Weaknesses:**

*Strengths*

- The problem addressed, online strategic classification with partial feedback, is interesting.
- The inclusion of Massart noise makes a great addition, and demonstrates the generality of the algorithm.
- The way the paper handles the challenges of strategic feature manipulation, partial feedback, and label noise is a central contribution. The specific techniques used, constructing proxy features and using pairwise contrastive inference, are novel aspects of their proposed algorithm.  In particular, for gradient estimation without true features, gradient estimates over bands around the boundary are needed. Yet, agents in those bands misreport, so the authors (i) construct proxy feature points from the reports and (ii) use pairwise contrastive inference with two shifted classifiers to unbias these estimates.
- The paper is well-written.

*Weaknesses*
- Usually, in online strategic classification, adversarial settings are considered, whereas this paper makes some distributional assumptions. That said, this setting is also interesting.
- Only quadratic cost is considered.

*Comments*
- The paper mentions [Ahmadi et al., 2023, 2024], but it does not mention a closely related work by Cohen et al. 2024 [1], which is chronologically "in between", nor a more recent, subsequent relevant work by Shao et al. 2025 [2].
- The paper provides overviews and discussions of the Initialization, Refinement, and Batched Enhancement algorithms. However, I suggest that the authors provide even more discussion about them, particularly given the complexity of the methods.


[1] Lee Cohen, Yishay Mansour, Shay Moran, and Han Shao. Learnability gaps of strategic classification. COLT 2024.

[2] Han Shao, Shuo Xie, Kunhe Yang. Should Decision-Makers Reveal Classifiers in Online Strategic Classification? ICML 2025.

---

> ### Author Rebuttal · Authors · 2025-07-31
>
> We are grateful that you find our work interesting, novel and well-written. We also sincerely thank you for your insightful questions. We provide our feedback as follows.
>
> - Citations:
>
>  Thank you for providing the two related papers. We will add these two papers to our ''Related Literature'' section.
>
> - More discussions on each algorithm:
>
>  We agree that we should add more discussions on the algorithms. Due to space constraints, we provide our explanations for the significance of each sub-algorithm in the answers to Reviewer SQmo, and plan to update our manuscript accordingly.
>
> - Intuitions for how the algorithm and bound will change under other noises:
>
> Here we provide some thoughts on online strategic classification under adversarial noise (the nature confines the overall error rate of the `best' halfspace to be at most $\bar\eta$, but samples may not necessarily be i.i.d.) and malicious noise (the nature randomly draws an instance from the underlying feature-label distribution with a fixed probability $1-\bar\eta$ while returning an arbitrary feature-label pair with probability $\bar\eta$). We conjecture we could tackle these noises again based on our proposed idea of localization, proxy feature and contrastive inference, with a slight difference in the design of loss function and the gradient. Their regret bounds are also expected to be similar to ours.
>
> Here is our rough intuition. Shen (2021, 2023)
> adapts the algorithm in Zhang et al. (2020) to learning halfspaces under these alternative types of noises (in non-strategic and full-feedback setting).
> The resulting algorithms again use online mirror descents with "localization" and their theoretical guarantees under these alternative noises are similar to that in Zhang et al. (2020) under Massart noise. Moreover, the type of noise does not appear to influence our construction of proxy features and contrastive inference. Therefore, we conjecture that we can adapt our algorithm to these alternative noises while attaining similar theoretical guarantees. We leave this extension for future study.
>
> - Extending the partial feedback setting to more complex classes:
>
> In our setting, we use the contrastive inference method to tackle the partial feedback setting. By contrasting positively-classified agents' responses under two slightly different classifiers, we can fine-tune our classifier and simultaneously maintain a relatively high accuracy of our declared classifiers. However, this method relies on the clear characterization of agents' best responses and a gradient-based optimization algorithm to update the classifier, both of which depend on the linear structure of classifiers. When adapting to more general settings, it may be hard to characterize the agent's strategic manipulation and directions for updating the classifiers in a closed form.  These challenges may prevent us from directly extending our setting to more complex classes.
>
>   In addition, as you noted, online strategic classification problems within finite VC classes are typically examined under adversarial settings with full feedback. In particular, their approaches often involve discretizing the  hypothesis class into multiple experts and then implementing a weighted-majority voting scheme across these experts (see, e.g., Ahmadi et al., 2024; Cohen et al., 2024). However, this cannot be extended to partial label observation setting, either. Specifically, in the adversarial setting with partial feedback, nature can hurt the principal's learning algorithm by keeping generating agent examples that will be classified as negative. This time the principal will never observe the agents' true label due to partial observation, and thus has no chance to update the classifier.
>
> - Setting with unknown budget $\gamma$:
>
> In our paper we consider the setting that the manipulation budget (or manipulation radius/maximum manipulation distance) $\gamma$ is known, which is a common practice in the literature (Chen et al. (2020), Shen et al. (2024)). However, considering unknown $\gamma$ is a quite interesting direction of  future study. Ahmadi et al. (2021) proposed a strategic perceptron algorithm with unknown manipulation budget. Their key idea is running a binary search to find a proper estimate on this budget. However, this idea relies heavily on their assumption that the positive and negative classes are strictly separated by a margin without any noise. Their algorithm design requires either knowing the value of margin having an estimation process on it. These requirements cannot be fulfilled in the noisy setting without any margin. Therefore, it is hard to extend their binary search approach to our setting with unknown $\gamma$.
>
> - Guarantees on more general cost functions:
>
> Our work can be easily adapted to any norm-formed manipulation cost with the regret guarantee unchanged. The way for adaptation is similar to that by Shen et al. (2024). In this setting, given a classifier $\tilde h(r)=\text{sgn}(\langle w,r\rangle+m)$ and agent manipulation cost $\text{Cost}(x,r)=\frac{2}{\gamma}\|x-r\|$ for any norm $\|\cdot\|$, an agent with true feature $x$ will report $r^\star(x,\tilde h)$ that satisfies  $r ^ \star (x , \tilde h ) = x -\frac { \langle w , x \rangle+m } { \|w\| _ \star } v(w) $ if
> $-\gamma\|w\| _ * \le \langle w , x \rangle + m < 0 $; and otherwise, $r ^ \star (x , \tilde h ) = x$,  where $ \|\cdot\| _ * $ denotes the dual norm of $\|\cdot\|$, and $v(w)\in \arg \max_{\|v\|\le 1}\langle w,v\rangle$. From the above expression we know that when extending the $\ell _ 2 $ norm cost to general norm cost, the key modification is changing agents' manipulation direction (from $w$ under $\ell_2$ norm  to $v(w) / \|w\| _ * $ under general norm) and converting the distance under the general norm to that under $\ell_2$ norm (multiplying a $\|w\|_*$ factor on the general norm-based distance). Therefore, when designing proxy features and contrastive inference, we can accordingly change the projection direction and adjust the offset. These modifications do not change our key idea of algorithm design.
>
> - The title:
>
> Our final title is "Online Strategic Classification with Noise and Partial Feedback". We will finalize this in our camera-ready version.
>
> Thanks for your interesting questions about potential extensions of our work. We will add these as our future work direction to the manuscript.
>
> Shen, J. (2021). On the power of localized perceptron for label-optimal learning of halfspaces with adversarial noise. In International Conference on Machine Learning (pp. 9503-9514). PMLR.
>
> Shen, J. (2023). PAC learning of halfspaces with malicious noise in nearly linear time. In International Conference on Artificial Intelligence and Statistics (pp. 30-46). PMLR.
>
> Zhang, C., Shen, J., & Awasthi, P. (2020). Efficient active learning of sparse halfspaces with arbitrary bounded noise. Advances in Neural Information Processing Systems, 33, 7184-7197.
>
> Ahmadi, S., Yang, K., & Zhang, H. (2024). Strategic littlestone dimension: Improved bounds on online strategic classification. Advances in Neural Information Processing Systems, 37, 101696-101724.
>
> Cohen, L., Mansour, Y., Moran, S., & Shao, H. (2024). Learnability gaps of strategic classification. In The Thirty Seventh Annual Conference on Learning Theory (pp. 1223-1259). PMLR.
>
> Chen, Y., Liu, Y., & Podimata, C. (2020). Learning strategy-aware linear classifiers. Advances in Neural Information Processing Systems, 33, 15265-15276.
>
> Shen, L., Ho-Nguyen, N., Giang-Tran, K. H., & Kılınç-Karzan, F. (2024). Mistake, manipulation and margin guarantees in online strategic classification. arXiv preprint arXiv:2403.18176.
>
> Ahmadi, S., Beyhaghi, H., Blum, A., & Naggita, K. (2021). The strategic perceptron. In Proceedings of the 22nd ACM Conference on Economics and Computation (pp. 6-25).

---

> > ### Comment · Reviewer_pYbv · 2025-07-31
> >
> > Thank you for your thorough and careful revisions. You have satisfactorily addressed all of my concerns. I therefore confirm my original recommendation to accept the paper.

---

> > > ### Author Response · Authors · 2025-08-01
> > >
> > > We would like to sincerely thank you once again for your helpful comments and for drawing our attention to the related literature. We will make corresponding revisions to our manuscript in the camera-ready version. We also appreciate your valuable advice and inspiring questions, which will be of great help to us in improving the work.

---

### Official Review · Reviewer_SBKe · 2025-07-02

**Clarity:** 4
**Significance:** 4
**Originality:** 4
**Rating:** 5
**Confidence:** 4

**Summary:**

This paper explores binary classification in a strategic online setting, where a classifier must immediately label incoming agents. Crucially, it receives noisy feedback only for positively classified agents ("apple tasting"). The feature-label relationship uses a linear half-space model with bounded noise, and the classifier aims to minimize regret against the best fixed omniscient classifier.

Following prior work, the authors first show that for half-spaces, agents' best responses can be characterized, proving that an omniscient classifier yields identical outputs whether features are true or manipulated.

They then propose a three-component algorithm. First, an "exploration" subroutine uses contrasting classifiers to find an initial normal vector close to the optimal one. Second, a "localization" subroutine refines this vector by focusing on informative data near the classifier's boundary, where true features are observable due to no manipulation incentives. Despite new complications from strategic reporting, they solve an online linear optimization problem using techniques akin to learning with label noise, achieving only logarithmic regret in this phase.

The third and most novel component tackles strategic manipulation in this refined localization. As manipulable classifiers lead to high regret from misreporting, they introduce a modified classifier that sets a higher bar for positive classification. To address the unobservability of true features in key regions, they develop proxy features to mimic true ones for gradient construction, and pairwise contrastive inference. The latter uses two successive classifiers to statistically infer information about unobservable true data in critical localized regions, and which in turn enables estimation of gradients.

**Questions:**

If I understand correctly, our results hold for a sequence of myopic buyers. I am wondering whether you had some thoughts regarding a repeated game where a single agent aims to maximize the sum of positively classified points (over cycles). I guess that this can be modeled as a Stackelberg game and that it is also a consideration for the model without noise and with two-sided feedback.

**Ethical Concerns:**

["NO or VERY MINOR ethics concerns only"]

**Final Justification:**

Thank you for the response to all of my questions and comments. They explain everything I asked clearly and made me understand even better where the additional challenges in future work lie. I, thus, confirm my score.

**Limitations:**

Yes

**Paper Formatting Concerns:**

No formatting issues noticed

**Quality:**

4

**Strengths And Weaknesses:**

This paper tackles a tougher version of online strategic classification, to accommodate partial, noisy feedback. A key and novel strength of the paper is the development of pairwise contrastive inference to address strategic reporting from the agents. In particular, by employing two slightly different classifiers in succession and contrasting the observed data from each, they cleverly infer statistical information about the true, unobservable features in critical regions. Moreover, the integration of proxy features, designed to mimic true features from reported ones, further strengthens their approach by allowing the construction of effective proxy gradients. In general, I appreciated the technical novelty the authors developed in this paper.

Even the “localization” procedure, which has already been developed for the non-strategic counterpart of the problem, is here devised in a clever way. Namely, it handles strategic interaction via an instantiation of Online Mirror Descent and suffers only logarithmic regret.

The exposition is clear and, despite its technical depth, the treatment maintains conceptual clarity at all points of the main body. The authors explicitly explain what the algorithm subroutines are needed for, and where each piece of the analysis comes into play.

Overall, the contributions and presentation are strong, novel and well-conveyed. The authors design a single algorithm that simultaneously handles strategic feature reports and partial, noisy feedback, suffering $\sqrt{T}$ regret, essentially providing conclusive answers for the analyzed setting.

There is a typo in line 168: inccurs —> incurs.

---

> ### Author Rebuttal · Authors · 2025-07-31
>
> We are grateful that you find our contributions and presentation ''strong, novel and well-conveyed''. We also sincerely thank you for your thought-provoking question. Our response is as follows.
>
> - Repeated games with a single and non-myopic agent:
>
> Our work indeed focuses on a sequence of myopic agents, who always follow the best response to the declared classifier. Specifically, agents with true features within $\gamma$ distance below the classification hyperplane will project their true features onto the hyperplane as their reported feature, while other agents will report truthfully (Lemma 1). Our algorithm heavily leverages the structure of this best response, both in the proxy feature construction and pairwise contrastive inference. The learning problem becomes much more difficult when the principal repeatedly interacts with the same agent. This is because the agent may have complicated forward-looking behaviors that drastically deviate from the myopic best response described in our Lemma 1. In particular, the agent may report features that do not appear immediately optimal to mislead the principal's learning algorithm, thereby securing future advantages. Under the noisy setting, the agents' feature manipulation direction may be even opposite to the myopic optimal direction, slowing down the learning of the optimal classifier. While one may model the agent's behaviors by a dynamic programming (DP) model, analyzing the DP model and developing effective learning algorithms appear much more challenging. This is a totally different but very interesting problem that warrants future study.
>
> - We also thank you for pointing out our typo. We have corrected it in our paper.

---

> ### Comment · Reviewer_SBKe · 2025-08-05
>
> Thank you for the response to all of my questions and comments. They explain everything I asked clearly and made me understand even better where the additional challenges in future work lie. I, thus, confirm my score.

---

> > ### Author Response · Authors · 2025-08-05
> >
> > Thank you very much for the inspiring question on extending the model to a setting involving a single and non-myopic agent. This is truly an interesting problem for future research. In our revision, we will also correct the typo you pointed out. We greatly appreciate your time and valuable feedback. Thank you again.

---

### Official Review · Reviewer_Q4JY · 2025-07-06

**Clarity:** 3
**Significance:** 3
**Originality:** 2
**Rating:** 5
**Confidence:** 4

**Summary:**

They study online strategic classification(SC) with massart noise and a partial feedback model (only receive feedback when the example is labeled as positive). In online strategic classification, in each round the principal deploys a classifier and then an agent in round $t$ arrives. The agent with feature set $x$ best-responds to the deployed classifier in order to maximize their utility, that is the classification that they receive in their manipulated state $x'$ minus the cost of manipulation, i.e., movement from $x$ to $x'$. Usually in SC, the true label is revealed after the principal announces their classification based on the deployed classifier; however, here they assume partial feedback, that is the true label is only revealed if the agent is classified as positive. For example, if a student is admitted in the college, after that the principal would see if her decision was good or not (the student was actually a positive  example (good student) or not.)

 Massart noise is when the actual label of the points are flipped with some feature-dependent probability $\eta(x)$, where all values $\eta(x)$ are bounded (at most $\eta\leq 1/2$). Some of the previous work in SC assumes the clean examples( before manipulation) are linearly separable, the Massart noise relaxes this assumption and considers the scenario where most of the points are linearly separable.

They show sublinear regret in the setting of online SC+Massart Noise+partial feedback. Their techniques are mostly a generalization of Zheng et al.2020 to a strategic setting. There are three steps in their algorithm: initialization, refinement, and batch contrastive inference step. In each step, they find a linear classifier $w$ that gets closer to the opt classifier $w^*$. (lines 206-217 do a great job of summarizing these steps.)

I think the main difference with Zheng et al is in the last step of batched enhancement, where they solve a sequence of adaptively constructed online linear optimization problems to refine the reported classifier $w$. Here, they need to estimate the gradients $g_{k,i}$ (you need to define what $g_{k,i}$ is, I didn't find its definition.) However, these gradients depend on the true features, but all agents in the localized regions D_{k,i} misreport their features. So, here they need to use proxy features and estimate the true location of the agents by projecting the reported feature set to the upper or lower boundary of $D_{k,i}$. They would argue that using these proxy features, the gradients will be well-constructed and hence they would converge to the optimal $w^*$ after bounded number of iterations.[This proxy feature construction is a trick also used in earlier work in online SC.]

**Questions:**

Line 168, why do you need the multiplicative factor of 2?

I don’t understand lines 218-220. What is the relationship between batches, indices and t?

Trying to understand lines 232-245, in the first paragraph, it says that you are considering the band above the classifier where the points have no incentive to move. Then in the second paragraph, it says there are challenges due to the strategic nature of the points. Could you please clarify? It seems a bit contradictory.


In line 6 in Alg 2, perhaps you can add an explanation on how you construct w_0?

**Ethical Concerns:**

["NO or VERY MINOR ethics concerns only"]

**Final Justification:**

The rebuttal has addressed some of my concerns and I decided to raise my score from 4 to 5.

**Limitations:**

yes, no potential negative societal impact

**Paper Formatting Concerns:**

no formatting concerns

**Quality:**

3

**Strengths And Weaknesses:**

I appreciate the write-up and technical details of the paper. In my opinion, this is mostly a combination of the techniques from Zheng et al 2020 on learning half-spaces under label noise and the techniques from Ahmadi et al. on online learning with strategic input, where they use proxy features to update their linear classifiers. They employ a similar idea to adapt the algorithm by Zheng et al. to a SC setting. I have explained this in detail in the summary. Please correct me if I am wrong.

---

> ### Author Rebuttal · Authors · 2025-07-31
>
> We appreciate that you like our write-up and technical details. We provide our feedback as follows.
> - Comparison with Zhang et al. (2020) and Ahmadi et al. (2021):
>
> While both Zhang et al. (2020) and Ahmadi et al. (2021) inspire separate components of our algorithm, we hope to clarify that our work considers a  fundamentally different setting. In particular, naively extending the algorithms in those papers cannot resolve the challenges unique to our setting.
>
> Zhang et al. (2020) propose an algorithm to learn halfspace under Massart noise with full feedback, without strategic manipulation. According to Zhang et al. (2020), under a noisy setting, data points near the classification  hyperplane are the most informative for learning the classifier.  Therefore, they utilize a key technique, known as “localization”, that uses only data within a small band near the classification  hyperplane. However, our problem involves agents' strategic manipulation. If we ignore the strategic manipulation from agents and naively adopt Zhang et al. (2020)'s method to announce classification hyperplanes, then agents whose true feature resides near the announced classifier, qualified or not, will strategically misreport their feature to be positively classified.  In particular, the unqualified agents who get positively classified through strategic manipulation lead to high regret.
>
> Ahmadi et al. (2021), on the other hand, investigate the problem of linear classification under agents' strategic manipulations with full feedback. Compared with their setting, ours involves Massart noise and partial feedback. If we directly extend Ahmadi et al. (2021)’s method to address agents' strategic manipulations by elevating the classification hyperplane parallelly by $\gamma$, then all agents with true features within $\gamma$ distance below the announced classification  hyperplane will strategically manipulate their features onto the classification hyperplane (see Figure 1(a) in our paper). These manipulating agents consist of both agents whose true features are within the localization band and agents whose true features are outside the band, and their reported features all lie on classification hyperplane. So it is difficult to tell whose true features are indeed within the localization band. This poses significant challenges to applying the localization technique.
>
> In short, the joint consideration of label noise, agents' strategic manipulation behavior, and partial feedback introduces challenges that are not present in the existing literature. Our key contribution consists of a novel contrastive inference method with proxy features to address these non-trivial challenges. By declaring two parallel but slightly different classification hyperplanes and comparing agent responses under these two classifiers, we can simultaneously make  inferences about true features within the localization band and largely discourage the unqualified agents from feature manipulation. This idea is unique and novel and has not been considered by the previous works.
>
> - The definition of the gradient:
>
>  We will follow your suggestion and add an explicit definition of the gradient $g_{k,i}$. It is derived from the gradient of Leaky-ReLU loss function, which is defined as $ \text{ LeakyRelu } _ { \bar\eta }(-y\langle{w},{x}\rangle) $, where $ \text{LeakyRelu} _ {\bar\eta}(z)=(1-\eta)z\mathbb{I}({z\ge 0})+\eta z\mathbb{I}({z<0}) $. Specifically, $ \mathbf{g} _ {k, i}=[-\bar{\eta} \mathbf{x} _ {k, i} \mathbb{I}\left(y _ {k, i}=1\right)+(1-\bar{\eta}) \mathbf{x} _ {k, i} \mathbb{I}\left(y _ {k, i}=-1\right)] \mathbb{I}\left(\mathbf{x} _ {k, i} \in D _ {k, i}\right) $ as we defined in Lines 270-271.
>    In our previous submission, we only briefly mentioned the loss function in a footnote, without giving its formal definition. This is because it did not appear in our subsequent theoretical analysis.
>    Instead, all of our analyses are based on the linear formulation in terms of the gradients.
>    Take the refinement stage as an example. The theoretical analyses thereof involve upper bounding $\sum_{i=1}^{T_k}\langle w_{0, i},g_{0,i}\rangle - \sum_{i=1}^{T_k}\langle w^\star,g_{0,i}\rangle$, lower bounding $\sum_{i=1}^{T_k}\langle w_{0, i},g_{0,i}\rangle$ and relating $ \langle w^\star,-g_{0,i}\rangle$ to the angle between $w_{0, i}$ and $w^\star$. These only need to leverage the form of $g_{0, i}$, without regard to its connection to the Leaky-ReLU loss.
>
> - The reason for requiring the multiplication factor of 2:
>
> Consider a general manipulation cost $c\|x-r\|_2$ for a generic constant $c$, where $x$ and $r$ are true and reported features, respectively. Note that the payoff that an agent can gain from the classification outcome is either $+1$ or $-1$, so the  manipulation distance $\|x-r\|_2$ is at most $2/c$. For notation simplicity, we set $\gamma = 2/c$ as the largest manipulation distance, leading to $c = 2/\gamma$. This is the source of the multiplication factor $2$.
>
> - The relationship between batches, indices and the global time $t$:
>
> We partition the horizon of $T$ cycles into consecutive batches indexed by $k \in \{\text{init}, 0,1,2,\cdots,K\}$. Batch $k$ has $T_k$ iterations indexed by $i\in \{1,2,\cdots,T_k\}$, where each iteration $i$ contains one or two cycles and performs only one gradient estimate followed by a single parameter update. The superscript $j= 1 \text{ or } 2$ distinguishes the two cycles inside the same iteration during Initialization and Enhancement. During Refinement, each iteration issues only one cycle. To map any $(k,i,j)$ to the global time-step $t$, we use the closed-form formula $t = 2(i-1)+j $, if $k=\text{init}$; $t = 2T _ {\text{init}}+I$, if $k=0$; $t = 2T _ {\text{init}}+T _ 0+2\sum _ {k'=1}^{k-1}T _ {k'}+2(i-1)+j$, if $k \in \{1,2,\cdots, K\}$.
>
> With this mapping, every variable indexed by $(k, i, j)$ can be immediately translated to its corresponding global time $t$.
>
> - Explanation for lines 232–245:
>
> These two sentences are indeed a bit confusing. Let us clarify here.
> In our Refinement algorithm, we set the classification hyperplane as if there is no feature manipulation ($\tilde h_{0,i}(r)=\text{sgn}(\langle w_{0,i},r\rangle)$). Then, agents with true features above the hyperplane are already classified as positive, so they have no incentive to manipulate their features ($x_{0,i}=r_{0,i}$).
> In other words, we can directly observe the true features of agents within the localization region ($\{x|\ 0<\langle w_{0,i}, x_{0,i}\rangle\le b_0\}$). This is the “band with no manipulation” mentioned in the first paragraph. In contrast, agents with true features within $\gamma$ distance below the hyperplane ($\{x| -\gamma\le\langle w_{0,i},x_{0,i} \rangle < 0 \}$) do have an incentive to project their features onto the hyperplane and report the projections. These are the “strategic” agents that cause high instantaneous regret during the Refinement phase. Indeed, with this, we can still run the localization approach as usual following Zhang et al. (2020) and refine the coefficient estimate, while bearing the cost of misclassifying unqualified agents. In the camera-ready version, we will remove the sentence "Nonetheless, strategic feature manipulation and partial label ... challenges to the localization approach" in the second paragraph to avoid confusion.
> Furthermore, to control the regret, we limit the length of Refinement phase to only $\tilde O(\ln T)$ cycles. For the rest of the decision horizon, we have to balance the need to update classifiers using data with true features within the localization region and the need to prevent high regrets due to unqualified agents' strategic manipulation. This is achieved by our novel Enhancement algorithm based on proxy features and pairwise contrastive inference.
>
> - Explanation for the construction of $\bar w_0$:
>
> We apologize that there is actually a typo in Line 6, Algorithm. The correct expression should be $$\bar w_0=\frac{1}{T_{\text{init}}}\sum_{i=1}^{T_{\text{init}}}\sum_{j=1}^{2}r_{\text{init},i}^{(j)}y_{\text{init},i}^{(j)}\mathbb{I}\left((-1)^{(j-1)}\langle{w_{\text{init},i}}{r_{\text{init},i}^{(j)}}\rangle>0\right) \ .$$
> Let us explain the intuition for this construction. In the non-strategic, noiseless and full feedback case, $y\langle{w^\star},{x}\rangle = \langle{w^\star},y{x}\rangle>0$ for all $(x,y)$, so $yx $ always forms an acute angle with the optimal normal vector $w^\star$. Analogously, under the noisy feedback setting, we should have $\langle w^\star, \mathbb{E}[yx]\rangle > 0$ so $\mathbb{E}[yx]$ forms an acute angle with $w^\star$.
> Now, consider our strategic  and partial-feedback setting. When we declare a classifier $\tilde h (r)=\text{sgn}(\langle w,r\rangle)$, agents whose true features are originally above the hyperplane ($\langle w,x\rangle>0$) have no incentive to manipulate their feature and will report truthfully ($r=x$), then $yr$ is exactly $yx$. Therefore, $r_{\text{init},i}^{(1)}y_{\text{init},i}^{(1)}\mathbb{I}\left(\langle{w_{\text{init},i}}{r_{\text{init},i}^{(1)}}\rangle>0\right)+r_{\text{init},i}^{(2)}y_{\text{init},i}^{(2)}\mathbb{I}\left(\langle{-w_{\text{init},i}}{r_{\text{init},i}^{(2)}}\rangle>0\right)=x_{\text{init},i}^{(1)}y_{\text{init},i}^{(1)}\mathbb{I}\left(\langle{w_{\text{init},i}}{x_{\text{init},i}^{(1)}}\rangle>0\right)+x_{\text{init},i}^{(2)}y_{\text{init},i}^{(2)}\mathbb{I}\left(\langle{-w_{\text{init},i}}{x_{\text{init},i}^{(2)}}\rangle>0\right)$ provides information for $\mathbb{E}[yx]$. By averaging them over the $T_{\text{init}}$ cycles, we get the estimator $\bar{w} _ 0$ that well approximates $\mathbb{E}[xy]$. So we expect that $\bar{w} _ 0$ forms an acute angle with $ w^\star $ with high probability if $T_{\text{init}}$ is large enough.
>
> Thank you again for your comments. We will incorporate our discussions into the camera-ready version to clarify the issues you raise.

---

### Note · Authors · 2025-08-12

We would like to extend our sincere gratitude to the review team once again for their valuable time, careful consideration, and insightful comments. During the rebuttal phase, we have tried our best to address the comments from the review team. We are grateful that all reviewers found our rebuttal satisfactory, and we greatly appreciate their recognition of our efforts to address their concerns. Moving forward, we will incorporate the suggested modifications into the camera-ready version.

---

### Decision · Program_Chairs · 2025-09-17

**Decision:**

Accept (spotlight)

**Comment:**

This paper makes a strong and timely contribution by studying online strategic classification with Massart noise and partial feedback, a practically motivated and technically challenging setting, with impact on both learning theory and fairness communities. Reviewers highlight that the authors successfully combine techniques from strategic classification and noisy online learning while introducing new tools such as pairwise contrastive inference and proxy feature gradients. The work delivers the first sublinear regret guarantees in this setting, which several reviewers view as a conclusive result.
The writing could be improved in places, particularly in explaining intuition behind the algorithmic steps, but overall the paper is technically solid, well-motivated, and advances the state of the art.